# PrismBench: Dynamic and Flexible Benchmarking of LLMs Code Generation with Monte Carlo Tree Search

**Vahid Majdinasab**                                        *vahid.majdinasab@polymtl.ca*
*Polytechnique Montreal, Montreal, Quebec, Canada*

**Amin Nikanjam**[*]                                        *amin.nikanjam@h-partners.com*
*Huawei Distributed Scheduling and Data Engine Lab, Canada*

**Foutse Khomh**                                        *foutse.khomh@polymtl.ca*
*Polytechnique Montreal, Montreal, Quebec, Canada*

**Reviewed on OpenReview:** *https://openreview.net/forum?id=OObsC6FDly*

## Abstract

The rapid advancement of LLMs' code generation capabilities is outpacing traditional evaluation methods. Static benchmarks fail to capture the depth and breadth of LLM capabilities and eventually become obsolete, while most dynamic approaches either rely too heavily on LLM-based evaluation or remain constrained by predefined test sets. To address these limitations, we introduce *PrismBench*, a dynamic benchmarking framework grounded in Reinforcement Learning that comprehensively evaluates LLMs by framing code generation evaluation as an RL task. We formulate the evaluation process as a Markov Decision Process over a structured tree of coding challenges, leveraging a customized Monte Carlo Tree Search algorithm to traverse this tree and discover high-failure scenarios. Our multi-agent setup orchestrates task generation, model response, and analysis, enabling scalable assessment across diverse coding challenges. Additionally, we propose metrics that combine structural traversal patterns with performance across different tasks and difficulty levels to enable diagnostic and systematic comparison of LLMs' performance. We conduct large-scale experiments on eight state-of-the-art LLMs and analyze how model architecture and scale influence code generation performance across varying coding tasks. All code, evaluation trees, and a public leaderboard are available at `https://prismbench.github.io/Demo/`

## 1 Introduction

The rapid advancement of Large Language Models (LLMs) for code generation has outpaced existing evaluation methodologies. Both static and dynamic benchmarking approaches exhibit fundamental limitations that hinder their ability to comprehensively assess evolving models' capabilities. Static benchmarks, while widely used, suffer from three key weaknesses. First, they provide a limited and often superficial assessment of an LLM's capabilities, reducing evaluation to pass/fail metrics that fail to capture the complexity of the model's reasoning McIntosh et al. (2024); Banerjee et al. (2024); Tambon et al. (2024a). Second, as benchmarks gain popularity and become evaluation targets, they are increasingly prone to being incorporated into LLMs' training data, leading to data leakage and artificially inflated performance metrics Xu et al. (2024a); Zhou et al. (2023). Third, static benchmarks lack flexibility, preventing a targeted evaluation of specific problem-solving strategies or particular aspects of model behavior McIntosh et al. (2024). To address these limitations, dynamic benchmarking approaches have been introduced Zhu et al. (2023b); Li et al. (2023); Wang et al. (2023a); Zhang et al. (2024c); Li et al. (2024c). Most dynamic approaches typically adopt one of two strategies. The first relies on LLM-as-Judge frameworks, where another LLM is used to

---

[*]Work completed while at Polytechnique Montreal.

evaluate the responses Zhu et al. (2023b); Li et al. (2024a). While this approach offers adaptability, it is inherently unreliable, as evaluation outcomes are constrained by the limitations and biases of the judge model. The second strategy involves dynamically selecting test cases from predefined datasets based on model performance Zhang et al. (2024c); Kiela et al. (2021). Although more structured, these methods remain fundamentally tied to static test sets, limiting their capacity to evolve alongside increasingly capable models. These issues indicate a clear need for evaluation frameworks/methodologies that can adapt to the increasing capabilities of LLMs, provide systematic and reproducible evaluation pipelines, and reveal comprehensive insights about the model's performance and weaknesses.

We introduce *PrismBench*, a dynamic, multi-agent benchmarking framework for evaluating LLMs' code generation capabilities. We formalize the space of all possible evaluation scenarios as a Markov Decision Process (MDP) Sutton & Barto (2018), where each state represents a unique combination of programming concepts (e.g., recursion, data structures) and difficulty levels (e.g., easy, medium, hard). Within this formulation, the evaluation process becomes a search problem where the objective is to search this space to discover evaluation scenarios (i.e., states) where models consistently fail. To systematically conduct this search, we instantiate the MDP as a search tree and use a modified Monte Carlo Tree Search (MCTS) Russell & Norvig (2016) algorithm to traverse it, balancing exploration of new evaluation scenarios with exploitation of discovered high-failure regions of the evaluation space. Instead of sampling from a static dataset, we generate each evaluation scenario dynamically based on the current state's concepts, difficulty, and prior model performance to mitigate benchmark memorization and data leakage. In this manner, as the search progresses deeper into the tree, the generated evaluation scenarios become progressively more challenging and allow for a structured and adaptive assessment of model capabilities.

Unlike prior dynamic evaluation approaches that rely on LLMs to judge each other's outputs, *PrismBench* assesses model performance by executing the generated solutions within a multi-agent sandbox. This provides objective, reproducible signals based on functional correctness, rather than textual similarity or the errors/preferences of the judge model Thakur et al. (2024). Instead, LLM agents in *PrismBench* are assigned analytical roles, for example, analyzing execution traces to identify failure patterns or examining solutions to understand how models approach specific challenges. This separation ensures that evaluation remains grounded in actual model performance, while LLM agents assist in interpreting the results and identifying patterns in model behavior. Additionally, in contrast to pass/fail metrics (e.g., accuracy, precision, etc.), which mask how models approach problems and offer little insight into their performance, we introduce metrics that track model performance across tasks, difficulty levels, and structural exploration paths, allowing for more fine-grained diagnostics and comprehensive assessments.

Overall, the contributions of our work are as follows: (1) We introduce a dynamic benchmarking framework that adapts to model capabilities and systematically explores the space of programming challenges. (2) We propose an extensible, multi-agent evaluation architecture that assesses multiple aspects of LLMs' code generation capabilities. (3) Using *PrismBench*, we evaluate eight leading LLMs to identify their failure patterns and capability boundaries. (4) We release the code for *PrismBench* alongside a leaderboard and an interactive showcase for each of the studied models at `https://prismbench.github.io/Demo`.

To structure our work, we define the following RQs:

**RQ₁:** *How can we design evaluation tasks that adapt to model capabilities and surface hidden failures?*

> To answer this question, we introduce *PrismBench*, a dynamic, multi-agent benchmarking framework that models the evaluation process as a search problem over the space of all possible evaluation scenarios. Furthermore, instead of selecting evaluation scenarios blindly or from a static pool of pre-defined challenges, *PrismBench* dynamically generates tasks in response to the model under the benchmark's performance, which allows for probing edge cases, subtle weaknesses, and regressions that are otherwise missed in static benchmarks.

**RQ₂:** *How can we evaluate different models in a dynamically generated benchmark?*

> To answer this question, we ground *PrismBench* in Reinforcement Learning (RL) by decoupling the benchmarking environment from the benchmarking process. We formalize a single, shared environment for all models and evaluate them by their traversal trajectories in this environment. To

further reduce the risk of bias arising from LLM stochasticity, we design a multi-agent interaction sandbox that tests the models' capability in test generation, solution generation, and program repair, separately. Furthermore, we propose a set of performance metrics that can be used to compare models based on their performance throughout the search process. We analyze how models perform under identical dynamic conditions, compare their behavior across multiple failure modes, and show how *PrismBench* enables more fine-grained model capability comparison than static benchmarks.

**RQ₃:** *How can dynamic benchmarks provide insights into model performance, behavior, and failures?*

To answer this question, we examine the outputs collected during the evaluation process for each model and propose a set of diagnostic metrics that offer a richer view of model performance and help identify patterns in how and why models fail.

## 2 Motivating Example

In this section, we discuss how current static and dynamic benchmarking approaches evaluate LLMs alongside their shortcomings. Regardless of the underlying evaluation methodology (static or dynamic), all benchmarking approaches face a bias/variance trade-off. Here, **bias** is a property of the evaluation *dataset* which contains tasks that consistently over/under-estimate a model's true capability (e.g., task overlap with pretraining data, coarse aggregate pass/fail metrics, uneven task distributions, etc.) Xu et al. (2024b); Roberts et al. (2023); Jiang et al. (2024). On the other hand, **variance** is a property of the *sampling process* and is a result of instability in evaluation estimates between different runs arising from small sampling pools from the underlying dataset or too few per-task trials (e.g., one extra attempt flips a failure to a pass) Dong et al. (2024); Chiang et al. (2024); Lin et al. (2024). Given the high cost of querying LLMs and finite evaluation budgets (either in time, compute, or both), to comprehensively evaluate a model, each benchmarking methodology must focus on a specific aspect of this trade-off: either increase coverage on task variety and decrease sampling per task which results in increasing variance between evaluation runs or increase resampling or per-task trials and instead focus on a narrow subset of tasks which results in high bias of evaluation results between different evaluation subsets.

Consider two LLMs, $M_A$ and $M_B$, and a bank of four coding tasks, $T_1, ..., T_4$ from an evaluation dataset (e.g., HumanEval Chen et al. (2021), MBPP Austin et al. (2021), ARC-AGI Chollet (2019), etc.). In a standard static benchmark, both $M_A$ and $M_B$ are evaluated on each task, independently (i.e., performance on $T_1$ has no effect on $T_2$), and their overall performance is summarized with aggregate metrics (e.g., pass@k, binary pass/fail over each task, etc.). Suppose $M_A$ solves $T_1, T_2, T_3$ but fails $T_4$, while $M_B$ solves $T_1, T_3, T_4$ but fails $T_2$. Under this setting, both models received a 3/4 score even though their failures were on different tasks. Such aggregate pass/fail metrics hide *where* and *how* each model fails, and compress different behaviors into the same score. While exhaustive per-task trials and normalization of scores over the number of attempts average out stochasticity and allow for fairer comparison between models, two problems remain:

- If $M_A$ consistently fails $T_4$ and $M_B$ consistently fails $T_2$, both models will still receive the same score; however, each model has failed on a different task. Here, the same score is an artifact of coarse-grained metrics, not evidence of equal capability.

- Given the scale of pretraining corpora and the unavailability of training data, some tasks may have been included in the models' training dataset. Therefore, the subsequent successes might be a result of memorization, not generalization Balloccu et al. (2024).

Moreover, as the set of underlying tasks grows, multiple samples per each task in the dataset become infeasible (e.g., for a dataset of $1k$ tasks and 10 trials per task, $10k$ queries to the model are required), forcing the bias/variance trade-off mentioned above: either the number of distinct tasks must be reduced (lower variance but higher bias from reduced coverage and selection effects) or fewer trials per each task need to be carried out (lower bias but higher variance between runs).

Dynamic benchmarking approaches aim to address these limitations by: (1) using multi-step task bundles by decomposing a task into sequential subtasks that the model must pass end-to-end (e.g., SWE-bench Jimenez

et al. (2024), Benchmark Self-Evolving Wang et al. (2024)), (2) adaptively selecting challenges from a fixed bank by picking the next task conditioned on prior evaluation results (e.g., DARG Zhang et al. (2024c), DyVal2 Zhu et al. (2024b)), or (3) generating tasks at test time to mitigate memorization (e.g. DyVal Zhu et al. (2024a), TreeEval Li et al. (2024c)). However, each approach comes with its own bias/variance trade-off. Specifically:

- Testing each model on multi-step task bundles lowers the chances of memorization and superficial success, but introduces a fairness problem in how models are evaluated against each other. $M_A$ and $M_B$ will be tested on a different distribution of sub-tasks and aggregate scoring (only end-to-end success being counted) introduces variance in evaluation results. As such, results depend heavily on the specific bundle graph and scoring rule, not just underlying capability.

- Conditioning the next task on past outcomes allows for more informative exploration of the evaluation space, but comparability issues between models' evaluation results still remain. $M_A$ and $M_B$ are now evaluated on different task distributions, as incidental failures can result in different evaluation trajectories. Stabilizing estimates still requires multiple samples per task, and given fixed evaluation budgets, variance stays high unless task coverage is narrowed, which reintroduces bias via selection effects.

- Testing each model on a problem generated at test time from a fixed distribution addresses the memorization problem and aligns the underlying task distribution across models. However, comparable estimates of models' performance still require many samples per task to control for variance, which is costly under finite budgets.

As such, both static and dynamic benchmarking approaches suffer from two main shortcomings: (1) *high bias* of evaluation results due to different evaluation task distributions and (2) *high variance* of evaluation results due to LLMs' stochasticity and limited sampling budgets Blackwell et al. (2024). In the next section, we will first describe *PrismBench*'s RL framing, which allows for defining a fixed task distribution for all models-under-test. Afterward, we establish our search methodology, which allows for adaptive sampling budget allocation through MCTS-guided exploration and enables efficient use of limited benchmarking resources by focusing on informative regions of the evaluation space.

## 3 Methodology

In this section, we first provide an end-to-end overview of *PrismBench*'s evaluation pipeline in Section 3.1. Next, we detail how we formalize the search space as an MDP and model it as a tree in Section 3.2. Afterward, we explain the details of our agents, our multi-agent orchestration approach, and the evaluation workflow at each node in Section 3.3. In Section 3.4, we explain the rationale behind a multi-phase evaluation pipeline, the details of each phase's objective, and how they integrate with the overall evaluation process. Section 3.5 outlines how we ensure the validity of dynamically generating challenges at test-time, and finally, we detail our proposed metrics for evaluating models' performance in Section 3.6.

### 3.1 *PrismBench* Overview

As mentioned in Section 2, evaluating models against a large bank of tasks under a finite evaluation budget forces a bias/variance trade-off. Conditioning the choice of the next evaluation task on prior observations mitigates this by directing trials towards informative regions of the evaluation space (i.e., evaluation tasks that maximize **information gain** regarding models' capability). To formalize this process, we model the space of all possible evaluation scenarios as an **MDP**, where each state corresponds to a unique scenario defined by a $(c, d)$ pair, with $c$ being a list of programming concepts (e.g., recursion, data structures) and $d$ being the task's difficulty level (e.g., easy, medium, hard). We instantiate the MDP as a **search tree**, where **nodes** represent states and **edges** represent transitions between scenarios, corresponding to actions that either combine additional concepts or increase difficulty. A node's **depth** is defined as the number of nodes along the path to the root, and **child** nodes may extend or share the same concepts as their **parents**.

In this manner, the search tree forms a fixed evaluation **environment** shared between all models (identical state space and transition dynamics), where transitions between nodes reflect a model's ability to generalize and solve increasingly complex tasks. Furthermore, this formalization forms a **hierarchical dependency** between evaluation tasks and reduces variance by concentrating sampling where outcomes are uncertain or high impact, and reduces bias by standardizing the task distribution between all models. As such, all models are evaluated in the same environment, based on how they explore and perform within it Sutton & Barto (2018).

To address the brittleness of end-to-end task bundles, where an early misstep can cascade into a full failure and result in different trajectories, each node in the tree is evaluated using a multi-stage, isolated pipeline. Each task is decomposed into role-based steps (e.g., test generation, solution generation, program repair, etc.) where each step is handled by a dedicated LLM **agent**. Agents interact with each other via a shared node state, which allows for modularity (addition or removal of agents) and avoids information leakage. This decoupling allows for evaluating multiple capabilities within a single node (e.g., correctly understanding the requirements, solution/test generation, debugging, etc.) without allowing failures/errors in one step dominating the final outcome. Importantly, we do not rely on LLM judgments: a node's **reward** is a composite weighted score derived from execution-grounded signals (i.e., unit-test pass rates, error traces, and bounded retry/repair attempts). As such, the reward signal reflects the specific steps responsible for success or failure instead of collapsing everything into a binary pass/fail signal for the entire task.

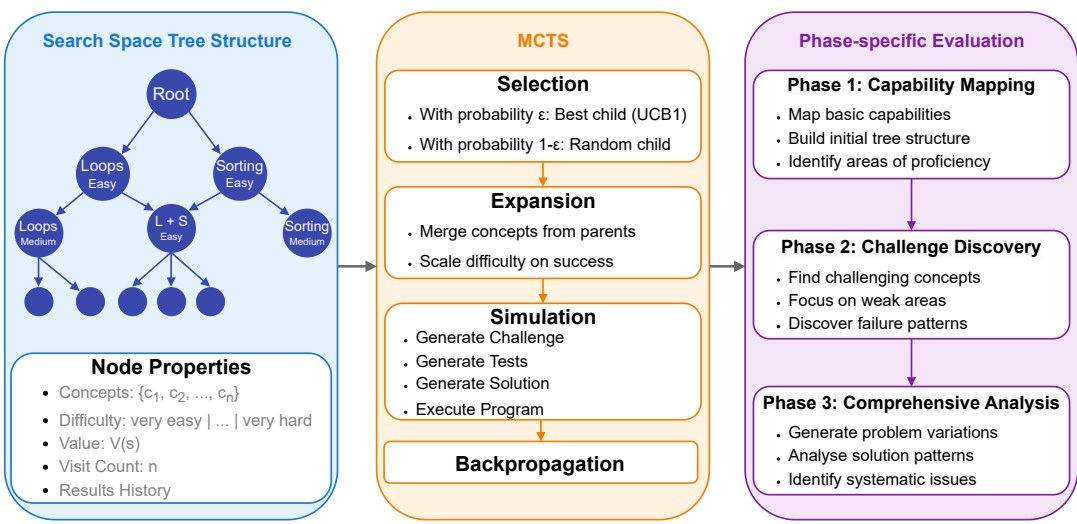

Figure 1: Overview of *PrismBench*'s search tree, agent workflow, and phased evaluation strategy.

Given the hierarchical evaluation space and execution-grounded rewards, the remaining challenge is to allocate the evaluation budget to where it most increases our knowledge about a model's capabilities. As mentioned in Section 2, multiple samples per evaluation task quickly become infeasible as the underlying dataset grows, and simply conditioning the next task on prior observations can result in different task distributions. Therefore, we need a traversal policy that balances exploration of uncertain regions with exploitation of the promising ones, while allowing for comparison between different models. As such, *PrismBench* uses **MCTS** with $\epsilon$-greedy **state selection policies** to balance exploration and exploitation over the search tree (i.e., environment) and control for LLMs' performance variability. After each node's evaluation, the observed reward is **backpropagated** along the sampled path, updating ancestors' estimates so that observations at each node (i.e., state) can inform beliefs about selection of the next node.

Finally, *PrismBench*'s search process follows a 3-**phase** strategy. The 1st phase broadly explores the tree to establish the model's baseline performance across different coding concepts and difficulty levels. The 2nd phase focuses on failure-prone nodes to discover systematic weaknesses and boundary cases. The 3rd phase focuses on low-performing regions by generating multiple variations of high-failure nodes to determine

the root causes of model failures. Figure 1 shows an overview of our state representation and evaluation pipeline. Rather than exhaustively evaluating the model on all possible scenarios, *PrismBench*'s targeted search approach adapts evaluation to the model's demonstrated capabilities in the environment, allowing for detailed analyses of model performance and failure modes.

## 3.2 Search and Tree Structure

In this section, we detail how we model the LLM evaluation task as an environment in order to dynamically benchmark LLMs' code generation capabilities and address **RQ$_1$**.

### 3.2.1 MDP Formalization

The search space, which contains all possible evaluation scenarios, is modeled as an MDP defined by the tuple $(\mathcal{S}, \mathcal{A}, \mathcal{P}, \mathcal{R})$, representing the state space, action space, transition dynamics, and reward function, respectively. We formalize the MDP as follows:

$\mathcal{S}$  denotes the state space where each state $s \in \mathcal{S}$ is a unique combination of a set of programming concepts $\mathcal{C}$, a single difficulty level $\mathcal{D}$, and a an evaluation phase identifier $\mathcal{V}$:

$$\mathcal{S} = \{(c, d, v) \mid c \subseteq \mathcal{C}, d \in \mathcal{D}, v \in \mathcal{V}\} \tag{1}$$

with $v$ denoting the evaluation phase the state belongs to which we will discuss in Section 3.3. For example, the state (`[functions, dynamic programming]`, `very easy`, `Phase 1`) represents an evaluation scenario designed to test the model on a programming challenge that requires knowledge of both "functions" and "dynamic programming" to solve, is intended to be "very easy" in terms of difficulty, and is carried out in Phase 1.

$\mathcal{A}$  denotes the action space. At each state, an action corresponds to selecting a reachable state from the current state:

$$\mathcal{A}(s) = \{a_{s \to s'} \mid s' \in \text{Children}(s)\} \tag{2}$$

with Children($s$) denoting the states connected to state $s$ via tree edges. We define two types of actions: (1) $\mathcal{A}_{select}$ which selects a previously explored state for reevaluation and (2) $\mathcal{A}_{expand}$ which selects an unexplored state for evaluation. For example, expanding the state (`[functions, dynamic programming]`, `very easy`, `Phase 1`) could yield a new state such as (`[functions, dynamic programming, conditionals]`, `very easy`, `Phase 1`), which contains a new concept, or (`[functions, dynamic programming]`, `medium`, `Phase 1`), which has an increased difficulty level, for evaluation in Phase 1 (see Section 3.1).

$\mathcal{P} : \mathcal{S} \times \mathcal{A} \times \mathcal{S} \to [0, 1]$  denotes the transition dynamics between states. Given the tree structure described in Section 3.1, transitions are only possible between states that are connected via tree edges:

$$P(s' \mid s, a_{s \to s'}) = \begin{cases} 1 & s' \in \text{Children}(s) \\ 0 & \text{o.w.} \end{cases} \tag{3}$$

with $s' \in \mathcal{S}$ being the state we transition to after taking action $a_{s \to s'}$ at state $s$.

$\mathcal{R} : \mathcal{S} \times \mathcal{A} \times \mathcal{S} \to \mathbb{R}$  denotes the phase-specific reward functions. The reward function quantifies the model's performance at each state using a composite score calculated based on multiple factors, including success rates, error penalties, challenge difficulty, and attempt counts. Each phase uses a different reward function according to the phase's goal, which we will explain in detail in Section 3.4.

### 3.2.2 Tree Representation and Traversal Mechanism

We instantiate the MDP as a tree as described in Section 3.1. To do so, we consider each state $s \in \mathcal{S}$ as a node $n$ in the tree, and each action $a \in \mathcal{A}$ determines whether to expand a node or continue exploring the tree as shown in Figure 1. Given the tree structure, it is essential to assign a value to each node in order to

guide tree traversal. To achieve this, we use the phase-specific reward function $\mathcal{R}$ to determine each node's value based on the model's performance at that node which is calculated based on how well the model solves the programming challenge associated with a given node which we will detail in Section 3.4.2. Considering the high cost of sampling and to better account for LLM's performance variability, we use **TD(0)** Sutton & Barto (2018) to incrementally estimate each node's value by incorporating past performance:

$$v(n) = v_{prev}(n) + \alpha(r - v_{prev}(n)) \tag{4}$$

where $v(n)$ is the node's value, $r$ is the immediate reward, and $\alpha$ is the hyperparameter that controls sensitivity to new observations based on benchmarking requirements. We then backpropagate $r$ to $n$'s ancestors using a discounted update:

$$v(n_a) = v_{prev}(n_a) + \gamma^{d(n)} \cdot r \tag{5}$$

where $n_a$ is an ancestor of $n$, $d_{(n)}$ is its distance from $n$, and $\gamma \in [0,1]$ is the discount factor. This allows us to incorporate results from the evaluated nodes to the ancestors and prioritize promising regions of the search space over time.

In order to traverse the tree, we use MCTS to determine the transition probabilities between nodes in the tree given the node values. Transition probabilities are calculated using MCTS based on node visit frequency and value. We use an $\epsilon$-greedy policy to balance exploration and exploitation:

$$\pi(ch|n) = \begin{cases} \text{uniform}(children(n)) & \text{with probability } \epsilon \\ \arg\max_{ch \in children(n)} UCB1(ch) & \text{with probability} 1 - \epsilon \end{cases} \tag{6}$$

Transitions between nodes (from node $n$ to its child $ch$) represent changes in difficulty or the introduction of new concepts. UCB1 Russell & Norvig (2016) is used to dynamically adjust the probabilities based on the model's historical performance using the node's value $v(n)$.

Once a node's evaluation is finished, the decision to expand it or to select another node determines how the tree grows and how the search space is explored. Node expansion $E(n)$ is governed by two criteria:

$$E(n) = \begin{cases} 1 & \text{if } v(n) \geq \theta_p \text{ and } d(n) \leq d_{max} \\ 0 & \text{otherwise} \end{cases} \tag{7}$$

where $v(n)$ is the node's value, $\theta_p$ is the normalized value threshold of each phase, $d(n)$ is the node's depth, and $d_{max}$ is the maximum allowed depth for each phase. $\theta_p$ controls the collective difficulty of each phase, with higher thresholds indicating harder acceptance criteria for solution acceptance at each node. $d_{max}$ defines a hard limit on how deep the tree can get at each phase, with higher values allowing for more in-depth analysis at each phase. When expansion is triggered, the node can be expanded in two ways:

$$\mathcal{A}_{expand}(n) = \begin{cases} \text{combine\_concepts}(n, n') & \text{with probability } p_e \\ \text{increase\_difficulty}(n) & \text{with probability } 1 - p_e \end{cases} \tag{8}$$

where $n'$ is another selected node for concept combination and $p_e$ is the probability of which expansion action is selected and tuned based on the benchmark's desired level of exploration.

The search process begins by generating foundational nodes that span the entire set of concepts at the lowest difficulty level. As the search progresses, node creation and expansion are guided by the model's performance, allowing the tree to dynamically adapt to the model's demonstrated capabilities and limitations, which allows *PrismBench* to adaptively prioritize promising areas during exploration. To enable effective exploration of the search space, we extend the tree structure to support multiple parents for each node, allowing us to represent and evaluate combinations of programming concepts, which we explain below.

### 3.2.3 Multi-Parent Tree Structure

Programming challenges rarely involve solutions based on a single concept. Instead, they typically require the application of multiple programming concepts to produce a correct solution. To model this accurately,

in *PrismBench*, each node in the search tree can have multiple parents, with each parent representing a different concept-difficulty combination. For example, a node with the concept set `[dynamic programming, recursion]` corresponds to a challenge that requires applying both concepts. This node is therefore the child of both the `[dynamic programming]` and `[recursion]` nodes, as shown in Figure 1. This setup reflects how complex challenges are often built upon simpler, foundational ones.

Furthermore, a multi-parent structure allows us to backpropagate performance signals along all relevant paths. If the model succeeds or fails on a node such as `[dynamic programming, recursion]`, the outcome is not just indicative of its proficiency in "dynamic programming" *and* "recursion". Instead, it also shows how the model is capable of handling each concept as well. This helps determine broader patterns in the search space where the model performs well or struggles, and not just isolated successes and failures. These signals then guide *PrismBench*'s exploration toward weak areas to better determine *where* and *why* the model succeeds or fails.

As such, to capture concept combinations effectively, we extend MCTS's UCB1 Russell & Norvig (2016) to support multi-parent nodes:

$$UCB1(n) = \frac{v(n)}{N(n)} + C\sqrt{\frac{\ln(\sum_{i \in parents(n)} N(p))}{N(n)}} \tag{9}$$

with $v(n)$ being the node $n$'s value, $N(n)$ being the number of times $n$ has been visited, $C$ being the exploration constant, and $parents(n)$ being the set of $n$'s parents. In this manner, a failure to solve the challenges at one node is indicative of the model showing a behavior of interest for each concept and difficulty level in that node.

### 3.3 Node Evaluation Workflow

As described in Section 3.2, each node in the tree represents a distinct evaluation scenario, defined by a set of concepts and difficulty level. The evaluation of a node consists of generating a challenge, producing corresponding tests and solutions, and iteratively refining these outputs based on the model's performance. To structure this process, we employ a set of agents where each agent has a specific role. Even though the term "agent" is well-established in RL literature, LLM providers maintain different interpretations of what constitutes an agent Shavit et al. (2023); Anthropic (2024). In order to have a uniform definition throughout our work, we adopt the definition from Roucher et al. (2025), which characterizes LLM-based agents as "programs where LLM outputs control the workflow." In this manner, *PrismBench* evaluates each node in a multi-agent sandbox where each agent handles a distinct step in the process as shown in Figure 2.

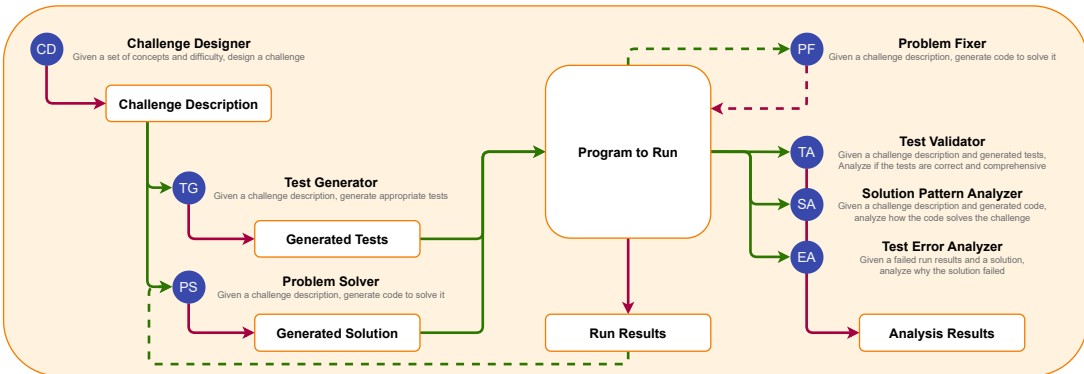

Figure 2: The evaluation begins with the *Challenge Designer* and continues until the challenge is finished. Green arrows: inputs, red arrows: outputs, dashed lines: conditional triggers.

As shown in Figure 2, for each node, evaluation starts with a coding challenge and proceeds through solution and test suite generation, multiple attempts, repairs, and analysis. To avoid data contamination, agents

are isolated, and context is only shared through the node state. Importantly, the same model is used for both test and solution generation. During solution generation, the model cannot access the tests, and during test generation, it cannot access the solution. This constraint ensures that task understanding is genuine and mirrors real programming challenges Dibia et al. (2022). In order to succeed, the model must interpret the challenge correctly Chen et al. (2024), design valid tests and solutions without access to hidden information Zhang et al. (2024b); Li & Yuan (2024), and correct mistakes using execution feedback Koutcheme et al. (2024). Isolating these steps within the pipeline allows us to evaluate each capability independently, resulting in consistent and comparable node-level evaluations across models and compare LLMs' code generation capabilities in a dynamic environment to address **RQ₂**.

### 3.3.1 Agent Roles

*PrismBench* is comprised of seven agents. All of our agents utilize prompting best practices Marvin et al. (2023); OpenAI (2025a); Anthropic (2025): clearly defined roles, structured output schemas, and few-shot examples. Our agent roles are as follows:

- *Challenge Designer*: Generates programming challenges of specified difficulty levels targeting particular computer science concepts, following an LC-style format with clear input/output specifications and constraints.

- *Test Generator*: Generates comprehensive test suites that validate functional correctness, corner cases, and performance constraints of submitted solutions while ensuring full coverage of the challenge requirements.

- *Problem Solver*: Generates code to implement a solution for the programming challenge with an emphasis on efficiency and adherence to best practices, while handling all specified corner cases and constraints.

- *Problem Fixer*: Analyzes the outputs of program execution and implementation details (solution + tests) to fix failures for the solutions and tests.

- *Test Validator*: Evaluates generated test suites for comprehensiveness, identifying potential gaps in code coverage, missing corner cases, and opportunities for improvement.

- *Test Error Analyzer*: Performs detailed analysis of test execution failures, categorizing error patterns and providing insights into the root causes of solution failures.

- *Solution Pattern Analyzer*: Examines implemented solutions to identify algorithmic approaches, data structure usage, and implementation patterns, providing metrics for solution quality and efficiency.

Listing 1 shows the system prompt for the *Challenge Designer* agent. The complete prompt templates and configuration parameters for these agents are available in our replication package Majdinasab (2025).

### 3.3.2 Multi-Agent Orchestration

Our aim in *PrismBench* is to design a general framework to evaluate LLMs on code-related tasks. Therefore, given that many LLMs lack function-calling/tool-use capabilities or may not consistently adhere to preset output formats, we designed a controlled sandbox environment that enables structured agent interactions while maintaining evaluation integrity. The evaluation for **each node** in the search tree is presented in Algorithm 1 and proceeds through the following steps:

**Challenge and Test Generation** Building a comprehensive dataset of diverse, high-quality challenges for every possible node is infeasible. To address this, the interaction cycle at each node begins with the *Challenge Designer* agent, which generates LeetCode-style programming challenges based on specified concepts and difficulty levels as shown in Listing 3. This ensures diverse evaluations per concept-difficulty pair, capturing true generalization rather than memorization or performance variation. The *Test Generator* agent is then

You are an expert computer science educator specializing in creating coding challenges. Your expertise spans various computer science concepts, and you have a knack for designing problems that are both challenging and educational. Your role is to create coding questions that test a student's understanding of specific CS concepts while also encouraging them to think critically and apply their knowledge in practical scenarios.

When given a CS concept and a difficulty level, design a problem similar to LeetCode challenges. The difficulty levels are:

- Very Easy
- Easy
- Medium
- Hard
- Very Hard

Your response should include:

1. A clear and concise problem statement
2. Input format specification
3. Output format specification
4. Constraints on input values
5. At least two examples with input and expected output
6. A brief explanation of the concept's relevance to the problem
7. The specified difficulty level

Ensure that the challenge matches the given difficulty level. Do not provide any code or solution. Focus on creating a problem that tests understanding of the given concept at the appropriate difficulty.

**IMPORTANT:** You must enclose the entire problem description within `<problem_description>` and `</problem_description>` delimiters. This is crucial for extracting the problem from your output.

Here's an example of the format you should follow, based on a LeetCode-style problem:

```
<problem_description>
Two Sum
Difficulty: Easy
Given an array of integers nums and an integer target, return indices of the two numbers such that they
add up to target. You may assume that each input would have exactly one solution, and you may not use
the same element twice.
You can return the answer in any order.
Input:
    - nums: An array of integers (2 <= nums.length <= $10^4$)
    - target: An integer ($-10^9$ <= target <= $10^9$)
Output:
    - An array of two integers representing the indices of the two numbers that add up to the target
Constraints:
    - 2 <= nums.length <= $10^4$
    - $-10^9$ <= nums[i] <= $10^9$
    - $-10^9$ <= target <= $10^9$
    - Only one valid answer exists
Examples:
    1. Input: nums = [2,7,11,15], target = 9
    Output: [0,1]
    Explanation: Because nums[0] + nums[1] == 9, we return [0, 1].
    2. Input: nums = [3,2,4], target = 6
    Output: [1,2]
    Explanation: Because nums[1] + nums[2] == 6, we return [1, 2].
Relevance to Array Manipulation and Hash Tables:
This problem tests understanding of array traversal and efficient lookup. While it can be solved with
nested loops, an optimal solution uses a hash table to achieve O(n) time complexity, demonstrating the
power of hash tables for quick lookups in coding interviews.
</problem_description>
```

Design your problem in a similar format, focusing on the CS concept and difficulty level provided.

Listing 1: System prompt for the *Challenge Designer* agent.

---

**Algorithm 1:** Agent Interaction at Each Node

---

**Input:** $C$: List of concepts, $D$: Difficulty level
**Output:** $S$: Node score, $M$: Collected metrics

**1** $challenge\_description \leftarrow \text{CHALLENGEDESIGNER}(C, D)$;
**2** $g\_t \leftarrow \text{TESTDESIGNER}(challenge\_description)$ // Generated tests
**3** $g\_s \leftarrow \text{PROBLEMSOLVER}(challenge\_description)$ // Generated solution
**4** $p\_r \leftarrow g\_s \oplus g\_t$ // Combine solution and tests
**5** $(Success, Run\_results) \leftarrow \text{RUN}(p\_r)$;
**6** **if** $Success$ **then**
**7** $\quad$ $T_{validation} \leftarrow \text{TESTVALIDATOR}(g\_t)$ // Analyze the tests
**8** $\quad$ $P_{solution} \leftarrow \text{SOLUTIONPATTERNANALYZER}(g\_s)$ // Analyze the solution
**9** **else**
**10** $\quad$ **for** $i = 1$ **to** $num\_attempts$ **do**
**11** $\quad\quad$ $e\_f \leftarrow$ errors during run // Collected errors
**12** $\quad\quad$ $f\_s \leftarrow \text{PROBLEMSOLVER}(g\_s, e\_f)$ // Generate fixed solution
**13** $\quad\quad$ $E_{analysis} \leftarrow \text{TESTERRORANALYZER}(g\_s, g\_t, e\_f)$ // Analyze the errors
**14** $\quad\quad$ $p\_r \leftarrow f\_s \oplus g\_t$;
**15** $\quad\quad$ $(Success, Run\_results) \leftarrow \text{RUN}(p\_r)$;
**16** $\quad\quad$ **if** $Success$ **then**
**17** $\quad\quad\quad$ break;
**18** $\quad$ **if** $not\ Success$ **then**
**19** $\quad\quad$ $fixed\_p\_r \leftarrow \text{PROBLEMFIXER}(p\_r)$ // Use *Problem Fixer* agent
**20** $\quad\quad$ $(Success, Run\_results) \leftarrow \text{RUN}(fixed\_p\_r)$;
**21** $M \leftarrow (T_{validation}, P_{solution}, E_{analysis}, Run\_results)$ // Store all run results
**22** **return** $S, M$;

---

tasked with generating a comprehensive test suite based on the generated challenge description. We further discuss the risks of dynamic challenge generation in the evaluation process's validity and how we mitigate those risks in Section 3.5.

**Initial Solution Attempt** The *Problem Solver* generates a solution based on the challenge description, which is then executed against the test suite generated by the *Test Generator*. Both agents operate independently, with access limited to the challenge description to prevent cross-contamination of their outputs. The sandbox environment then executes the generated solution against the generated test suite, capturing detailed metrics including passed tests count, failure types/counts, error types/counts, and execution traces as shown in Listing 2. Upon **successful completion** of all tests, the node is marked as resolved, and control returns to the search algorithm. However, if test failures occur, *PrismBench* initiates an iterative feedback process.

**Retry via Feedback** If the execution fails, a feedback phase is initialized. In the feedback phase, the *Problem Solver* agent receives execution results and error details, attempting to correct the solution. This approach serves two purposes: it accounts for the stochasticity in LLM performance which we will further describe in Section 3.4.1 (if the model fails, it is provided with context to revise its response) while also evaluating the model's capability to learn from feedback.

**Fallback to Repair** If multiple solution attempts fail to resolve the issues within a predetermined limit, a final attempt to fix the failing solution using the *Problem Fixer* agent is made. The *Problem Fixer*, which can be set as either the model under benchmark itself or a separate model depending on evaluation requirements, receives comprehensive context including the challenge description, implementation history, test suite, and failure results. Allowing the model to have access to all of the previously collected information enables assessment of the program repair capabilities of the model by providing it with full contextual information.

```
problem_statement: "## Even or Odd..."
success: True
tests_passed: 10
tests_failed: 2
tests_errored: 0
fixed_by_problem_fixer: False
data_trail:
    attempt_1:
        test_cases: "import unittest\n\n..."
        solution_code: "def solution(...)"
        output: "'Tests failed. Output:\n\n....F.....\n=.."
    attempt_2:
        test_cases: "import unittest\n\n..."
        solution_code: "def solution(...)"
        output: "All Tests passed"
```

Listing 2: An example of the run results for a node

The cycle of testing and refinement continues until the resulting *program to run* (`p_r` in Algorithm 1) executes with no errors or failures within a predetermined limit and success is achieved. Otherwise, the node is marked as failed.

**Post-Evaluation Analysis**  After the loop is finished, regardless of success or failure, the *Test Validator* reviews whether the test suite is logically aligned with the challenge. At the same time, the *Solution Pattern Analyzer* examines the final solution to extract structural and algorithmic patterns.

**Metric Aggregation**  All run results, retries, error traces, and analysis outputs are logged in the node's state, which are then transformed into scalar reward signals using the phase-specific reward function and backpropagated to parent nodes.

In this manner, our multi-agent system integrates with the search tree and MCTS through an orchestrated feedback mechanism: nodes are selected for evaluation using MCTS based on their values. Once a node is selected for evaluation, the model's performance on a generated challenge determines the node's value; the updated value, in turn, influences the next round of node selection and expansion. Furthermore, the model under benchmark can be configured to any of the specified agent roles, enabling fine-grained and targeted capability assessment. This flexibility makes *PrismBench* adaptable to diverse evaluation requirements (e.g., focusing solely on program repair or test suite generation capabilities).

### 3.4 Evaluation Phases

Our 3-phase approach guides MCTS to explore the search space using phase-specific reward functions based on each phase's evaluation strategy, as shown in Figure 3. To further account for LLM performance variability and sampling stochasticity, we define a **node value threshold**, $\Delta(v)$, per phase, and only proceed to the next phase when changes in sampled nodes' values remain within this threshold across 5 consecutive **value convergence checks**. We define our phases as follows:

**Phase 1: Capability Mapping**  This phase establishes a baseline assessment of the model's strengths and weaknesses across the concepts-difficulty space. Here, the node scoring mechanism is based on challenge success:

$$R_1(s) = b \cdot w(d) + p(s) \tag{10}$$

with $b$ being the base score for success, $w(d)$ being the difficulty weight, and harder difficulties assigned higher weights, and $p(s)$ being the penalty for failures. In this manner, higher successes result in higher rewards, which increase the node's value, which in turn encourages MCTS to further explore the search space to find challenging areas and map the model's baseline capabilities.

```
title: "Even or Odd"
concepts:
    - "conditionals"
    - "functions"
difficulty: "very easy"
description: "## Even or Odd
  Write a function that takes an integer as input and determines whether the number is even or
  ↪   odd. The function should return the string \"Even\" if the number is even, and \"Odd\" if
  ↪   the number is odd.

  ### Input:
  - n: An integer (-10^9 <= n <= 10^9)

  ### Output:
  - A string \"Even\" or \"Odd\" based on the parity of the input integer.

  ### Constraints:
  - -10^9 <= n <= 10^9

  ### Examples:
  1. Input: n = 4
     Output: \"Even\"
     Explanation: The number 4 is divisible by 2, hence it is even.

  2. Input: n = 7
     Output: \"Odd\"
     Explanation: The number 7 is not divisible by 2, hence it is odd.

  ### Relevance to Conditionals and Functions:
  This problem tests the understanding of basic conditionals, as the solution requires checking
  ↪   the remainder when the number is divided by 2. It also reinforces the use of functions for
  ↪   encapsulating logic, demonstrating how to structure a simple program."
```

Listing 3: *Challenge Designer* output for a set of concepts and difficulty level

**Phase 2: Challenge Discovery**   By focusing on Phase 1's low-value nodes, the search objective changes to finding challenging combinations of concepts and difficulties where the model consistently fails. Node scoring in this phase is based on failure rate and repeated attempts:

$$R_2(s) = \lambda(1 - r_{success}) + \eta \cdot n_{attempts} + \beta \cdot I_{fixer} \tag{11}$$

with $r_{success}$ being the ratio of successfully passed tests, $n_{attempts}$ being the number of attempts to fix a failed/errored solution, and $I_{fixer}$ indicating whether *Problem Fixer* was used. The hyperparameters ($\lambda$, $\eta$, $\beta$) weight each term according to benchmarking needs. Using the complement of success ratio assigns higher rewards to nodes with lower success rates, resulting in higher values. This produces a set of nodes where the model consistently fails and indicates the challenging areas of the search space for the model.

**Phase 3: Comprehensive Evaluation**   The objective of this phase is to reveal not just *where* but *why* the model struggles and provide insights into failures' root causes. Therefore, the underperforming nodes from Phase 2 are revisited. However, for each node in this phase, we create multiple variations (same concept and difficulty but different challenge descriptions) to distinguish between incidental failures (scenario-specific) and systematic limitations (consistent failures across variations). By analyzing the results across these variants, we collect failure traces to identify core capability gaps, whether from incorrect syntax, incorrect logic patterns, or incorrect concept implementation.

As explained above, each phase has a distinct objective, which allows us to break down the benchmarking process into smaller, focused tasks. By first mapping the model's general capabilities and then systematically

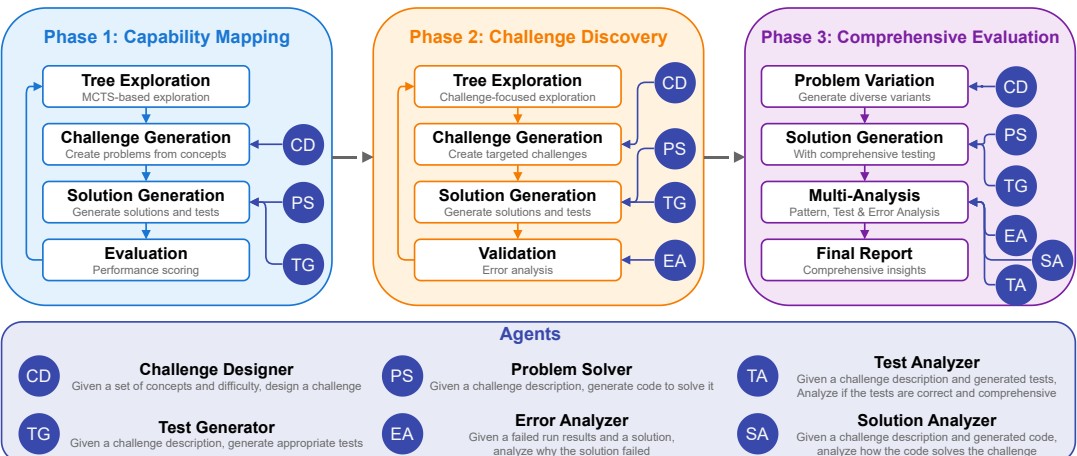

Figure 3: Overview of *PrismBench*'s evaluation pipeline and multi-phase assessment. Arrows indicate information flow and agent roles across each phase.

narrowing in on regions of consistent failure, we can iteratively refine the search space and ultimately pinpoint the root causes of these failures. As mentioned in Section 3.2, each phase's objective is defined through phase-specific state selection policies and reward functions that guide the tree traversal and expansion process, which we explain in detail below.

### 3.4.1 State Selection Policies

Studies have shown that controlling LLMs' stochasticity through low-temperature settings (e.g., $T \approx 0$) systematically reduces the diversity of their outputs Renze & Guven (2024); Peeperkorn et al. (2024). Although this performance degradation may be minor in some contexts, it becomes crucial when the objective is to comprehensively benchmark a model's capabilities. As shown by Xu et al. (2022); Ouyang et al. (2023), for code generation, at higher temperatures, LLMs explore novel solutions more effectively, while at near-zero temperatures, outputs become repetitive and risk underestimating true performance boundaries. In *PrismBench*, while we provide temperature as a tunable parameter, we preserve recommended temperature ranges rather than enforcing low-temperature values. This ensures thorough exploration of the search space but also introduces the problem of stochasticity in LLMs' responses and subsequent performance variations as a result. These performance variations are especially problematic in dynamic benchmarks that rely on LLMs as judges, where even small changes in output can lead to different assessment results. As such, we need to consider that the score for a node might not be representative of the LLM's true capability due to performance variations. To address this problem without the risk of underestimating LLMs' performance by setting low-temperature values, we define $\epsilon$-greedy state selection policies to traverse the tree as described in Equation 6. These policies mitigate the drawbacks of purely deterministic approaches (e.g., solely using UCB1 for traversal or setting the temperature to zero) by balancing exploration and exploitation and ensuring comprehensive capability assessment while mitigating performance variations. We detail the policies for each phase below.

**Phase 1: Capability Mapping** At the very beginning of the evaluation, the root generates multiple nodes as starting points for the search process. However, since the search requires the nodes to be evaluated first, the policy for evaluating the root's children (initial nodes) is defined as:

$$\pi_1^{root}(n) = \begin{cases} \frac{v(n)}{\sum_{n' \in N} v(n')} & \text{if } \forall n' \in N, \ v(n') > 0 \\ \frac{1}{|N|} & \text{otherwise} \end{cases} \tag{12}$$

This policy encourages early exploration of less-visited nodes and allows for the exploration of the initial nodes to establish a starting point for the search.

Once initial evaluations are complete, we use an $\epsilon$-greedy policy for traversing the tree:

$$\pi_1^{traverse}(ch|n) = \begin{cases} \text{uniform}(children(n)) & \text{with probability } \epsilon_1 \\ \arg\max_{ch \in children(n)} UCB1(ch) & \text{with probability } 1 - \epsilon_1 \end{cases} \tag{13}$$

This policy accounts for the stochasticity of LLMs' responses by using a uniform exploration component with a probability of $\epsilon_1$, which can be tuned based on benchmarking requirements (0.2 in our study), to mitigate the impact of occasional LLM performance variations. This ensures a thorough exploration of the LLM's capability while still focusing on promising directions using MCTS.

**Phase 2: Challenge Discovery** Similar to Phase 1, Phase 2 uses an $\epsilon$-greedy policy that focuses on challenging scenarios while maintaining exploration to mitigate LLMs' performance variations. Therefore, even though our search is guided by UCB1, we consider a small probability of selecting another state in our policy:

$$\pi_2(ch|n) = \begin{cases} \text{uniform}(children(n)) & \text{with probability } \epsilon_2 \\ \arg\max_{ch \in children(n)} UCB1(ch) & \text{with probability } 1 - \epsilon_2 \end{cases} \tag{14}$$

Similar to Phase 1, $\epsilon_2$ can be tuned based on the benchmarking requirements (0.1 in our study), with higher values resulting in more stochastic state selections and enabling more exploration of the search space regardless of how the model under benchmark performs.

**Phase 3: Comprehensive Evaluation** In Phase 3, state selection becomes deterministic based on node value thresholds (i.e., which nodes to select from Phase 2), which can be tuned depending on the benchmark's desired difficulty level:

$$\pi_3(ch|n) = \begin{cases} 1 & \text{if } v(n) > \theta \\ 0 & \text{otherwise} \end{cases} \tag{15}$$

with $\theta \in [0, 1]$ being the normalized node value threshold. Tuning $\theta$ allows for determining the benchmark's overall analysis granularity. Lower thresholds result in the selection of more nodes from Phase 2 for analysis in this phase.

### 3.4.2 Phase-Specific Reward Functions

As mentioned in Section 3.2, in each phase we employ different mechanisms to calculate the immediate received reward. We describe the details of each phase's reward calculation in the following.

**Phase 1: Capability Mapping** In Phase 1, the goal is to thoroughly map the capabilities of the model under study. Therefore, the reward function is focused on task success: the better the model is at successfully passing a challenge at a node, the higher the reward that it will receive. For this phase, the reward function for each state $s$ is defined in Equation 10:

$$R_1(s) = b \cdot w(d) + p(s)$$

with $b$ representing the base reward given to the model if it can pass the challenge regardless of the challenge's complexity or the number of attempts it took the model to solve it. $d$ is the state's difficulty level, and $w(d)$ is the weight assigned to each difficulty level, with higher difficulty levels having a higher weight. This way, the more challenging the problem the model has solved, the higher the reward it receives. Finally, the performance penalty, $p(s)$, is defined as:

$$p(s) = (r_{failed} \cdot P_{failure}) + (r_{errors} \cdot P_{error}) + ((n_{attempts} - 1) \cdot P_{attempt}) + P_{fixer} \cdot I_{fixer} \tag{16}$$

with $r_{failed}$ being the ratio of tests the model's solution failed, $r_{errors}$ being the ratio of errors in the model's solution, $n_{attempts}$ being the number of attempts it took for the model to solve the challenge, and $I_{fixer}$ being 1 if the *Problem Fixer* agent was required to fix the model's solution and 0 otherwise. $P_{failure}$, $P_{error}$, $P_{attempt}$, and $P_{fixer}$ are the weights assigned to each penalty type and are set as hyperparameters. These

hyperparameters allow for tuning the penalty's impact on the reward and therefore, provide fine-grained control of which aspects of the model's capabilities should be explored in-depth during the benchmarking process. Given that the tree generated in Phase 1 is used for all subsequent phases, high weights for errors and failures will decrease the overall reward at each node and therefore increase the overall difficulty level of the entire benchmarking process.

In this way, Equation 10 allows for mapping model capabilities: the more successful the model is at solving challenges, the higher the values for the tree's nodes, and the more MCTS is encouraged to continue exploring the search tree to find challenging areas.

**Phase 2: Challenge Discovery**  Phase 2 shifts focus from broad capability mapping in Phase 1 to systematically identifying the model's capability boundaries. Here, the reward function, as described in Equation 11, prioritizes the challenges where the model struggles. In this phase, for each state $s$, the reward is calculated as:

$$R_2(s) = \lambda(1 - r_{success}) + \eta \cdot n_{attempts} + \beta \cdot I_{fixer}$$

With $r_{success}$ being the ratio of successfully passed tests (no errors or failures). Using the complement of the success ratio assigns higher rewards to nodes where the success rate is low. Therefore, nodes that consistently expose the model's inability to generate correct solutions receive higher rewards. The hyperparameter $\lambda$ allows for controlling how aggressively the benchmark focuses on nodes with low success rates. $n_{attempts}$ is the number of attempts it took for the model to fix a solution that had failed/errored tests. Therefore, nodes requiring multiple attempts receive higher rewards and $\eta$ adjusts the weight given to repeated failures. Finally, $I_{fixer}$ is calculated in the same way as in Phase 1, with $\beta$ controlling the weight of the penalty for dependency on the *Problem Fixer* agent.

Equation 11 allows MCTS to explore regions of the search space where the model *consistently* fails (i.e., the more the model fails at each node, the higher the node's value will be). As such, Phase 2 generates a refined set of nodes where the model has constantly underperformed. These nodes will be used for analysis of the underlying root causes of poor performance in Phase 3.

**Phase 3: Comprehensive Evaluation**  Phase 3 focuses on analyzing the root causes of model failures while using the same reward formula as in Phase 2.

### 3.5 Ensuring Evaluation Validity and Comparability

As discussed in Section 2, depending on the approach, dynamic benchmarks suffer from two main shortcomings when it comes to the validity and comparability of evaluation results:

- **High bias due to different task distributions**: Evaluating models on tasks that are conditioned on models' performance on pervious tasks, will result in different models, being evaluated on different task distributions. Therefore, evaluation results cannot be compared across different models as each model will be evaluated on a different task distribution.

- **High variance due to LLM stochasticity and sampling budget**: Given the inherent stochasticity of LLMs, even with enforcing deterministic sampling parameters Atil et al. (2024), evaluation results can vary between different runs. As such, evaluations need to be carried out multiple times in order to estimate models' capabilities which reduces task diversity given the sampling budget.

In addition to the risks mentioned above, in *PrismBench* we use the *Challenge Designer* (see Section 3.3) to dynamically generate programming challenges at test-time. While doing so lowers the risk of benchmark memorization (see Section 6), it introduces two risks in return. Specifically, given a state's $(c, d)$ pair:

- **Invalid challenge**: A challenge is invalid when it is off-concept or incorrectly defined relative to the selected state's concepts $c$. For example, challenges that do not test for the intended concept $c$ (e.g., a sorting task labeled as dynamic programming), or infeasible or contradictory constraints (e.g., ambiguous input/output specifications, or evaluation criteria that cannot be met).

- **Miscalibrated difficulty**: A challenge is miscalibrated when its actual difficulty does not match the selected state's desired difficulty $d$. For example, a challenge is generated for an easy difficulty level that is actually hard or vice versa.

Furthermore, as discussed in Section 3.3, the rewards calculated for each node, are dependent on executing the generated solutions against the generated tests. While grounding evaluations on objective execution of codes reduces the risks of incorrect capability estimations, an objectively correct estimation requires that the generated solutions and generated tests to be correct. Meaning that the amount of false positives and false negatives needs to be considerably low for the evaluation outcomes to be reliable.

Below, we describe how *PrismBench*'s MDP framing, multi-parent tree structure, execution-based evaluation pipeline, TD(0) value estimation, and $\epsilon$-greedy state selection alongside multi-phase evaluation pipeline mitigate these risks.

### 3.5.1 Reducing Bias

To address the problem of different task distributions, *PrismBench* defines a fixed MDP over the space of all possible evaluation scenarios. As such, that the underlying task distribution is the same for all models (see Section 3.2). In this manner *PrismBench* becomes a fixed RL environment: the rules (Section 3.2.1), dynamics (Section 3.4.1), and reward structure (Section 3.4.2) are pre-defined and fixed. Evaluations start from the same root node and interactions with the environment are carried out under the same constraints for all models. Therefore, evaluation results between models can then become comparable not by matching identical trajectories, but instead by how well they perform in the same environment.

### 3.5.2 Reducing Variance

While constructing a fixed environment for evaluation allows for mitigating the high bias of evaluation results, LLMs' stochasticity can still result in high variance between evaluation runs. Specifically, given our methodology, there are two sources of stochasticity which can result in high variance:

- **Search and Traversal**: As detailed in Section 3.2.2, transition probabilities are calculated using MCTS and state transitions are determined by $\epsilon$-greedy policies (see Equation 6) to balance between exploration and exploitation. Therefore, different runs may follow different search trajectories, especially at the early stages of the evaluation process when value estimates are not stable.

- **LLM stochasticity**: While this aspect of LLMs' performance can be mostly controlled via tuning the sampling parameters, it does not guarantee absolute deterministic performance Atil et al. (2024). Furthermore, as discussed in Section 3.4.1, enforcing deterministic sampling systematically reduces output diversity and risks underestimating true performance boundaries.

Below, we describe how *PrismBench* accounts for these sources of stochasticity to mitigate the problem of high variance and stabilize performance estimates:

**Incremental value updates**   Given the stochasticity of LLMs, the immediate reward calculated for a node, might not be representative of the model under evaluation's true capability due to incidental successes/failures (e.g., incorrect solutions pass incorrect tests, correct solutions fail incorrect tests, etc.). Therefore, as detailed in Section 3.2.2, node values are updated incrementally with TD(0) (see Equation 4) instead of using the immediate reward, to smooth out outlier observations rather than allowing single evaluations from affecting node value estimates.

**Initial exploration policy**   As detailed in Section 3.2.2, all evaluations start with generating a set of foundational nodes that span the entire set of concepts at the lowest difficulty levels. At the beginning of the search, Phase 1 follows the initial state selection policy $\pi_1^{root}$ (see Equation 12), which relies upon the inverse value of initial nodes (instead of UCB1) to encourage early exploration of the search space. After all these initial nodes have been assigned a value, Phase 1's state selection policy is switched to $\pi_1^{traversal}$ (see Equation 13), which uses UCB1 for calculating the traversal probabilities.

$\epsilon$-**greedy exploration policies**    Relying solely on MCTS to guide tree traversal can result to over-committing to noisy early estimates of node values due to incidental performance variations Russell & Norvig (2016). Therefore, *PrismBench* uses $\epsilon$-greedy policies for state selection which allow for controlled re-sampling of alternative branches, which reduces sensitivity to any single exploration path and dampens the effect of occasional LLM performance variations (see Section 3.4.1). This in turn, allows for multiple re-visitation of leaf nodes based on their estimated value. These revisits in addition to TD(0) value estimation average out stochastic variations between search trajectories.

**Multi-parent tree structure**    *PrismBench*'s MDP is modeled over the space of all possible $(c, d)$ tuples to accurately capture the fact that programming challenges might involve the application of multiple concepts (see Section 3.2.1). Therefore, the nodes in *PrismBench*'s tree can have multiple parents as we detail in Section 3.2.3. This in turn means that the model's performance at a node with multiple concepts can inform estimates about each single concept. In this manner, observations at each leaf node can be backpropagated to all ancestor nodes with a discount factor based on their distance, to inform MCTS' search focus (see Equation 5). As such, performance signals are aggregated across the search trajectory and are not isolated to individual nodes. For example, failure at the node (`[functions, dynamic programming]`, `very easy`) can inform estimates about the models' capability to handle programming challenges involving (`[functions]`, `very easy`) and (`[dynamic programming]`, `very easy`) as well.

**Transition conditioned on value stability**    Finally, as detailed in Section 3.4, *PrismBench* only transitions between evaluation phases, after value changes across branches remain within a phase-specific threshold across multiple consecutive value convergence checks. As such, evaluation phases only progress after performance estimates have stabilized across the tree (i.e., Phase 2 focuses on consistent failures only after baseline assessments of models' capabilities in Phase 1 have stabilized).

Therefore, *PrismBench*'s estimates are less sensitive to incidental generations and early-run trajectory differences by (1) smoothing node values over repeated observations, (2) aggregating signals through discounted multi-parent backpropagation, and (3) delaying phase progression until value estimates stabilize. We show the empirical evidence of their effects in Section 5 and discuss their limitations in Section 7.

### 3.5.3   Dynamic Challenge Generation

The mechanisms discussed above allow for mitigating the high bias/variance problems of dynamic benchmarks. However, dynamic challenge generation introduces the risks of using invalid or miscalibrated challenges to evaluate models' performance and therefore, invalidating evaluation results. Below, we describe how execution based value estimations coupled with search time self-correction, account for these risks.

**Execution based value estimation**    As detailed in Section 3.2, state selection is based on past observed performance. Once a state is selected, the *Challenge Designer* agent produces a challenge based on the state's $(c, d)$, **independent** of other states at test-time. Afterwards, all rewards are derived from objective execution signals which are then used to estimate nodes' values and are backpropagated through all ancestor nodes (see Equations 4 and 5). Therefore, an invalid or miscalibrated challenge will result in an immediate reward and value estimates that deviate from past observed performance along the search trajectory. This deviation in state values informs state selection in the next iteration.

**Search-time Self-correction**    MCTS operates over a fixed MDP on $(c, d)$ states, where transitions either add concepts or increase difficulty, and at each iteration, MCTS selects a state based on its current value estimate. As detailed above, when the observed reward is inconsistent with prior observations for that state or its ancestors (e.g., unexpectedly high success at a "hard" state or repeated failure at an "easy" state), the value update produces a noticeable deviation in the state's value. This, in turn, steers *PrismBench* to revisit the invalid/miscalibrated state and its ancestors, which results in re-evaluation using fresh generations (given that challenges are generated at test time) and subsequent reward and value estimation calculations. Therefore, repeated revisits based on value estimations isolate noisy generations:

- If the original outlier was due to a one-off bad generation, subsequent evaluations will stabilize the value, and the branch is de-prioritized, resulting in MCTS exploring other branches.

- If the signal persists (meaning that previous observations were incorrect), then the value stabilizes at the new level and MCTS either expands adjacent states or prevents expansion when performance is systematically poor.

Therefore, states that produce inconsistent rewards are automatically detected for further exploration, and stable states' values converge. This feedback loop corrects for invalid/miscalibrated challenges at search time by continuous and iterative re-sampling and averaging over execution outcomes, and prevents unstable branches from affecting evaluation results.

### 3.5.4 Deterministic Challenge Selection

While the approaches described above lower the risks of evaluating models using invalid/miscalibrated challenges, as described in Section 2, there still exists the risk of the *Challenge Designer* itself being miscalibrated. Meaning that the agent used as the *Challenge Designer* has a different understanding of the difficulty levels in comparison to human preferences (i.e., the model's definition of a "hard" task corresponds to a "medium" task according to human preferences). In this case, the overall evaluation results become miscalibrated as the results of a model's performance on "hard" tasks correspond to real-world performance on "medium" tasks.

While this does not affect the benchmarking methodology itself (given execution based estimation coupled with search time self-correction), it affects how the results are interpreted. As such, we mitigate this risk by fixing the *Challenge Designer* agent throughout the entire evaluation pipeline, regardless of the model being evaluated. In this manner, all models are evaluated on the same distribution of task difficulty. Furthermore, *PrismBench*'s modular design allows integration of task evaluation frameworks, such as TaskEval Tambon et al. (2024b) or DyVal Zhu et al. (2023a), to verify each challenge's validity. Finally, dynamic challenge generation can be entirely bypassed. Specifically, given a challenge bank with defined $(c, d)$ properties for each challenge, such as CodeForces[1], the challenges for each node can be selected from said bank instead of being dynamically generated by an LLM.

### 3.5.5 No Reliance on LLM Judgments

As detailed in Sections 2 and 6, offloading assessment to a judge LLM introduces risks in both reproducibility and reliability of the evaluation results due to the limitations and biases of the judge model, especially when the objective is to evaluate models' capability boundaries. Therefore, *PrismBench* does not use any LLM judgment signal in the evaluation loop. Specifically, as detailed in Section 3.4.2, all rewards are derived exclusively from execution outcomes and are the only signals used for TD(0) updates and MCTS traversal. Therefore, the search trajectory and node value estimates are invariant to any subsequent LLM judgments.

The analyzer agents introduced in Section 3.3.1 (*Test Validator*, *Test Error Analyzer*, and *Solution Pattern Analyzer*) are utilized after a node's evaluation is finished. These agents are utilized only to derive the diagnostic metrics which we will describe in Section 3.6.4. Their outputs are never used for reward calculation, node value estimation, or traversal. As such, they can be disabled without affecting the evaluation process, which also reduces evaluation runtime and costs as we discuss in more detail in Section 4.5.

Finally, to reduce variance in the diagnostic outputs themselves, the analyzer agents are configured with structured outputs Cohere Inc. (2026); OpenAI (2025b), which validates their outputs against predefined schema at inference-time as shown in Listing 4.

---

[1]https://codeforces.com

When given a coding problem statement and test cases, analyze the test cases and ensure they properly verify the requirements of the original problem.

Given the problem statement and test cases, your response should:

1. Check that all aspects of the problem requirements are tested
2. Check that edge cases are properly covered
3. Check that test assertions are correct and meaningful
4. Check that test cases are properly structured and follow best practices
5. No redundant or unnecessary tests are present
6. Test names and descriptions are clear and accurate

When reviewing test cases, provide:

1. A list of any missing test scenarios (Missing Test Scenarios)
2. Identification of incorrect assertions (Incorrect Assertions)
3. Suggestions for improving test coverage (Suggestions for Improving Test Coverage)
4. Analysis of edge cases that should be tested (Analysis of Edge Cases)

Here's an example of the format you **SHOULD** follow, given a problem statement:

```
<test_validation>
Missing Test Scenarios:
    - edge_case: Matrix with all same values needs validation
    - boundary_values: Test with maximum allowed matrix size 100x100
    - performance_tests: Large sparse matrices need performance validation
    - error_handling: Missing tests for invalid matrix dimensions
Incorrect Assertions:
    - wrong_expectations: test checks for incorrect output values
    - invalid_assertions: test checks for incorrect number
    - incomplete_checks: test does not verify all output elements
Suggestions for Improving Test Coverage:
    - path_coverage: Add tests for all possible matrix traversal paths
    - condition_coverage: Include tests for all branching conditions
    - data_coverage: Test with different data distributions
    - path_coverage: Need coverage for diagonal traversal cases
    - condition_coverage: Add boundary condition tests
Analysis of Edge Cases:
    - boundary_conditions: Test matrix with negative elements
    - corner_cases: Matrix with all zeros needs testing
    - special_inputs: Test with floating point values
    - boundary_conditions: Maximum integer value tests missing
    - corner_cases: Single row/column matrix tests needed
</test_validation>
```

Analyze the provided test cases and provide a detailed validation report following this structure.

Listing 4: System prompt for the *Test Validator* agent.

### 3.6 Evaluation Metrics

We define four metric categories to capture distinct aspects of LLMs' code generation capabilities. These metrics provide a structured and thorough evaluation of a model's strengths, weaknesses, and solution strategies:

#### 3.6.1 Structural Metrics

These metrics focus on the tree and how models perform in the search space. Node counts and depth distributions show where models struggle (persistent exploration) or succeed (rapid convergence), and tree growth patterns demonstrate how challenge complexity impacts performance. In this manner, we can provide fine-grained and detailed insights into model performance, behavior, and failures in order to address **RQ₃**.

We denote the total number of nodes in the tree with $|N|$ and the depth of each node $n \in N$ with $D_n$. We track the distribution and connectivity of explored concepts through:

$$N(c) = \sum_{n \in \text{nodes}} 1_{[c \in \text{concepts}(n)]} \tag{17}$$

$$N(d) = \sum_{n \in \text{nodes}} 1_{[d \in \text{difficulties}(n)]} \tag{18}$$

with $N(c)$ and $N(d)$ being the number of times each concept and each difficulty was encountered throughout the entire tree. The node distribution across concepts and difficulties provides a broad view of where the model succeeds and struggles: the greater the number of nodes associated with each concept and difficulty level, the less successful the model has been in addressing related challenges. Consequently, additional nodes were generated to better identify and isolate the problematic areas.

This is complemented by the branching factor at each node:

$$B(n) = \frac{\text{children}(n)}{|N|} \tag{19}$$

where children$(n)$ is the number of children of node $n$ and $|N|$ represents the total number of nodes. Nodes with higher branching factors have more children compared to the other nodes and, therefore, have been more challenging for the model.

The convergence rate $C(n)$ measures the stability of a model's performance at each node $n$ by measuring the difference between consecutive TD values:

$$C(n) = |v_{t+k} - v_t| < \epsilon \text{ for } k = 1, \ldots, K \tag{20}$$

where $v_t$ represents the node's TD value at attempt $t$. The convergence rate reflects how drastically the model's output changes between attempts. A phase is terminated when all nodes exhibit convergence rates below a predefined threshold $\epsilon$ for $K$ consecutive attempts. A lower convergence rate indicates greater stability, meaning the model's performance has plateaued at node $n$. When this condition holds across all nodes in a phase, the phase is deemed to have been sufficiently explored, and the next phase begins.

#### 3.6.2 Mastery Metrics

These metrics focus on the model's progress in understanding and applying concepts over the course of the benchmarking process. These metrics quantify performance stability as challenge complexity increases and success rates on challenges with combinations of concepts across benchmarking phases.

The primary measure of concept mastery is the success rate:

$$SR(c) = \frac{1}{N(c)} \sum_{n \in N} \text{success}(n) \tag{21}$$

where success$(n)$ is 1 if the model has successfully passed the challenge at node $n$ and 0 otherwise, as shown in Listing 2.

Similarly, we measure the model's success rate at each difficulty level:

$$SR(d) = \frac{1}{N(d)} \sum_{n \in N} \text{success}(n) \tag{22}$$

To understand the effort required for solving a challenge related to a concept $c$, we measure the average number of attempts regardless of success or failure:

$$A(c) = \frac{1}{N(c)} \sum_{n \in N(c)} \text{attempts}(n) \tag{23}$$

with attempts$(n)$ representing the number of attempts made at node $n$ as shown in Listing 2.

These three metrics alongside each other, indicate how well the model performs in solving challenges for each specific concept/difficulty with the average number of attempts indicating how many times the model encountered errors while solving the challenge. High success rates and low number of attempts indicate a high capability (the challenge was solved with a low number of errors and attempts) while lower success rates and higher number of attempts indicate struggles in solving challenges with that specific concept/difficulty.

### 3.6.3 Performance Metrics

Performance metrics build upon the mastery metrics and assess the model's performance across different concepts and difficulty levels by providing a granular understanding of the model's capabilities using challenge success rates, the number of interventions required to fix the model's code, and problem-solving efficiency across concepts and difficulty levels.

The fixer intervention rate indicates when the model requires external help and is unable to solve the challenge:

$$F(c) = \frac{1}{N(c)} \sum_{n \in N(c)} 1_{[I_{fixer}(n)]} \tag{24}$$

with $I_{fixer}$ being 1 if the *Problem Fixer* agent was used at each node $n$ and 0 otherwise.

As shown in Algorithm 1 the *Problem Fixer* agent is only used when the model fails in all of its attempts to solve the challenge. This is caused by either incorrect solutions or incorrect tests. Therefore, we can measure the model's program repair capabilities for each concept by tracking whether the use of *Problem Fixer* resulted in success:

$$R(c) = \frac{\sum_{n \in N(c)} 1_{[\text{success}(n)]}}{\sum_{n \in N(c)} 1_{[I_{\text{fixer}}(n)]}} \tag{25}$$

where success$(n)$ is 1 if the model has successfully passed the challenge at node $n$ and 0 otherwise after the *Problem Fixer* intervention at node $n$.

### 3.6.4 Diagnostic Metrics

These metrics reveal behavioral patterns through solution analysis (preferred coding patterns), error categorization (common failure modes), and test set evaluation (correct tests, testing for corner cases, etc). They capture how the model succeeds or fails in specific scenarios and allow for identifying and characterizing the model's behavior (how it solves the challenges and how it fails).

The distribution of solution patterns across concepts shows how many times the model has used a specific solution for each concept $c$:

$$SP(p, c) = \frac{\text{count}(p, c)}{N(P)} \tag{26}$$

with $N(P)$ being the number of identified patterns throughout the entire tree and patterns indicating algorithmic approaches, data structure usage, and implementation patterns.

Pattern effectiveness quantifies which solutions the model executes successfully, helping identify its preferred problem-solving strategies:

$$PE(p) = \frac{\sum_{n \in N(p)} 1_{[\text{success}(n)]}}{N(P)} \tag{27}$$

Conversely, using failure rate $(1 - success(n))$ instead of success rate quantifies the patterns the model struggles with the most.

Test validation scores test suite quality:

$$TV(v,c) = \frac{\text{count}(v,c)}{N(V)} \tag{28}$$

where $N(V)$ is the number of identified validation issues throughout the entire tree with $v$ being an identified validation issue with validation issues including analyses on missing, incorrect, coverage, and corner case issues for each generated test suite for each concept $c$.

Error pattern distribution by concept shows where exactly the model has failed in solving the challenges related to that concept $c$:

$$EP(e,c) = \frac{\text{count}(e,c)}{N(E)} \tag{29}$$

where $N(E)$ is the number of identified errors throughout the entire tree with $e$ being an error that was raised during the execution of the program.

### 3.6.5  Comparative Analysis of Metrics

Code generation benchmarks differ in methodology and therefore in the metrics they provide. However, several metrics such as pass@k(probability at least 1 of k generated solutions passes tests), accuracy/solve rate (fraction of tasks passing all related tests), unit-test pass rate (percentage of provided tests passed), and diagnostics related to test quality and failure modes are inherent to code generation and are reported in different contexts according each benchmark's objectives. *PrismBench*'s metrics, adhere to the same general formulation, but differ in *how* they are collected and *where* they are derived from.

- **Accuracy/solve rate**: These metrics corresponds directly to *PrismBench*'s node-level execution outcomes, and are reported as success rates across the concepts and difficulty levels throughout the tree as mastery metrics (see Section 3.6.2).

- **Success@k**: In *PrismBench*, we do not sample $k$ independent solutions per challenge scenario (pass@k). Instead, we evaluate end-to-end task success (spec understanding, test generation, solution generation, and program repair) under a fixed attempt budget as shown in Algorithm 1. Therefore all mastery and performance metrics are reported as success@k: the probability that the model solves a challenge end-to-end within $k$ attempts, where an attempt corresponds to one full evaluation iteration with execution feedback Yang et al. (2023); Wang et al. (2023b).

- **Unit-test pass rate / partial credit**: Similar to test-driven benchmarks, *PrismBench* grounds correctness in execution of generated solutions against tests. However, in *PrismBench* we consider test generation as an important aspect of coding capability, and as such, pre-defined unit-tests do not exist. Instead, we incorporate these metrics in the reward function (see Section 3.4.2). For each attempt, we record the total number of passed, failed, and errored tests alongside error/failure breakdowns, which provides both test pass rates and partial credit. We aggregate these metrics across the tree over concepts and difficulties and report them as diagnostic metrics(see Section 3.6.4).

As discussed in Section 2, one of the limits in reporting metrics over the entire evaluation suite regardless of the details of the evaluation tasks is that aggregate performance signals become noisy and differences in

models' capabilities on different task definitions become obscured by global pass/fail metrics. Therefore, in *PrismBench*, we categorize all reported metrics over (concept, difficulty) pairs and define additional metrics that allow for identifying not only whether models succeed on a set of tasks, but *where* and *how* they fails across different coding capabilities and domains.

# 4 Experimental Design

To show *PrismBench*'s effectiveness, we evaluate 8 LLMs on their code generation, test suite creation, and program repair capabilities: **GPT4o** (4o), **GPT4o-mini** (4o-M) Hurst et al. (2024), **GPT-OSS-20b** (GPT-OSS) Agarwal et al. (2025), **Llama3.1-8b** (L-8b), **Llama3.1-70b** (L-70b), **Llama3.1-405b** (L-405b) Dubey et al. (2024), **Llama4-Scout** (L4S) Meta (2025), and **DeepSeekV3** (DS3) Liu et al. (2024). For our experiments, we use LeetCode (LC) LeetCode (2024) style programming challenges and 4o-M as the *Challenge Designer* to create problems based on each node's concepts and difficulty levels for **all** models under evaluation. The test generation, code generation, and repair tasks are performed by the models under evaluation, while 4o is used for analyzer agents in Phase 3. All reported results are averaged over 3 independent benchmarking runs for all models under study.

For all LLMs under study, the concepts are chosen similarly to the fundamental concepts of computer science in LC, with difficulty levels of "very easy", "easy", "medium", "hard", and "very hard", the same as the difficulties of LC challenges, which we describe in detail below.

## 4.1 Problem Formulation

As described in Section 3.2, we define the evaluation space as an MDP over all possible evaluation scenarios where each state is defined by a unique `(concepts, difficulty)` pair. Once a state is selected, the model under study must solve an end-to-end coding challenge as detailed in Section 3.3: given a natural-language specification, the model must generate a solution, generate tests, and iteratively repair its code based on execution feedback as shown in Algorithm 1.

For our experiments, we instantiate each state using dynamically generated LC-style programming challenges as the LC challenge format (i.e., given a challenge description a solution must be submitted and refined until all tests pass) matches *PrismBench*'s node evaluation workflow. It should be noted, that the LC-style challenge descriptions are an *instantiation* of an end-to-end programming workflow and as detailed in Section 3.5, *PrismBench* can be configured to use alternative problem formulations as well. We further discuss the limitations of using LC-style challenges for our experiments in Section 7.

## 4.2 Concepts

Here, we provide a concise explanation of each concept and what we expect the models to achieve in tasks involving these concepts.

- **Loops**: A loop is a control structure that repeatedly executes a block of code as long as a specified condition is true. Examples include `for`, `while`, and `do-while` loops. As such the models should:
  - Correctly implement loops to traverse data structures or repeat operations.
  - Optimize loop usage for efficiency and avoid common pitfalls such as infinite loops.

- **Conditionals**: Conditionals are control structures that execute specific code blocks based on boolean conditions. Examples include `if`, `else`, and `else if` statements. We expect the model to:
  - Accurately implement conditionals to manage decision-making logic.
  - Handle edge cases and ensure logical correctness when combining multiple conditions.

- **Functions**: Functions are reusable blocks of code that perform a specific task, defined by a name, parameters, and a return value. The models should:
  - Design modular and reusable functions.

- – Handle parameter passing and scope effectively.

- **Data Structures**: Data structures organize and store data to facilitate efficient access and modification. Examples include arrays, linked lists, stacks, queues, and trees. The models should:

  - – Choose appropriate data structures for given problems.
  - – Implement and manipulate data structures accurately and handle edge cases.

- **Algorithms (logic)**: Step-by-step procedures for solving problems or performing computations. As such, the models should:

  - – Devise efficient algorithms to address specified problems.
  - – Optimize time and space complexity, demonstrating an understanding of computational trade-offs.

- **Error Handling**: Error handling involves detecting, managing, and responding to runtime errors. As such, the models should:

  - – Implement robust error-handling mechanisms, including exception handling and validation.

- **Recursion**: Recursion is a technique where a function calls itself to solve a problem by breaking it into smaller sub-problems. As such, the models should:

  - – Correctly implement recursive functions, ensuring termination through base cases.
  - – Optimize recursion to avoid excessive memory usage and stack overflow issues.

- **Sorting**: Sorting involves arranging data in a specific order, such as ascending or descending, such as quicksort, mergesort, and bubble sort. As such, the models should:

  - – Implement sorting algorithms correctly and select appropriate algorithms for the given data size and constraints.

- **Searching**: Searching involves finding specific elements in a dataset, such as linear search, binary search, and hash-based lookups. As such, the models should:

  - – Apply efficient search techniques suited to the dataset's structure.
  - – Ensure correctness and handle cases where the element is not present.

- **Dynamic Programming**: Dynamic programming is a technique for solving complex problems by breaking them into overlapping sub-problems and solving each sub-problem only once. We expect the models to:

  - – Develop dynamic programming solutions to problems requiring optimization.
  - – Demonstrate the ability to use memoization or tabulation correctly.

These concepts are foundational to CS and cover the essential problem-solving skills required to implement solutions and tests for a problem. By benchmarking models on these concepts, we aim to assess their ability to generalize to unseen tasks based on single concepts and concept combinations critical for coding and reasoning. The concepts for benchmarking are modifiable, meaning that they can be changed to any desired topic, allowing *PrismBench* to be used in more specific scenarios and subjects (e.g., instead of foundational concepts, implementation patterns and challenges closer to LC challenges such as "Two Sum", "Valid Sudoku", etc. can be chosen).

### 4.3 Combination of Concepts

As we detailed in Section 3.2, in real-world programming scenarios, the implementation of solutions rarely requires implementing isolated, single concepts. Instead, they require the integration of multiple concepts to address complex problems effectively. For example, developing a functional application often involves combining loops for iteration, conditionals for decision making, and data structures to organize information. In addition, advanced challenges frequently require recursion, algorithms for processing logic, and error handling to ensure that the program does not fail when it encounters unexpected inputs or conditions.

Therefore, to simulate real-world programming scenarios, *PrismBench* generates challenges that combine these core concepts into unified problems. This allows us to evaluate a model's capabilities to synthesize knowledge across programming concepts. For example, a single problem might require using dynamic programming alongside data structures for optimal solutions or using sorting and searching techniques to manage/query datasets. This approach ensures that the model can demonstrate competency in scenarios requiring cross-concept integration. As such, failure to solve problems involving multiple concepts is an indicator of deficiencies in one or more of the constituent concepts. Such failures signal areas where the model struggles to integrate distinct methodologies or lacks a deep understanding of specific concepts. For instance, if a model fails a task combining functions and error handling, it might reflect difficulties in managing exceptions within modularized code. In this manner, *PrismBench* can investigate these failures further by identifying the exact concepts or combinations responsible for failures.

Alongside combining concepts, we also use a range of difficulty levels: very easy, easy, medium, hard, and very hard in order to perform fine-grained analysis of the model's capabilities. This enables us to assess performance not only on single concepts and their combinations but also on different complexities of these problems. For example, a model might perform well on easier problems related to a concept or group of concepts but fail on medium or hard ones, revealing limitations in its ability to scale solutions to more challenging scenarios.

By probing models across a variety of concept combinations and difficulty levels, we gain a comprehensive understanding of their strengths and weaknesses and gain valuable insights into their overall code generation capabilities by pinpointing root causes and systematically evaluating a model's limitations.

### 4.4 Experiment Settings and Reproducibility

In this section, we report the configuration values for all global and phase-specific parameters used throughout *PrismBench*'s multi-phase evaluation pipeline for the experiments reported in our study.

**Configuration parameters** We organize *PrismBench*'s configurable parameters into a set of global traversal settings that control the overall search and value-estimation behavior, alongside phase-specific evaluation parameters that control the reward functions and exploration policies at each stage. Specifically:

- **Traversal parameters**: The traversal parameters (e.g., discount factor, learning rate, etc.) control *how* the evaluation budget is allocated across the search space throughout the evaluation process. These parameters determine which states are sampled, how frequently states are revisited, and how quickly value estimates propagate. These parameters do not change what constitutes as success or failure at each state or influence the calculated rewards.

- **Evaluation parameters**: The evaluation parameters (i.e., phase-specific penalty weights) control *what* the evaluation process prioritizes. These parameters determine the evaluation objective such as, emphasizing program repair capabilities over strict correctness of generated solutions, or weighting test failures more heavily than test errors. These parameters directly influence the calculated rewards at each state but do not change the traversal behavior.

Table 1 lists all of *PrismBench*'s configurable parameters alongside the values used for our experiments. Since our aim with *PrismBench* is to propose an evaluation methodology, we expose these parameters as configurable settings to support different evaluation needs and objectives. Importantly, different configurations

correspond to different evaluation objectives and, consequently, different evaluation processes. As such, to obtain comparable results across models, all reported values were kept **fixed** across all of the models under study to ensure comparability and reproducibility of the results.

Table 1: Tunable parameters

| Description | Type | Parameter | Value |
|---|---|---|---|
| *General* | | | |
| Discount factor (Eq. 5) | Traversal | $\gamma$ | 0.9 |
| Learning rate (Eq. 4) | Traversal | $\alpha$ | 0.9 |
| Exploration constant (Eq. 9) | Traversal | $C$ | 1.414 |
| Number of convergence checks | Traversal | $-$ | 5 |
| *Phase 1* | | | |
| Node value threshold (Eq. 7) | Traversal | $\theta_p$ | 0.4 |
| Value delta threshold (Sec. 3.4) | Traversal | $\Delta(v)$ | 0.3 |
| Random exploration probability (Eq. 13) | Traversal | $\epsilon_1$ | 0.2 |
| Penalty per failure (Eq. 16) | Evaluation | $P_{\text{failure}}$ | 2 |
| Penalty per error (Eq. 16) | Evaluation | $P_{\text{error}}$ | 3 |
| Penalty per attempt (Eq. 16) | Evaluation | $P_{\text{attempt}}$ | 1 |
| Penalty for using Problem Fixer (Eq. 16) | Evaluation | $P_{\text{fixer}}$ | 5 |
| Max depth (Eq. 7) | Traversal | $d_{\max}$ | 5 |
| *Phase 2* | | | |
| Node value threshold (Eq. 7) | Traversal | $\theta_p$ | 0.6 |
| Value delta threshold (Sec. 3.4) | Traversal | $\Delta(v)$ | 0.1 |
| Random exploration probability (Eq. 14) | Traversal | $\epsilon_2$ | 0.1 |
| Max depth (Eq. 7) | Traversal | $d_{\max}$ | 10 |
| *Phase 3* | | | |
| Number of variations for each node | Traversal | $-$ | 5 |
| Node value threshold (Eq. 7) | Traversal | $\theta_p$ | 0.5 |

**Agent configurations**   Table 2 shows the agent configurations used during each model's evaluation. As outlined in Section 4, we use 4o-M as the *Challenge Designer* across all experiments. For each model being benchmarked, we use that model as the *Test Generator*, *Problem Solver*, and *Problem Fixer* agents. To avoid bias in error analysis, we use 4o as the *Test Validator*, *Test Error Analyzer*, and *Solution Pattern Analyzer* for all models except when benchmarking 4o itself. In that case, we use L-405b as the analyzer agent to prevent bias.

Table 2: Experiments configurations

| Agent | 4o | 4o-M | L-8b | L-70b | L-405b | L4S | DS3 | GPT-OSS |
|---|---|---|---|---|---|---|---|---|
| Challenge Designer | 4o-M | 4o-M | 4o-M | 4o-M | 4o-M | 4o-M | 4o-M | 4o-M |
| Test Generator | 4o | 4o-M | L-8b | L-70b | L-405b | L4S | DS3 | GPT-OSS |
| Problem Solver | 4o | 4o-M | L-8b | L-70b | L-405b | L4S | DS3 | GPT-OSS |
| Problem Fixer | 4o | 4o-M | L-8b | L-70b | L-405b | L4S | DS3 | GPT-OSS |
| Test Validator | L-405b | 4o | 4o | 4o | 4o | 4o | 4o | 4o |
| Test Error Analyzer | L-405b | 4o | 4o | 4o | 4o | 4o | 4o | 4o |
| Solution Pattern Analyzer | L-405b | 4o | 4o | 4o | 4o | 4o | 4o | 4o |

### 4.5 Benchmarking cost

Evaluating each node in the search tree requires independent calls to each agent, with the process being dependent on the model's performance. For every node, we use one query for the *Challenge Designer* to generate the challenge, one for the *Test Generator*, and one for the *Problem Solver*. If the model succeeds on the first attempt, the evaluation ends with these three queries. However, if the solution fails, up to three additional queries are used for the *Problem Solver* for iterative repair attempts. If still unsolved, one additional query is used for *Problem Fixer* to repair the solution. As detailed in Section 3.4, nodes may also be revisited up to five times if their children consistently receive low rewards (convergence checks up to 5 times). Finally, for each node generated in Phase 3, three more queries are used for the *Test Validator*, *Test Error Analyzer*, and *Solution Pattern Analyzer* agents. Therefore, the per-node benchmarking cost in our framework ranges from a minimum of 3 queries (one-shot success, no retries) to a maximum of 38 queries (persistent failures, repeated convergence checks, and full diagnostic analysis). On average, throughout all the trees in our experiments, a single node has triggered 6 queries.

The total query count for a full benchmarking run is dependent on the model's capabilities and provider costs (if using APIs). For the experiments conducted in this study, all models were accessed via their provider APIs. The benchmarking process (including execution of generated codes) was run on a single Apple M1 machine with 16 GB RAM. Table 3 shows the number of queries, total cost per model, and wall-clock runtime, averaged over 3 independent runs. Reported wall-clock runtimes include both API latency and local code/test execution time.

Table 3: Benchmarking Cost

| Model | Number of Queries | Total Cost (US$) | Wall Clock Runtime (Minutes) |
|---|---|---|---|
| 4o | 1,153 | 20 | 40.2 |
| 4o-M | 961 | 10 | 34.6 |
| GPT-OSS | 1,363 | 16 | 45.3 |
| L-405b | 1,094 | 24 | 55.3 |
| L-70b | 454 | 14 | 21.1 |
| L-8b | 183 | 8 | 16.3 |
| L4S | 480 | 10 | 23.3 |
| DS3 | 884 | 13 | 77.4[*] |

[*] Elevated wall-clock time due to high provider-side latency and rate-limits observed during runs.

### 4.5.1 Reducing Benchmarking Costs

As mentioned above, evaluating each node in the search tree requires 35 calls in the worst-case scenario. While *PrismBench* significantly lowers the sampling requirements for comprehensive evaluation of LLM capabilities, the workflow of using all 7 agents for each node can become computationally and financially expensive. As such, *PrismBench* allows for a lightweight version of the benchmarking process by using challenges generated from previous runs and skipping diagnostic metric calculation. Specifically:

- By using a bank of generated challenges from previous runs, the *Challenge Designer* agent can be bypassed.

- The 3 analyzer agents (*Test Validator*, *Test Error Analyzer*, and *Solution Pattern Analyzer*) are only used for analysis of the benchmarking results in order to compute the diagnostic metrics described in Section 3.6 and other metrics are not dependent on LLM analysis. Therefore, they can be turned off in case diagnostic metrics are not required.

The combination of these two solutions reduces the number of LLM calls per node in the worst-case scenario from 38 to 15, and reduces computational and financial costs of the benchmarking process by up to 50%.

# 5 Results and Analysis

In this section, we present the experimental results and analysis of our dynamic benchmarking approach, as described in Section 3. We begin with a comparative analysis of the eight LLMs introduced in Section 4 by evaluating their code generation performance using the metrics defined in Section 3.6. Afterwards, we provide a fine-grained breakdown of how each model performed across different dimensions in Section 5.2. Finally, in Section 5.3, we analyze the effects of model scale (i.e., number of parameters) and how it impacts code generation capabilities.

## 5.1 Comparative Analysis

In this section, we provide a comparative analysis of evaluation results across the four metric categories discussed in Section 3.6 for the models under study.

### 5.1.1 Structural Metrics

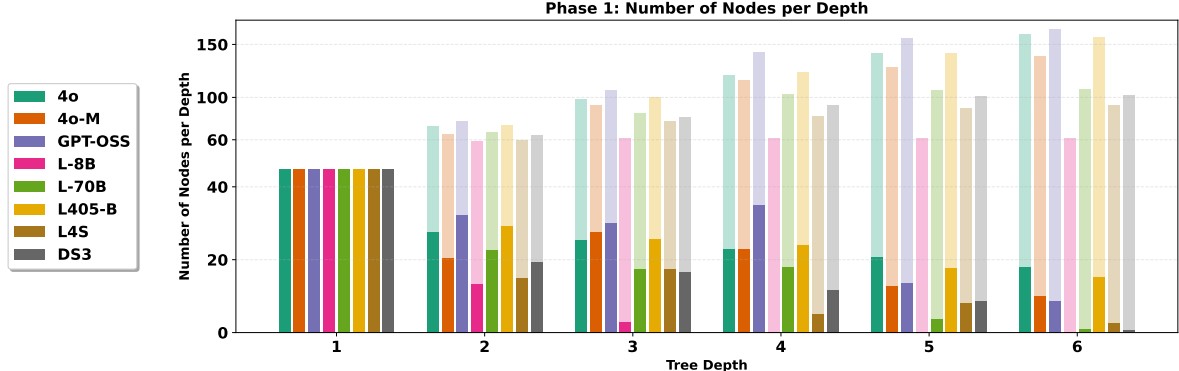

Figure 4: **Tree growth across models throughout Phase 1** The bars represent node counts by tree depth, and the shaded bars represent the cumulative number of nodes per depth across the tree in *Phase 1*.

Figure 4 compares the tree growth per depth in Phase 1. At the very beginning of the search, the tree is populated with initial nodes in order to provide the same starting point for all models and allow comparison between their performance as the evaluation process continues (see Section 3.4.1). Phase 1's reward function prioritizes task success and difficulty-weighted exploration (see Section 3.4.2). Therefore, the shaded bars for each model quantify how effectively they sustain problem-solving capability as challenges become more complex (i.e., we go deeper in the tree): higher number of nodes indicates broader exploration and lower failures. For instance, 4o and GPT-OSS achieve more than 150 nodes in Phase 1, demonstrating robust handling of complex challenges (e.g., multi-concept and high-difficulty tasks), while L-8b stalls at 60 nodes, failing beyond basic concepts and easy difficulties (depth<4).

Phase 2 uses low-scoring Phase 1 nodes to generate targeted challenges, prioritizing *task failure* and *repeated attempts*. Therefore, the ratio of generated nodes per depth, as shown in Figure 5, reveals where models struggle: higher ratios at shallower depths imply difficulty with simpler challenges, while increasing ratios at greater depths demonstrate stronger problem-solving capability at complex challenges. We can observe that even though 4o-M has a similar number of nodes than L-405b at the end of Phase 1 (142 vs 151 nodes), it struggles with complex challenges in Phase 2, while L-405b demonstrates a more consistent exploration of the tree and has a higher ratio of nodes compared to 4o-M at the end of Phase 2. Furthermore, we can observe that GPT-OSS has a similar problem-solving capability to 4o and L-405b despite being much smaller in scale, which we discuss in detail in Section 5.2.

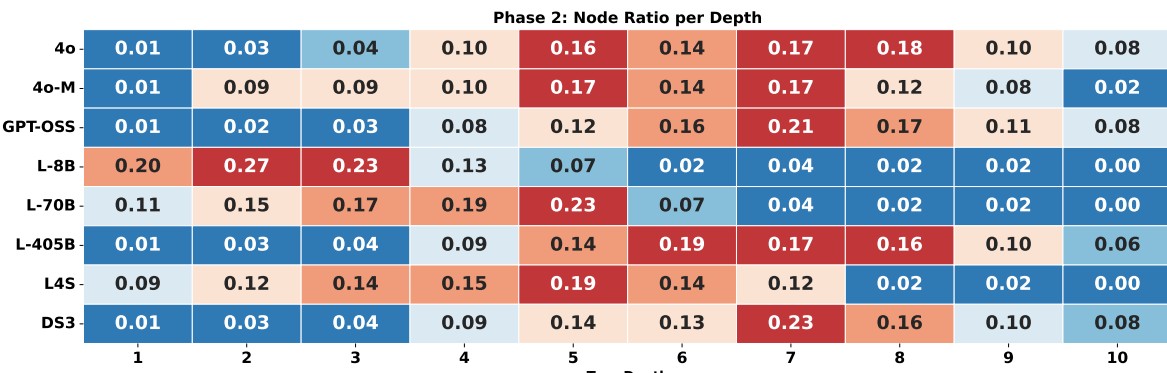

Figure 5: **Node ratio by depth across models throughout Phase 2.** Each cell shows the ratio of nodes in the tree at each depth, indicating relative search focus across the tree in *Phase 2*.

### 5.1.2 Performance Metrics

Table 4 summarizes model capability analysis at the end of the benchmark, with values showing *failure rates* across concepts and difficulty levels. *PrismBench* dynamically explores the search space to find challenging areas for the model, then focuses on these areas to uncover root causes of failure. The primary operational capability for each concept is determined by the ratio of nodes (concept-difficulty pairs) explored in the search tree and their average failure rates over 3 independent runs.

Table 4: **Model capability analysis by concept and difficulty**. Values represent failure rates (higher = more challenging). Colors indicate performance: green (good) to red (poor). † indicates primary operational difficulty level (most number of nodes), ✓ indicates mastered concepts (failure rate < 0.01), and ✗ indicates concepts beyond current capability (failure rate > 0.99). 95% CI broken down by concept and difficulty reported in Tables 5 and 6

| Concept | Very Easy/Easy | | | | | | | | Medium | | | | | | | | Hard/Very Hard | | | | | | | |
|---|---|---|---|---|---|---|---|---|---|---|---|---|---|---|---|---|---|---|---|---|---|---|---|---|
| | 4o | 4o-M | GPT-OSS | L405 | L70 | L8 | L4S | DS3 | 4o | 4o-M | GPT-OSS | L405 | L70 | L8 | L4S | DS3 | 4o | 4o-M | GPT-OSS | L405 | L70 | L8 | L4S | DS3 |
| Algorithms | ✓ | ✓ | ✓ | 0.50 | 0.81† | 0.78† | 0.30 | ✓ | ✓ | 0.53 | 0.39 | ✓ | 0.91† | 0.92† | 0.62† | ✓ | 0.66† | 0.89† | 0.40 | 0.73† | ✗ | ✗ | 0.70† | 0.67† |
| Conditionals | ✓ | ✓ | ✓ | ✓ | 0.81† | 0.79† | ✓ | ✓ | 0.75† | ✓ | 0.33 | 0.85† | 0.92† | ✗ | 0.41 | 0.32 | 0.67† | 0.88† | 0.44 | 0.64† | ✗ | ✗ | 0.48 | 0.71† |
| Data Struct. | ✓ | 0.60 | ✓ | ✓ | 0.80† | 0.77† | 0.29 | ✓ | 0.75† | 0.53 | 0.29 | 0.85† | 0.98† | ✗ | 0.50 | 0.48 | 0.67† | 0.88† | 0.40 | 0.68† | ✗ | ✗ | 0.35 | 0.77† |
| Dyn. Prog. | ✓ | 0.60 | 0.29 | 0.50 | 0.77† | 0.78† | 0.62† | ✓ | ✓ | 0.53 | 0.32 | 0.85† | 0.88† | ✗ | 0.78† | 0.53 | 0.67† | 0.89† | 0.58 | 0.71† | ✗ | ✗ | 0.54 | 0.89† |
| Error Hand. | ✓ | ✓ | 0.26 | ✓ | 0.82† | 0.78† | ✓ | ✓ | 0.75† | 0.53 | ✓ | 0.85† | ✗ | ✗ | 0.39 | ✓ | 0.67† | 0.89† | 0.38 | 0.71† | ✗ | ✗ | 0.87† | 0.30 |
| Functions | ✓ | 0.60 | ✓ | ✓ | 0.82† | 0.79† | ✓ | ✓ | 0.75† | 0.53 | ✓ | 0.85† | ✗ | 0.92† | 0.69† | ✓ | 0.66† | 0.90† | 0.44 | 0.70† | ✗ | ✗ | 0.57 | 0.51 |
| Loops | ✓ | ✓ | ✓ | ✓ | 0.81† | 0.79† | 0.33 | ✓ | ✓ | 0.53 | ✓ | 0.85† | 0.81† | 0.92† | 0.70† | ✓ | 0.67† | 0.88† | 0.29 | 0.71† | ✗ | ✗ | 0.38 | 0.86† |
| Recursion | ✓ | ✓ | ✓ | 0.50 | 0.79† | 0.77† | ✓ | ✓ | ✓ | 0.53 | ✓ | 0.85† | ✗ | ✗ | 0.60 | ✓ | 0.66† | 0.89† | 0.35 | 0.68† | ✗ | ✗ | 0.57 | 0.72† |
| Searching | ✓ | ✓ | ✓ | 0.50 | 0.80† | 0.78† | 0.32 | ✓ | 0.75† | 0.53 | ✓ | 0.85† | ✗ | ✗ | 0.52 | ✓ | 0.67† | 0.89† | 0.28 | 0.70† | ✗ | ✗ | 0.67† | 0.54 |
| Sorting | ✓ | 0.60 | ✓ | ✓ | 0.79† | 0.78† | 0.26 | ✓ | ✓ | 0.53 | ✓ | 0.85† | 0.92† | 0.98† | 0.39 | 0.42 | 0.66† | 0.89† | 0.30 | 0.71† | ✗ | ✗ | 0.47 | 0.70† |

4o shows no failures on easy tasks, demonstrating strong basic programming skills. However, performance drops at higher difficulty levels, especially for "dynamic programming" and "data structures", indicating limitations in handling programming challenges that require in-depth reasoning. L-405b fails on some easy challenges, but generally has lower failure rates on easy and medium tasks. Similar to 4o, it struggles with hard/very hard challenges that require integration of multiple concepts, such as "dynamic programming", "algorithms", and "functions". 4o-M has higher failure rates overall among the top-performing models, especially for challenges requiring compositional reasoning, such as "loops", "functions", "conditionals", and "recursion". These failures are more common when concepts are combined (e.g., loops with conditionals), as shown in Figure 20 and discussed in detail in Section 5.2.

Both GPT-OSS and DS3 display higher levels of capability than 4o and L-405b, achieving lower failure rates on medium and hard/very hard tasks. While this is indicative of better coding capability across concepts and difficulty levels, the majority of GPT-OSS's and DS3's successes, are a result of relying on Python's built-in functions and standard libraries (e.g., using `sort` on arrays instead of implementing the sorting function, using `itertools` to iterate through multiple arrays at once instead of implementing the functionality, etc.) **despite** task instructions that prohibit such solutions. Therefore, while GPT-OSS and DS3 are more capable in

solving challenges in contrast to 4o and L-405b, they are less capable in instruction following and adhering to specified requirements, which are revealed through the diagnostic metrics (error patterns and test validations) which we discuss in detail in Section 5.2.

L4S's higher success rates are a result of it reaching much fewer nodes at higher depths compared to both 4o and L-405b as shown in Figure 5, which indicates L4S's relative search focus, and Figure 7, which displays the success rates weighted by the number of node visits. On the other hand, L-70b and L-8b show different patterns: L-70b struggles even with easy challenges and fails more as difficulty increases, indicating a limited capacity for complex challenges. L-8b has high failure rates across all concepts and difficulties, indicating limitations in basic code generation capability. We provide a more detailed analysis of each model's performance and the effects of model scale in Sections 5.2 and 5.3.

### 5.1.3 Mastery Metrics

Figures 6 and 7 present *success rates* by concept and difficulty for the top models across Phases 1 and 2 *weighted* by the number of node visits for each concept at each difficulty level (higher number indicates more success at fewer attempts). While Performance metrics capture overall challenge outcomes (i.e., success/failure), mastery metrics highlight which concepts each model handles well and where they struggle. By mapping success rates to each concept-difficulty pair and weighing them by the number of visits to associated nodes, we can pinpoint common failure modes and determine each model's limits.

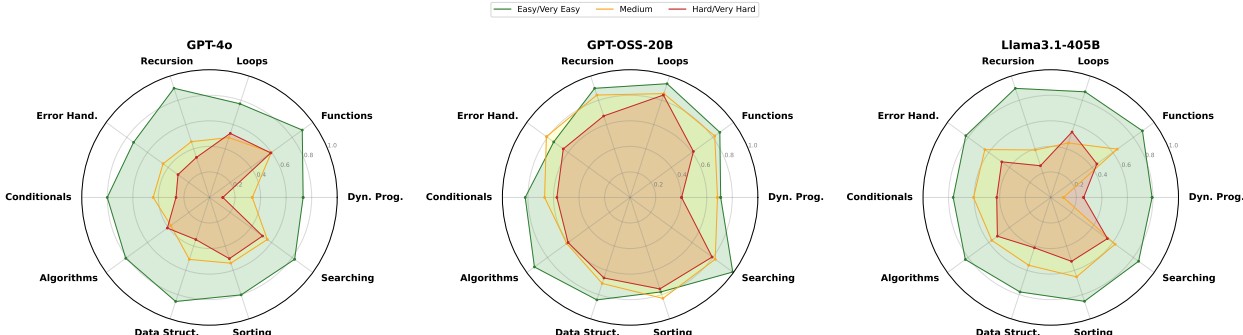

Figure 6: **Concept success rate analysis per difficulty for 4o, GPT-OSS, and L-405b**. Green: very easy/easy, Yellow: medium, Red: hard/very hard. The radial axis represents the success rate (between 0 and 1). Each axis corresponds to a programming concept. Higher values indicate better performance.

As shown in Figure 6, GPT-OSS outperforms all models on medium and hard/very hard difficulty challenges, particularly in "loops", "searching", "sorting", "data structures", and "algorithms" with a marginally lower capability in function-heavy composition and "dynamic programming". 4o struggles with compositional reasoning: challenges that require combining "algorithms", "data structures", or "dynamic programming", especially when multiple function calls, nested conditionals, or multiple levels of recursion are required. L-405b shows strong performance on very easy/easy challenges and similar performance to 4o on medium difficulty control flow and data structure tasks. However, similar to 4o, it struggles with complex challenges such as "dynamic programming" or "recursion" on hard/very hard difficulty levels.

4o-M shows high capability on easy/very easy challenges, but its success rates degrade as difficulty levels increase, specifically across challenges that require "recursion", nested conditionals, and "dynamic programming". On the other hand, while DS3 shows high success rates on easy and medium level difficulties, its performance sharply degrades on challenges involving "dynamic programming", "error handling", and loop-intensive compositions. As mentioned in Section 5.1.2 and shown in Figure 7, even though L4S achieves high success rates on hard/very hard difficulty challenges, it is incapable of solving them consistently because it visits far fewer nodes with such levels of difficulty. As such, when weighted by the number of node visits, L4S's success rates are much lower compared to other top models. We include a more detailed analysis of concept combinations and their effects on model performance in Section 5.2.

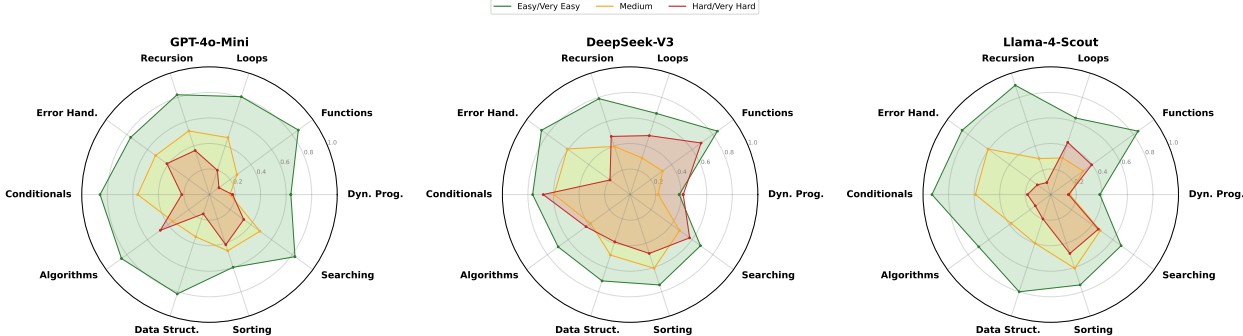

Figure 7: **Concept success rate analysis per difficulty for 4o-M, DS3, and L4S**. Green: very easy/easy, Yellow: medium, Red: hard/very hard. The radial axis represents the success rate (between 0 and 1). Each axis corresponds to a programming concept. Higher values indicate better performance.

### 5.1.4 Diagnostic Metrics

Figures 8 and 9 show the success ratios of the top-performing models for the four highest failure rate concepts from Table 4, grouped by the top three programming patterns found in the solutions of each model.

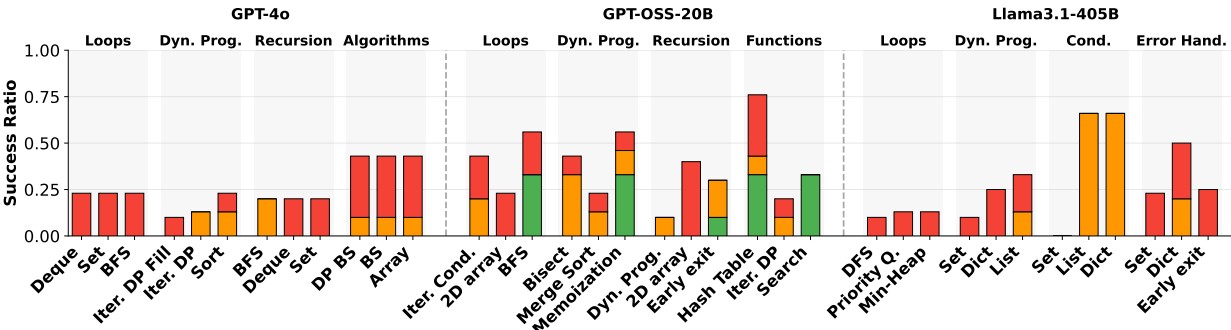

Figure 8: **Success ratios for the most challenging programming patterns for 4o, GPT-OSS, and L-405b**, grouped by the 3 most challenging concepts for each model. Stacked bars show performance by difficulty. Green: medium, orange: hard, red: very hard. Taller bars indicate better performance.

As shown in figure 8, 4o struggles significantly with "dynamic programming", even when the concept is not explicitly in the challenge. In contrast to the other models under study, the majority of GPT-OSS's solutions involve multiple nested function definitions (i.e., defining functions inside other functions), which, when combined with GPT-OSS's over-reliance on Python's standard libraries to either create hash tables or search through the inputs in intermediate steps, result in failures. L-405b shows the lowest success ratios for simple "data structures" and "tree/graph traversal". In contrast to the other top models, our analysis shows that L-405b's failures are not due to a lack of understanding of the problem itself but from failures in instruction-following and programming syntax. L-405b's failed solutions are often implemented using built-in data types (set, list, dict, etc.) and while the logic and pseudocode are often correct, L-405b frequently makes errors such as hallucinating keys in built-in types (using incorrect attributes), misplacing code snippets (calling a variable before defining it), or failing to follow the system prompt's format, which lead to immediate rejection of solutions by the framework.

On the other hand, 4o-M consistently fails in challenges involving composite problems (combinations of multiple concepts), "complex data structures", or "dynamic programming", regardless of how it attempts to solve the challenge. Similar to GPT-OSS, DS3 relies on Python's standard libraries and built-in functions in order to solve the challenges; however, the majority of DS3's failures are a result of incorrect input type and shape estimation (e.g., assuming the input will be a 1D array and failing when the input is higher-dimensional).

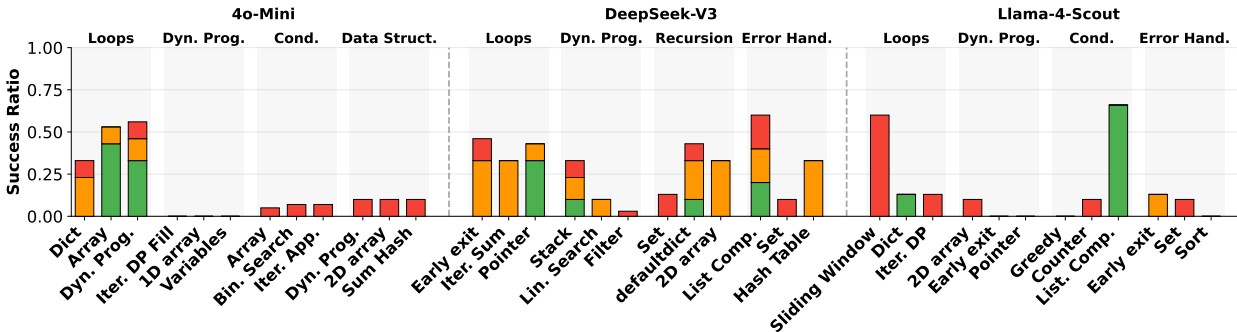

Figure 9: **Success ratios for the most challenging programming patterns for 4o-M, DS3, and L4S**, grouped by the 3 most challenging concepts for each model. Stacked bars show performance by difficulty. Green: medium, orange: hard, red: very hard. Taller bars indicate better performance.

Compared to 4o and L-405b, DS3 extrapolates requirements the most without considering other possibilities, which is the main root cause behind its consistent failures for solving challenges involving "error handling". Finally, L4S struggles the most with challenges requiring iterative processing of inputs or intermediate results. We provide a comprehensive breakdown of these observations through per-model analysis and cross-model comparison in Section 5.2.

## 5.2 Detailed Analysis of Results

Tables 5 and 6 show the average success rate and average intervention rate for each of the models under study, across concepts and difficulties, respectively. The metrics presented here are averaged from the values throughout the entire tree at the end of the benchmarking process, for 3 independent benchmarking runs for each model, and are not phase-specific.

Table 5: **Model performance by difficulty**. Colors indicate performance: green (good) to red (poor). Higher values for intervention rates indicate more usage of the *Fixer* agent. Each cell shows the mean value over 3 runs ± margin of error, calculated at a 95% confidence interval.

| Difficulty | 4o | | DS3 | | GPT-OSS | | L405b | |
|---|---|---|---|---|---|---|---|---|
| | Avg Succ. Rate | Avg Inter. | Avg Succ. Rate | Avg Inter. | Avg Succ. Rate | Avg Inter. | Avg Succ. Rate | Avg Inter. |
| Very easy | $0.83 \pm 0.11$ | $4.67 \pm 0.02$ | $0.83 \pm 0.01$ | $3.33 \pm 0.12$ | $0.89 \pm 0.05$ | $1.33 \pm 0.04$ | $0.83 \pm 0.04$ | $3.67 \pm 0.03$ |
| Easy | $0.73 \pm 0.12$ | $2.00 \pm 0.02$ | $0.64 \pm 0.11$ | $1.00 \pm 0.12$ | $0.85 \pm 0.09$ | $3.33 \pm 0.02$ | $0.72 \pm 0.10$ | $2.33 \pm 0.07$ |
| Medium | $0.42 \pm 0.01$ | $1.00 \pm 0.01$ | $0.51 \pm 0.02$ | $0.33 \pm 0.04$ | $0.75 \pm 0.04$ | $4.33 \pm 0.09$ | $0.39 \pm 0.10$ | $1.33 \pm 0.01$ |
| Hard | $0.33 \pm 0.09$ | $2.00 \pm 0.04$ | $0.64 \pm 0.12$ | $0.33 \pm 0.11$ | $0.60 \pm 0.12$ | $2.67 \pm 0.09$ | $0.46 \pm 0.07$ | $1.00 \pm 0.04$ |
| Very hard | $0.29 \pm 0.08$ | $2.00 \pm 0.10$ | $0.28 \pm 0.05$ | $0.00 \pm 0.01$ | $0.49 \pm 0.03$ | $9.67 \pm 0.12$ | $0.29 \pm 0.07$ | $2.50 \pm 0.06$ |
| | 4o-M | | L4S | | L70b | | L8b | |
| | Avg Succ. Rate | Avg Inter. | Avg Succ. Rate | Avg Inter. | Avg Succ. Rate | Avg Inter. | Avg Succ. Rate | Avg Inter. |
| Very easy | $0.83 \pm 0.10$ | $1.00 \pm 0.03$ | $0.84 \pm 0.02$ | $1.12 \pm 0.04$ | $0.42 \pm 0.08$ | $3.00 \pm 0.05$ | $0.19 \pm 0.11$ | $1.67 \pm 0.06$ |
| Easy | $0.63 \pm 0.03$ | $1.00 \pm 0.09$ | $0.66 \pm 0.03$ | $1.00 \pm 0.03$ | $0.22 \pm 0.01$ | $0.00 \pm 0.05$ | $0.22 \pm 0.11$ | $1.00 \pm 0.03$ |
| Medium | $0.35 \pm 0.02$ | $1.00 \pm 0.01$ | $0.51 \pm 0.08$ | $1.33 \pm 0.08$ | $0.16 \pm 0.09$ | $0.00 \pm 0.01$ | $0.03 \pm 0.06$ | $0.00 \pm 0.10$ |
| Hard | $0.21 \pm 0.11$ | $1.50 \pm 0.03$ | $0.33 \pm 0.12$ | $2.33 \pm 0.04$ | $0.21 \pm 0.12$ | $1.00 \pm 0.02$ | $0.11 \pm 0.06$ | $0.00 \pm 0.09$ |
| Very hard | $0.24 \pm 0.08$ | $1.00 \pm 0.10$ | $0.29 \pm 0.01$ | $0.00 \pm 0.10$ | $0.20 \pm 0.07$ | $1.00 \pm 0.05$ | $0.07 \pm 0.06$ | $1.00 \pm 0.06$ |

Looking at the performance data across all models, we observe a clear hierarchy in both success rates and the number of interventions. Starting with the model performance by difficulty level, there's a consistent degradation in success rates as difficulty increases across all models. As expected, the "very easy" difficulty level shows the highest success rates for all models. The success rates steadily decline to much lower values

Table 6: **Model performance by concept**. Colors indicate performance: green (good) to red (poor). Higher values for intervention rates indicate more usage of the *Fixer* agent. Each cell shows the mean value over 3 runs ± margin of error, calculated at a 95% confidence interval.

| Concept | 4o | | DS3 | | GPT-OSS | | L405b | |
|---|---|---|---|---|---|---|---|---|
| | Avg Succ. Rate | Avg Inter. | Avg Succ. Rate | Avg Inter. | Avg Succ. Rate | Avg Inter. | Avg Succ. Rate | Avg Inter. |
| Loops | $0.51 \pm 0.10$ | $2.50 \pm 0.03$ | $0.47 \pm 0.02$ | $0.33 \pm 0.04$ | $0.73 \pm 0.08$ | $5.00 \pm 0.05$ | $0.48 \pm 0.11$ | $3.00 \pm 0.06$ |
| Conditionals | $0.42 \pm 0.03$ | $4.33 \pm 0.09$ | $0.69 \pm 0.03$ | $0.00 \pm 0.03$ | $0.65 \pm 0.01$ | $4.33 \pm 0.05$ | $0.43 \pm 0.11$ | $2.33 \pm 0.03$ |
| Data Struct. | $0.43 \pm 0.02$ | $4.33 \pm 0.01$ | $0.52 \pm 0.08$ | $0.00 \pm 0.08$ | $0.68 \pm 0.09$ | $5.00 \pm 0.01$ | $0.43 \pm 0.06$ | $3.33 \pm 0.10$ |
| Algorithms | $0.49 \pm 0.11$ | $4.50 \pm 0.03$ | $0.55 \pm 0.12$ | $1.00 \pm 0.04$ | $0.66 \pm 0.12$ | $4.67 \pm 0.02$ | $0.50 \pm 0.06$ | $3.00 \pm 0.09$ |
| Dyn. Prog. | $0.29 \pm 0.08$ | $2.50 \pm 0.10$ | $0.32 \pm 0.01$ | $0.00 \pm 0.10$ | $0.56 \pm 0.07$ | $4.33 \pm 0.05$ | $0.42 \pm 0.06$ | $3.33 \pm 0.06$ |
| Error Hand. | $0.49 \pm 0.08$ | $1.67 \pm 0.01$ | $0.74 \pm 0.05$ | $0.33 \pm 0.05$ | $0.70 \pm 0.10$ | $8.00 \pm 0.07$ | $0.55 \pm 0.04$ | $2.33 \pm 0.12$ |
| Functions | $0.57 \pm 0.10$ | $2.67 \pm 0.05$ | $0.65 \pm 0.04$ | $0.33 \pm 0.03$ | $0.75 \pm 0.10$ | $5.00 \pm 0.12$ | $0.50 \pm 0.01$ | $2.00 \pm 0.10$ |
| Recursion | $0.49 \pm 0.10$ | $2.00 \pm 0.10$ | $0.53 \pm 0.04$ | $1.00 \pm 0.01$ | $0.70 \pm 0.03$ | $5.67 \pm 0.04$ | $0.55 \pm 0.03$ | $3.33 \pm 0.02$ |
| Searching | $0.51 \pm 0.11$ | $1.50 \pm 0.06$ | $0.62 \pm 0.08$ | $0.00 \pm 0.08$ | $0.74 \pm 0.05$ | $5.67 \pm 0.04$ | $0.49 \pm 0.06$ | $3.33 \pm 0.10$ |
| Sorting | $0.45 \pm 0.03$ | $1.00 \pm 0.06$ | $0.60 \pm 0.03$ | $0.33 \pm 0.07$ | $0.70 \pm 0.09$ | $7.67 \pm 0.12$ | $0.48 \pm 0.07$ | $2.33 \pm 0.06$ |
| | 4o-M | | L4S | | L70b | | L8b | |
| | Avg Succ. Rate | Avg Inter. | Avg Succ. Rate | Avg Inter. | Avg Succ. Rate | Avg Inter. | Avg Succ. Rate | Avg Inter. |
| Loops | $0.48 \pm 0.11$ | $2.00 \pm 0.02$ | $0.31 \pm 0.01$ | $2.81 \pm 0.12$ | $0.21 \pm 0.05$ | $1.00 \pm 0.04$ | $0.10 \pm 0.04$ | $1.00 \pm 0.03$ |
| Conditionals | $0.45 \pm 0.12$ | $2.00 \pm 0.02$ | $0.51 \pm 0.11$ | $2.25 \pm 0.12$ | $0.32 \pm 0.09$ | $3.00 \pm 0.02$ | $0.18 \pm 0.10$ | $2.00 \pm 0.07$ |
| Data Struct. | $0.44 \pm 0.01$ | $1.00 \pm 0.01$ | $0.46 \pm 0.02$ | $3.46 \pm 0.04$ | $0.24 \pm 0.04$ | $1.00 \pm 0.09$ | $0.11 \pm 0.10$ | $0.00 \pm 0.01$ |
| Algorithms | $0.48 \pm 0.09$ | $1.00 \pm 0.04$ | $0.55 \pm 0.12$ | $3.60 \pm 0.11$ | $0.33 \pm 0.12$ | $1.00 \pm 0.09$ | $0.17 \pm 0.07$ | $1.00 \pm 0.04$ |
| Dyn. Prog. | $0.34 \pm 0.08$ | $0.00 \pm 0.10$ | $0.19 \pm 0.05$ | $4.21 \pm 0.01$ | $0.23 \pm 0.03$ | $1.00 \pm 0.12$ | $0.08 \pm 0.07$ | $1.00 \pm 0.06$ |
| Error Hand. | $0.49 \pm 0.05$ | $1.00 \pm 0.03$ | $0.42 \pm 0.04$ | $2.03 \pm 0.06$ | $0.27 \pm 0.02$ | $1.50 \pm 0.02$ | $0.13 \pm 0.07$ | $1.00 \pm 0.02$ |
| Functions | $0.42 \pm 0.06$ | $1.50 \pm 0.06$ | $0.59 \pm 0.10$ | $2.42 \pm 0.05$ | $0.24 \pm 0.01$ | $2.00 \pm 0.12$ | $0.23 \pm 0.08$ | $2.00 \pm 0.09$ |
| Recursion | $0.43 \pm 0.02$ | $1.00 \pm 0.07$ | $0.13 \pm 0.02$ | $4.77 \pm 0.09$ | $0.29 \pm 0.05$ | $1.00 \pm 0.11$ | $0.20 \pm 0.10$ | $1.00 \pm 0.06$ |
| Searching | $0.47 \pm 0.10$ | $1.50 \pm 0.04$ | $0.67 \pm 0.12$ | $2.32 \pm 0.02$ | $0.30 \pm 0.01$ | $1.00 \pm 0.11$ | $0.20 \pm 0.04$ | $1.00 \pm 0.05$ |
| Sorting | $0.39 \pm 0.02$ | $0.00 \pm 0.04$ | $0.60 \pm 0.02$ | $2.28 \pm 0.07$ | $0.31 \pm 0.05$ | $1.00 \pm 0.08$ | $0.13 \pm 0.11$ | $1.00 \pm 0.06$ |

at "very hard" difficulties. The success rates of L-70b and L-8b even on the "very easy" difficulty level compared to the other models, already indicate the limited capability of these models given the number of their parameters, which we discuss in depth in 5.3.

In terms of concept mastery, we see varying performance across models. 4o performs best on "functions" and "searching" challenges, while struggling with "dynamic programming". 4o-M shows more consistent performance across concepts but with lower overall success rates. L-405b demonstrates solid capabilities on "error handling" and "searching" challenges while also struggling with "conditionals", and similar to 4o and 4o-M, on "dynamic programming". The smaller Llama models (L-70b and L-8b) show significantly lower success rates across all concepts, with L-8b particularly struggling with success rates mostly below 0.20.

Both 4o and L-405b show notably high intervention rates, especially at the "very easy" difficulty level (4.67 and 3.67, respectively). This is particularly interesting given that these models also maintain high success rates. Investigating node distributions helps explain these patterns with Figures 10a and 10b displaying the distribution of nodes in each depth per concept and difficulty for 4o and L-405b, respectively. Both models quickly progress beyond "very easy" difficulty challenges, as evidenced by their node distributions (15 and 14 nodes at "very easy" for 4o and L-405b, respectively). As such, the high number of interventions at lower difficulties are due to smaller sample sizes at these levels combined with specific challenging cases requiring multiple interventions. On the other hand, we can observe that both 4o and L-405b have high intervention rates for challenges related to "conditionals", "data structures", "algorithms", and "dynamic programming". Looking at the distributions of nodes per concept as shown in Figure 10a and 10b reveals that these concepts also have a high number of nodes in the deeper parts of the tree, meaning that *PrismBench* has identified that these concepts at high complexities have shown to be challenging for the models and has focused on these areas in order to thoroughly analyze models' capabilities.

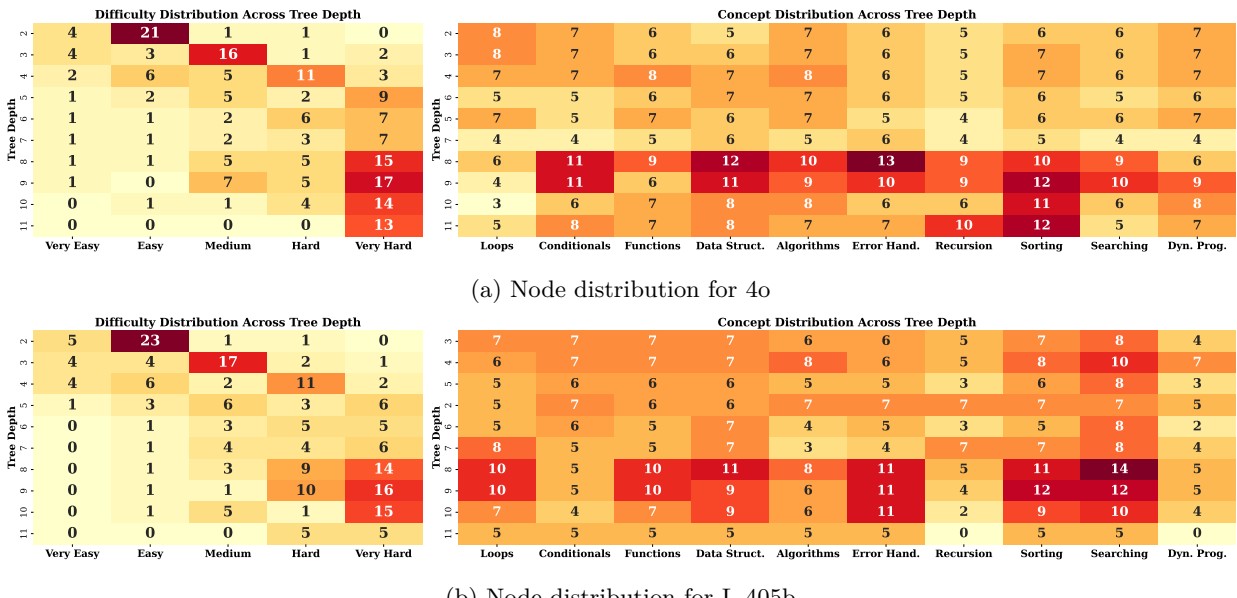

(a) Node distribution for 4o

(b) Node distribution for L-405b

Figure 10: Node distributions for 4o and L-405b averaged over 3 independent runs. The numbers in each cell indicate the number of nodes.

Since interventions in *PrismBench* are performed by the model itself through the *Problem Fixer*, the combination of success rate and number of interventions effectively measures the model's program repair capabilities. 4o and L-405b demonstrate strong program repair abilities with both high intervention and success rates. For example, at "very easy" difficulty, 4o shows 4.67 interventions with 0.83 success rate, L-405b shows 3.67 interventions with 0.85 success rate. As such, we can observe that when these models encounter failures, they can effectively analyze their own code, understand test failures, and implement successful fixes. This program repair capability persists even at higher difficulty levels, though with decreasing effectiveness. On the other hand, L-70b and L-8b have a lower number of interventions but significantly lower success rates as well. Furthermore, their success rates remain low despite interventions. For example, L-8b shows minimal interventions across "very easy" and "easy" but maintains very low success rates (0.19 for "very easy", dropping to 0.00 after "easy"). This indicates that even when given full context, including the original solution, test cases, and error outputs, these models struggle to identify and fix problems in their generated code.

GPT-OSS and L4S stand out with the highest intervention rates compared to the other models, with GPT-OSS's solutions requiring the highest intervention rates compared to all the other models, regardless of the concept or difficulty level of the underlying challenge. This indicates that GPT-OSS's first attempt at a solution often fails at first run. However, this also indicates that GPT-OSS has the highest program repair capability: given feedback about the failure, the original solution, and test cases, it reliably identifies root causes and produces a correct solution. Furthermore, when coupled with the node distribution across difficulties and concepts as shown in Figure 11a, we can observe that GPT-OSS quickly progresses beyond "very easy" to "medium" level challenges and *PrismBench* focuses on challenges with higher levels of difficulty for determining its capability. In contrast, L4S displays inconsistent performance across the search tree. When aggregated by difficulty level, it displays high success rates and moderate to low number of intervention calls, however, as shown in Figure 11b, it visits far fewer nodes at "hard" and "very hard" difficulty levels compared to the other top models. We can observe that as *PrismBench* progresses deeper through the tree, once it reaches "hard" difficulty levels, it repeatedly falls back to "medium" and "easy" level challenges as indicated by the node difficulty distributions across depths in Figure 11b. However, despite its inability to sustain problem-solving at higher difficulties, from table 6, we can observe that, similar to GPT-OSS, it has a high program repair capability.

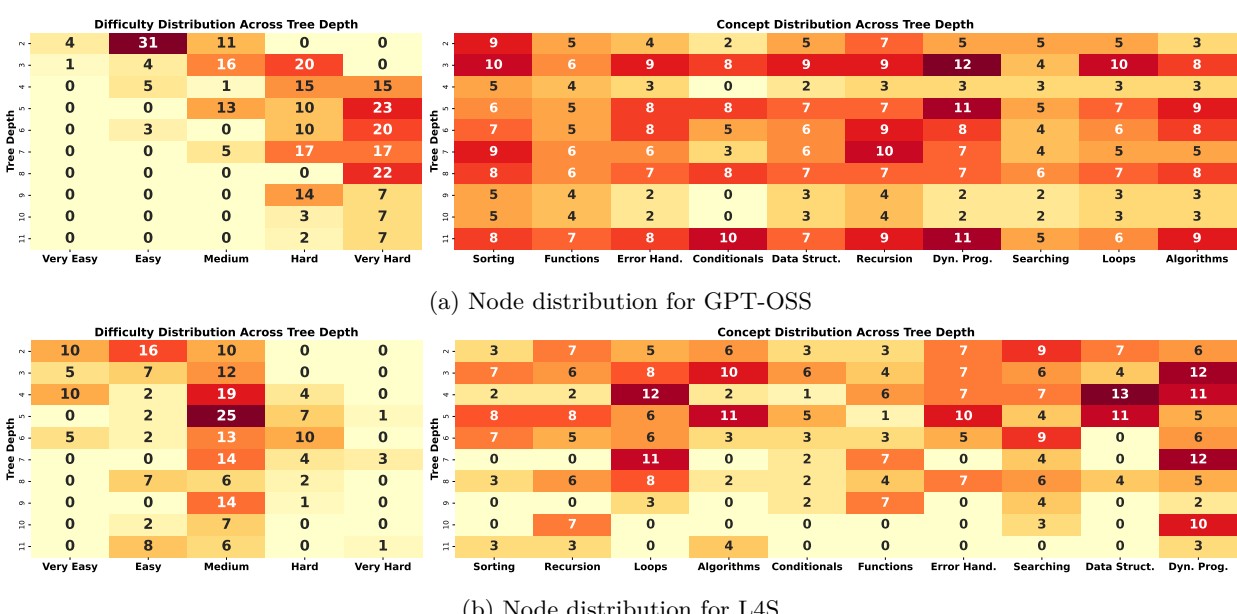

(a) Node distribution for GPT-OSS

(b) Node distribution for L4S

Figure 11: Node distributions for GPT-OSS and L4S averaged over 3 independent runs. The numbers in each cell indicate the number of nodes.

Figure 12 shows the error pattern analysis per concept for 4o and L-405b. The numbers in each cell represent the average occurrence of each error type per concept across 3 independent runs. 4o shows significantly higher errors in algorithm implementations, particularly in challenges related to "dynamic programming" where algorithm implementation errors peak at 35.8 occurrences and errors related to case sensitivity peak at 28.8. Index error rates in generated code for challenges involving "conditional" and "data structure" concepts (29.6 and 26.7 occurrences respectively) further demonstrate 4o's specific struggle with complex pointer and array manipulations. L-405b's errors, on the other hand, are mainly in "function" implementation challenges (19.5 occurrences for case sensitivity errors, 15.3 for index errors). We can observe that L-405b maintains consistent performance across most tested concepts, with notably lower error rates in recursive implementation challenges (consistently below 4.0 occurrences) compared to 4o. However, similar to 4o, L-405b also struggles with "dynamic programming", "sorting", and "data structure" challenges.

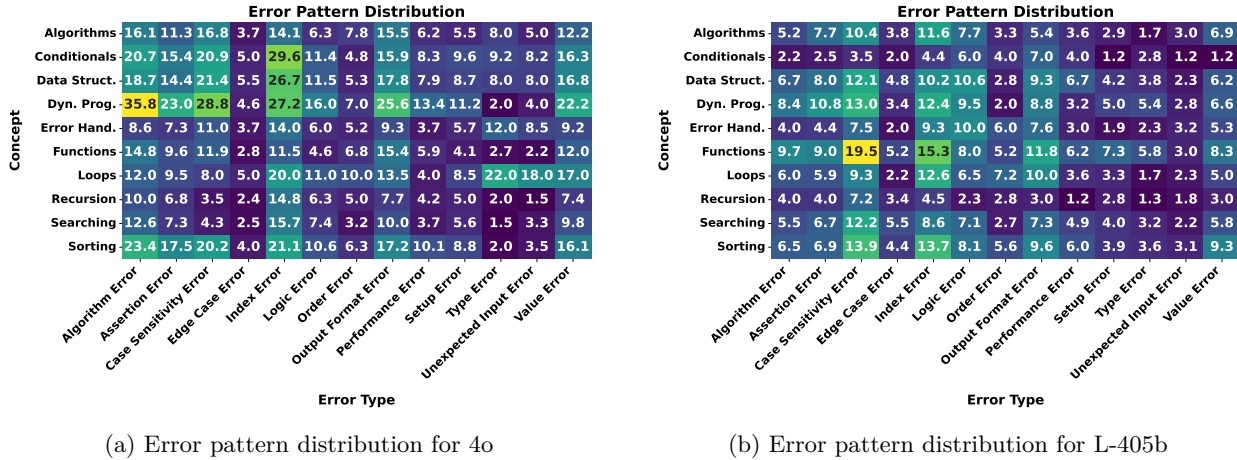

(a) Error pattern distribution for 4o

(b) Error pattern distribution for L-405b

Figure 12: Error pattern distributions for 4o and L-405b averaged over 3 runs. The numbers in each cell indicate the number of times each error was raised.

The most encountered error types (algorithm implementation, case sensitivity, and index errors) are consistently related to implementation details rather than fundamental algorithmic understanding. This observation is reinforced by the notably lower frequency of type, setup, and corner case errors across both models and all tested programming concepts. These patterns suggest that while both models demonstrate sound algorithmic understanding, their primary struggle lies in generating the correct code for solutions and tests. Analyzing test validation issues allows us to pinpoint whether the errors stem from incorrectly generated solutions or incorrectly generated tests by the models. Figure 13 presents the distribution of test validation issues for both 4o and L-405b across concept combinations. Each cell indicates how often a specific validation problem for a test, such as incorrect condition coverage or incorrect boundary checks, was identified. By comparing these data with the error pattern distributions in Figure 12a and 12b, we can discover correlations between the root cause of encountered errors and the concept areas where those errors were raised most frequently.

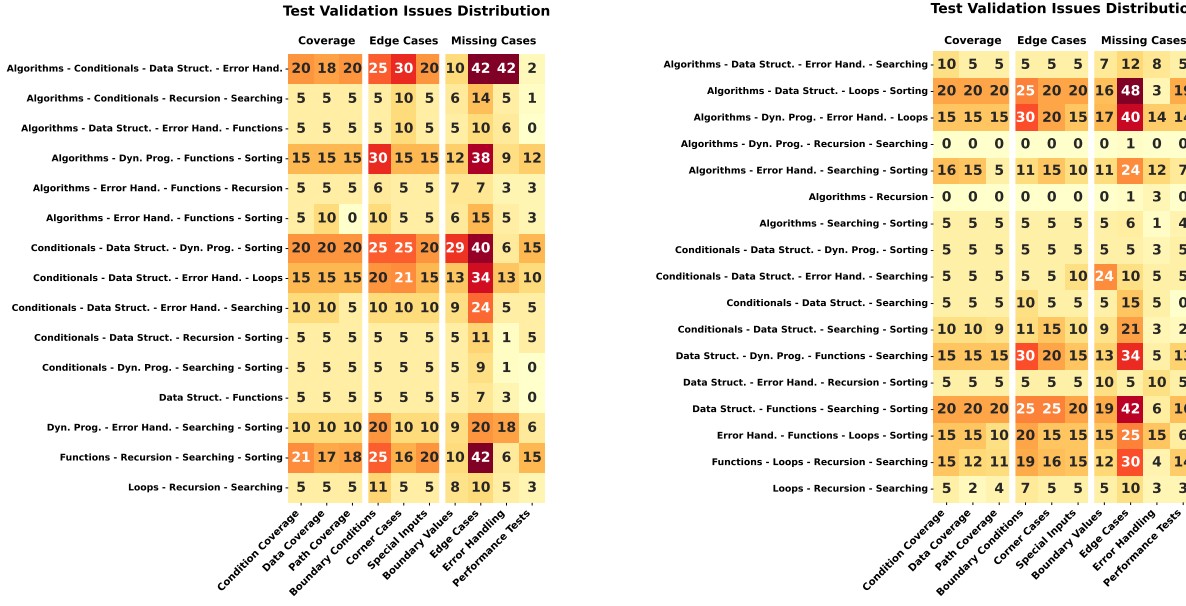

(a) Test validation issues distribution for 4o        (b) Test validation issues distribution for L-405b

Figure 13: Test validation issues distribution for 4o and L-405b averaged over 3 runs. The numbers in each cell indicate the number of times each issue was identified.

As mentioned in Section 5.1, both GPT-OSS and DS3, frequently rely on Python's built-in functionalities (e.g., using Python's in-place `sort` function on arrays) alongside standard and third-party libraries in their solutions (e.g., using `itertools` or `numpy` for array processing), particularly in challenges with high difficulty levels or multiple concept combinations. As our aim in *PrismBench* is to evaluate the models' coding capability rather than library knowledge, the *interactive sandbox* provides only a base Python environment without additional packages. Consequently, any dependence on third-party libraries leads to failure regardless of solution logic. Figure 14, summarizes the error pattern distributions for GPT-OSS and DS3. Similar to 4o and L-405b failures on the most challenging concepts for the model are mostly caused by algorithm and logic errors; however, both GPT-OSS and DS3 show a high concentration of errors clustered around "setup errors" and "unexpected input errors". These error types are raised when executions fail due to missing imports, incompatible program structure, or library usage outside what is available inside the environment. Moreover, by comparing Figures 12 and 14, we can observe that GPT-OSS has an overall higher level of error counts compared to 4o, L-405b, and DS3, which is consistent with its higher *Problem Fixer* intervention ratios as shown in Tables 5 and 6.

For 4o, the combinations of concepts that have the highest number of test validation failures are [algorithms, conditionals, data structures, error handling] and [functions, recursion, searching, sorting]. These concepts

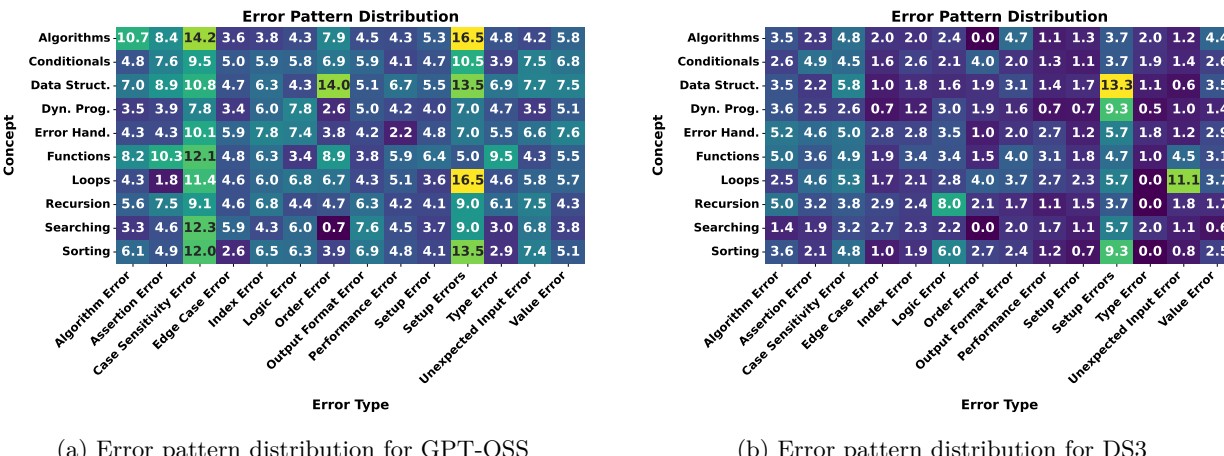

(a) Error pattern distribution for GPT-OSS        (b) Error pattern distribution for DS3

Figure 14: Error pattern distributions for GPT-OSS and DS3 averaged over 3 runs. The numbers in each cell indicate the number of times each error was raised.

Figure 15: Details on the concept combination effects on 4o's performance. The right matrix displays the average success rates for all nodes related to each specific combination. The left matrix displays the average number of times each concept combination was visited in the search tree, regardless of success/failure.

also have the highest number of errors, as shown in Figure 12a, particularly with index and case sensitivity errors. Furthermore, as shown in Figure 10a, *PrismBench* has specifically focused on these concepts by generating a high number of nodes for thorough validation and isolation of issues. This indicates that many of 4o's generated tests have the same underlying root cause of its generated solutions (for instance, mishandling pointer or array indices). Moreover, the high frequency of numeric and string value assertions that fail in these tests suggests that 4o often struggles to produce fully consistent test inputs or expected outputs, leading to assertion failures even when the generated solution is correct. We can observe a correlation between test validation issues and error types, demonstrating that these failures are not only in the generated solutions but also in the generated tests. The highest validation issues appear in concepts requiring numeric and string value assertions. This suggests that 4o struggles with processing such concepts during test generation. Therefore, even when 4o produces correct solutions, its limitations in numerical and string processing lead to incorrect test assertions, resulting in failures and error cascades. These issues are also reflected in the success rates and visit counts of concepts as shown in Figure 15, where we observe lower success rates and higher visit counts for the combinations of concepts with high error rates and test validation issues.

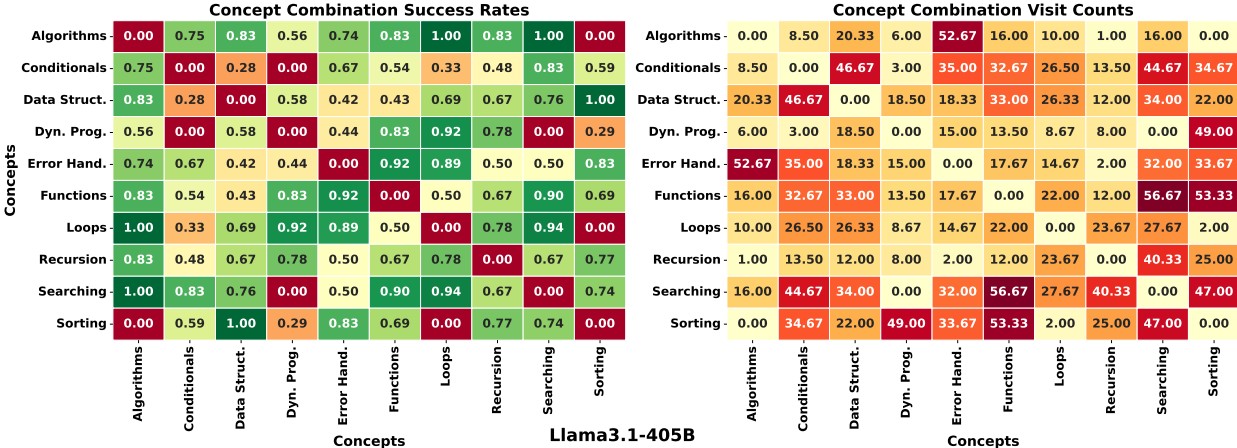

Figure 16: Details on the concept combination effects on L-405b's performance. The right matrix displays the average success rates for all nodes related to each specific combination. The left matrix displays the average number of times each concept combination was visited in the search tree, regardless of success/failure.

L-405b on the other hand, consistently struggles with "function" and "sorting" concepts, especially when "data structures" or "searching" are also included as shown in Figure 16. The test validation issues in Figure 13b demonstrate that combinations like [data structure, function, searching, sorting] exhibit high incorrect coverage issues and a large number of missing or incomplete test cases. Similarly, the error pattern distribution for L-405b in Figure 12b shows peaks in case sensitivity and index errors whenever function implementations are tested. We can observe that L-405b exhibits different root causes for failures compared to 4o, particularly in concepts combination of "data structures" and "error handling".

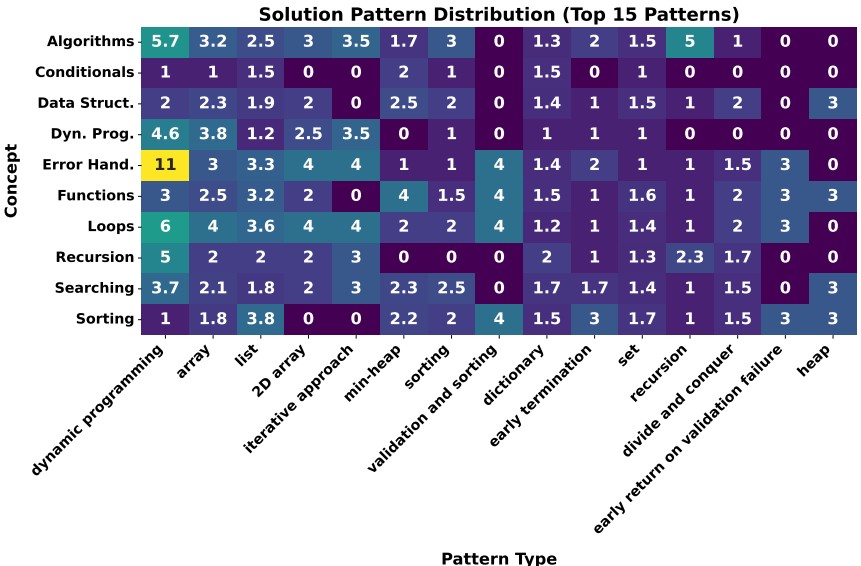

Figure 17: Patterns identified in solutions distribution for L-405b averaged over 3 runs. The numbers in each cell indicate the number of times each pattern was identified.

By correlating test validation issues and the solution patterns shown in Figure 17, we can see that L-405b's failures primarily stem from syntax errors and hallucinations rather than logical errors as evidenced by the high number of failures in using built-in data types (arrays, list, dictionary, etc.). The lowest-performing nodes and their corresponding patterns show that L-405b frequently generates non-existent syntax (e.g.,

non-existent built-in function calls, incorrect syntax for using built-in data types, etc.), creating a situation where both the generated code and its corresponding tests are incorrect. This leads to the high intervention rates observed in Table 6, as the *Problem Fixer* repeatedly intervenes to correct both solution and test issues.

**Concept Combination Success Rates**

| Concepts | Algorithms | Conditionals | Data Struct. | Dyn. Prog. | Error Hand. | Functions | Loops | Recursion | Searching | Sorting |
|---|---|---|---|---|---|---|---|---|---|---|
| Algorithms | 0.00 | 0.56 | 0.64 | 0.78 | 0.59 | 0.67 | 0.90 | 0.89 | 0.87 | 0.93 |
| Conditionals | 0.56 | 0.00 | 0.65 | 0.43 | 0.72 | 0.87 | 0.88 | 0.88 | 0.71 | 0.67 |
| Data Struct. | 0.64 | 0.65 | 0.00 | 0.87 | 0.66 | 0.89 | 0.78 | 0.67 | 0.89 | 0.82 |
| Dyn. Prog. | 0.78 | 0.43 | 0.87 | 0.00 | 0.50 | 0.92 | 1.00 | 0.74 | 0.60 | 0.20 |
| Error Hand. | 0.59 | 0.72 | 0.66 | 0.50 | 0.00 | 0.83 | 0.81 | 0.86 | 1.00 | 0.97 |
| Functions | 0.67 | 0.87 | 0.89 | 0.92 | 0.83 | 0.00 | 0.60 | 0.89 | 1.00 | 0.74 |
| Loops | 0.90 | 0.88 | 0.78 | 1.00 | 0.81 | 0.60 | 0.00 | 0.78 | 1.00 | 0.92 |
| Recursion | 0.89 | 0.88 | 0.67 | 0.74 | 0.86 | 0.89 | 0.78 | 0.00 | 1.00 | 0.96 |
| Searching | 0.87 | 0.71 | 0.89 | 0.60 | 1.00 | 1.00 | 1.00 | 1.00 | 0.00 | 0.89 |
| Sorting | 0.93 | 0.67 | 0.82 | 0.20 | 0.97 | 0.74 | 0.92 | 0.96 | 0.89 | 0.00 |

**GPT-OSS-20B**

**Concept Combination Visit Counts**

| Concepts | Algorithms | Conditionals | Data Struct. | Dyn. Prog. | Error Hand. | Functions | Loops | Recursion | Searching | Sorting |
|---|---|---|---|---|---|---|---|---|---|---|
| Algorithms | 0.00 | 55.33 | 21.67 | 25.33 | 31.67 | 32.67 | 34.67 | 49.00 | 20.33 | 38.67 |
| Conditionals | 55.33 | 0.00 | 59.67 | 26.67 | 50.00 | 12.33 | 38.00 | 60.67 | 14.00 | 24.67 |
| Data Struct. | 21.67 | 59.67 | 0.00 | 20.67 | 9.50 | 31.33 | 8.00 | 58.00 | 7.00 | 44.50 |
| Dyn. Prog. | 25.33 | 26.67 | 20.67 | 0.00 | 60.00 | 9.50 | 30.00 | 43.67 | 116.50 | 35.33 |
| Error Hand. | 31.67 | 50.00 | 9.50 | 60.00 | 0.00 | 3.50 | 11.33 | 45.00 | 38.00 | 20.67 |
| Functions | 32.67 | 12.33 | 31.33 | 9.50 | 3.50 | 0.00 | 119.33 | 24.00 | 24.00 | 104.67 |
| Loops | 34.67 | 38.00 | 8.00 | 30.00 | 11.33 | 119.33 | 0.00 | 18.33 | 16.00 | 76.00 |
| Recursion | 49.00 | 60.67 | 58.00 | 43.67 | 45.00 | 24.00 | 18.33 | 0.00 | 4.67 | 76.33 |
| Searching | 20.33 | 14.00 | 7.00 | 116.50 | 38.00 | 24.00 | 16.00 | 4.67 | 0.00 | 24.00 |
| Sorting | 38.67 | 24.67 | 44.50 | 35.33 | 20.67 | 104.67 | 76.00 | 76.33 | 24.00 | 0.00 |

Figure 18: Details on the concept combination effects on GPT-OSS's performance. The right matrix displays the average success rates for all nodes related to each specific combination. The left matrix displays the average number of times each concept combination was visited in the search tree, regardless of success/failure.

As shown in Figure 18, GPT-OSS's performance remains consistently high across most concept combinations, maintaining high success rates even as challenges become more complex. However, it demonstrates persistent weaknesses in challenges involving "dynamic programming" and "error handling" relative to other concepts. Interestingly, we can observe that nodes which combine "sorting" with "functions", "loops", or "recursion" are frequently revisited across the search tree. This pattern is also present in GPT-OSS's error distributions as displayed in Figure 14a, where nodes with these concepts have a high rate of "setup" and "unexpected input" errors. This is a direct result of GPT-OSS's over-reliance on third-party libraries, which are not available in the environment, and its tendency to define nested functions. However, from Figure 18, we can observe that these nodes have a high overall success rate. Investigating the solutions for these nodes reveals that GPT-OSS's first attempts at solving challenges requiring array manipulation rely on third-party libraries instead of built-in standard libraries and functionalities. It is only during the subsequent attempts, where the feedback contains information about these libraries not being available, that it generates a correct solution. As such, GPT-OSS exhibits a different coding profile compared to the much larger 4o and L-405b, which use Python's standard libraries.

It is important to note that these insights come from *PrismBench*'s automated analysis. The search trees generated by the framework enable deeper investigation of behavioral patterns and contain detailed analysis for each node. We only highlight the most significant behavioral patterns observed.

## 5.3 Effects of Scale

Given the evaluation results on the GPT-4o and Llama3.1 families, in this section, we investigate the effects of model scale on code generation and problem-solving capabilities. Importantly, we do not include GPT-OSS and L4S in our analysis in this section, as both models have different architectures compared to the others.

### 5.3.1 GPT-4o

GPT-4o and GPT-4o Mini, developed by OpenAI, are part of the same model family but differ in scale, performance, and application focus Hurst et al. (2024). GPT-4o is the high-performance, multimodal flagship model optimized for complex tasks requiring deep reasoning and nuanced language understanding, while GPT-4o Mini is a lightweight, cost-efficient variant designed for speed and accessibility, prioritizing rapid

token generation and affordability. While both models share core architectural features like Transformer-based design and multimodal capabilities, GPT-4o Mini is reported to be significantly smaller than GPT-4o.

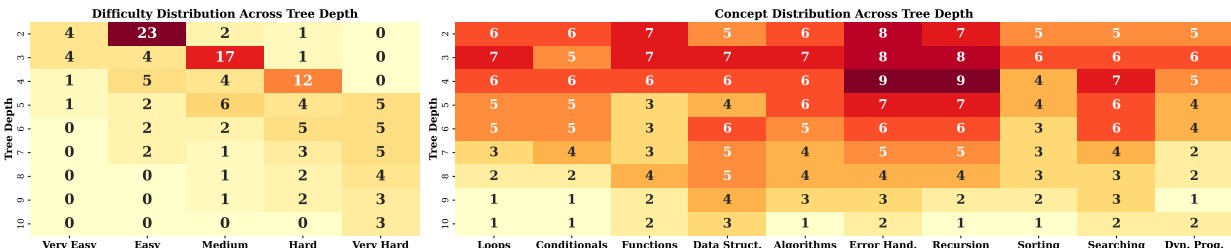

Figure 19: Node distribution for 4o-M averaged over 3 runs. The numbers in each cell indicate the number of nodes.

Figure 19 displays the node distribution and visit counts of 4o-M throughout the search tree. In the previous sections, we presented performance results for 4o, with its corresponding node distributions presented in Figure 10a. Comparing the distributions of 4o with 4o-M shows us how scale impacts performance. While the majority of 4o's nodes are distributed in challenges with "hard/very hard" difficulty and deeper parts of the tree (as shown in Figure 10a), we can observe that for 4o-M, the majority of nodes are distributed between challenges with "medium" and "hard" difficulty and in shallower depths. This is also evident in the search tree for 4o-M as shown in Figure 26.

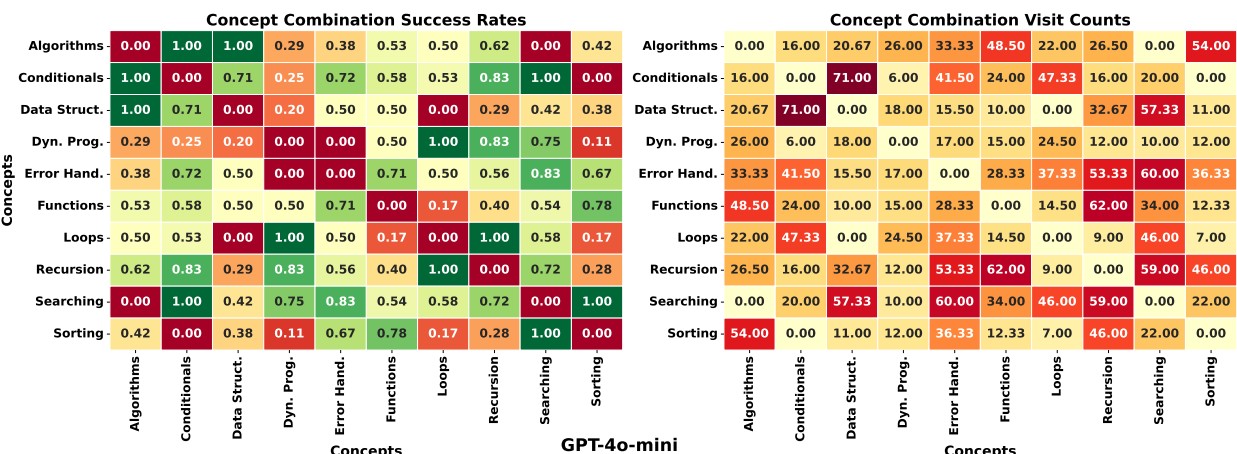

Figure 20: Details on the concept combination effects on 4o-M's performance. The right matrix displays the average success rates for all nodes related to each specific combination. The left matrix displays the average number of times each concept combination was visited in the search tree, regardless of success/failure.

Figure 20 shows the success rates and visit ratios for nodes corresponding to different concept combinations. As discussed in Section 5.1, we can see that the majority of 4o-M's failures occur when it encounters combinations of concepts that require compositional reasoning. For instance, we can observe that 4o-M has relatively low success rates for "dynamic programming", which fall even lower when the challenge combines another concept with "dynamic programming".

### 5.3.2 Llama 3

The Llama 3 herd of models, developed by Meta, is a family of LLMs designed to support multimodality, coding, reasoning, and tool use. The term "herd of models" refers to the diverse range of models within the Llama 3 family, each tailored for specific applications Dubey et al. (2024). The flagship model, L-405b, is a dense Transformer architecture with 405 billion parameters and a context window of up to 128,000 tokens,

enabling it to handle extensive datasets and complex tasks. While these models share foundational training data and post-training processes, they differ in architectural scale—such as the number of layers, model dimensions, attention heads, and FFN dimensions—to optimize performance across varying use cases. This allows us to leverage *PrismBench* to systematically evaluate how architectural and parametric scale impact code-generation capabilities. While we have already presented performance results for L-405b (the most capable variant) in prior sections, this section focuses on analyzing performance differences across scaled-down versions of the Llama 3 family.

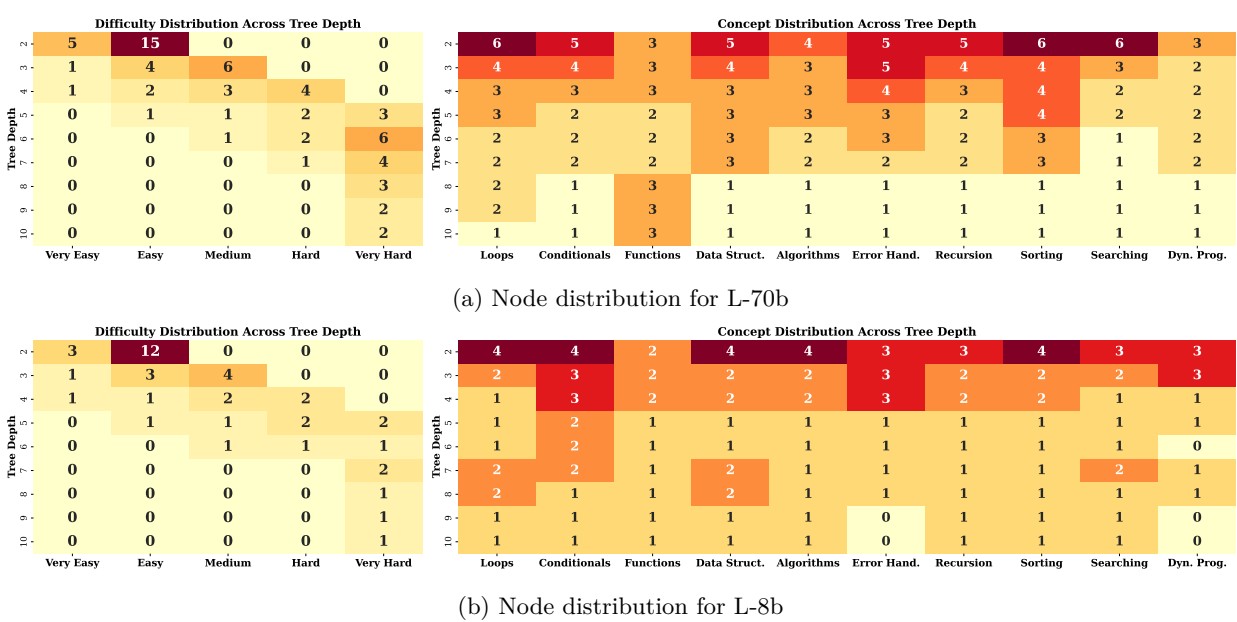

Figure 21: Node distributions for L-70b and L-8b averaged over 3 runs. The numbers in each cell indicate the number of nodes.

As shown in Figure 21a and 21b, the depth of explored nodes drops sharply for smaller models, indicating limitations in handling more difficult problems. In particular, the node distribution for L-8b shows that only the shallow parts of the tree (depths 2 and 3) and "easy" challenges have been explored in depth. The inability to explore deeper nodes suggests that the model fails to generate correct solutions when challenges are more difficult or contain multiple programming concepts. Conversely, L-70b reaches deeper parts of the tree more often and is capable of reaching nodes with "very hard" difficulty to an extent (even though it fails at all of them) as shown in Table 5. The node distributions of the search trees generated for L-405b, L-70b, and L-8b) clearly demonstrate how models' scale plays an important role in their problem-solving and code-generation capabilities. The search trees themselves for these two models (L-70b and L-8b), as presented in Figure 30, further show how these models struggle to complete challenges as they become more difficult. Figure 30b shows how the majority of nodes for L-8b are generated in Phase 3 (highlighted in blue). As explained in Section 3.4, Phase 3 is responsible for comprehensively inspecting areas of failure, and as such, we can see that L-8b consistently fails with high failure rates, with the majority of nodes being generated in Phase 3. On the other hand, L-70b shows a slightly better performance as pictured in Figure 30a. In the same manner as L-8b, the majority of L-70b's nodes are generated during Phase 3, However, we can see that, unlike L-8b, many nodes were also generated in Phase 2, indicating that the model was capable of solving some of these challenges albeit with low success rates.

Figure 22 and 23 further demonstrate performance degradation as the models get smaller. The success rates of concept combinations decrease significantly for L-8b, particularly for tasks requiring more advanced strategies (e.g., "dynamic programming" or multiple nested constructs). In comparison, L-70b shows moderate success with simpler "loops" and "conditionals" but similarly struggles to sustain the performance under combined, higher-level concepts.

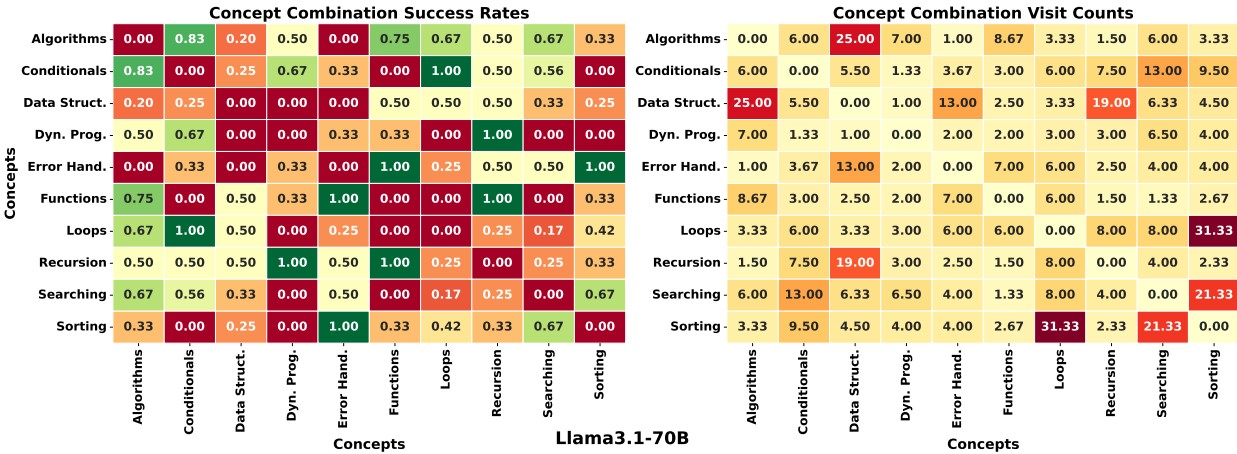

Figure 22: Details on the concept combination effects on L-70b's performance. The right matrix displays the average success rates for all nodes related to each specific combination. The left matrix displays the average number of times each concept combination was visited in the search tree, regardless of success/failure.

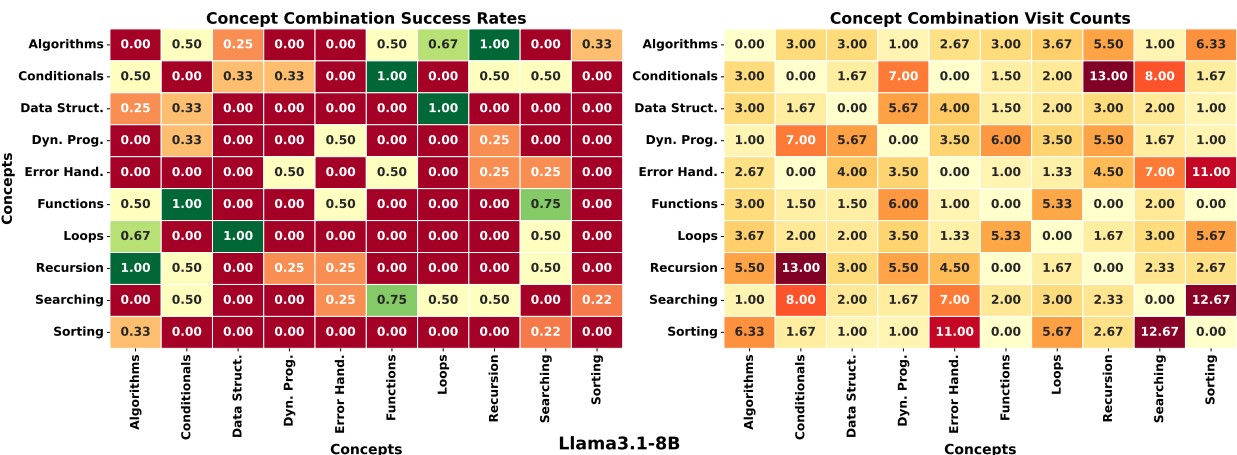

Figure 23: Details on the concept combination effects on L-8b's performance. The right matrix displays the average success rates for all nodes related to each specific combination. The left matrix displays the average number of times each concept combination was visited in the search tree, regardless of success/failure.

## 6 Related Works

Recent studies have demonstrated how current benchmarking approaches fall short in evaluating current models. Specifically, Xu et al. (2024b), Roberts et al. (2023), and Jiang et al. (2024) show how even extensive attempts to control the training dataset does not prevent data contamination, either because filtering out undesired data is difficult given the training dataset's size Balloccu et al. (2024) or because of models' increasing generalization capabilities and lack of diversity in benchmarks' challenges Dong et al. (2024). Crowd-sourced benchmarks such as Chiang et al. (2024); Fourrier et al. (2024) have proven to be much more effective in evaluating LLMs but suffer from 1) a limited coverage of real-world challenges Lin et al. (2024), and 2) user biases in preference ranking of the responses Chiang et al. (2024). As such, benchmarks such as ARC-AGI Chollet (2019), SWE-bench Jimenez et al. (2024), HELM Liang et al. (2023), and HLE Phan et al. (2025) have been proposed, which include either very difficult problems or data that the models could not have been trained on given the time of their release White et al. (2024); Franzmeyer et al. (2024). The majority of leading LLM providers are currently using these benchmarks. However, with the rapid increase in LLM capabilities, these benchmarks are quickly becoming obsolete, as models achieve increasingly higher

scores, rendering the evaluation metrics ineffective in distinguishing their performance Anthropic (2024); Jaech et al. (2024); Liu et al. (2024).

Dynamic benchmarking approaches have been proposed to address the limitations mentioned above. LLM-as-Judge frameworks Zhu et al. (2023b); Li et al. (2023); Wang et al. (2023a) use one or multiple LLMs (based on benchmark performance or fine-tuned for specific evaluation criteria) to assess and rank the responses of the model under evaluation. However, given the judge models' capabilities and biases, offloading performance assessment to an LLM introduces challenges in the reproducibility and reliability of the evaluation Thakur et al. (2024). On the other hand, approaches have been proposed for leveraging LLMs to generate novel and out-of-distribution evaluation scenarios based on existing static benchmarks Li et al. (2024b); Zhang et al. (2024a); Zhuge et al. (2024). Building on these approaches, frameworks such as Li et al. (2024c); Zhu et al. (2024a); Fan et al. (2023); Wang et al. (2024); Zhang et al. (2024c); Zhu et al. (2024b) extend dynamic evaluation to provide a more systematic assessment of a model's capabilities. Notably, DyVal Zhu et al. (2024a) models each evaluation scenario as a DAG-based composition of reasoning tasks, guided by a graph generation algorithm, defined constraints, and translation into natural language. DyVal2 Zhu et al. (2024b) replaces the DAG with LLM-based agents to transform existing benchmarks into new challenges to evaluate LLMs' performance and generalization on multiple domains. DARG Zhang et al. (2024c) and TreeEval Li et al. (2024c) represent the evaluation processes as graphs or trees, analyzing how the model navigates these structures. However, the stochasticity of LLMs and the variability in their performance introduce challenges in ensuring consistency and reliability in these evaluation methodologies Blackwell et al. (2024).

# 7 Threats To Validity

**Threats to internal validity** concern factors internal to our work that could have influenced our study. The inherent stochasticity of LLMs which results in variability of results across runs is the most important threat to our work's internal validity. We aimed to address this threat at each step of the process through using TD(0) scoring for nodes, $\epsilon$-greedy policies for node selection, and node value convergence checks at each evaluation phase as detailed in Sections 3.2 and 3.4. Additionally, while our multi-phase search strategy is designed to focus on promising areas of the search space, the stochasticity in LLM's performance may introduce state selection bias by underexploring edge cases. We mitigate this threat by conducting multiple trials for each model and reporting averaged results across trials to account for variability and reduce the impact of outlier behaviors. Finally, dynamic challenge generation using an LLM can result in miscalibrated evaluation results in comparison to human preferences. As detailed in Section 3.5, we mitigate this threat by using the same agent as the *Challenge Designer* (4o-M) for all models under study (except for evaluating 4o-M itself to minimize bias) and allowing for using verified, labeled challenge banks in order to evaluate models.

**Threats to construct validity** concern the relationship between theory and observation. In the context of our work, the main construct threat is whether our proposed evaluation metrics (as we detail in Section 3.6) holistically capture the multiple aspects of LLMs' code generation capabilities. We mitigate this threat by executing generated code within an isolated sandbox environment, ensuring that our evaluation is grounded in the actual result of the models' outputs. Furthermore, as detailed in Section 3.3, we isolate the different stages of code generation, namely solution generation, test generation, and program repair, by using separate agents for each step. The other threat to our work's construct validity is that Phase 3's pattern analyses rely on a judge model (4o in this study) and while the judge model can be changed to any available model, there still exists the risk of the judge's capability ceiling and potential bias resulting in incorrect analysis results. We address this threat by restricting the judge model's role to post hoc analysis so that it does not influence the overall search and benchmarking process.

**Threats to external validity** concern the generalizability of our findings beyond the specific setup used in this study. Our evaluations were carried out on eight LLMs and a finite set LeetCode style challenges on pre-defined concepts and difficulty levels, which, although diverse, may not capture the full range of coding tasks encountered in practice. We recognize that given experiment design, the results of our experiments may not reflect real-world coding capabilities. Additionally, as the benchmarking process is computationally and financially expensive, each node's evaluation requires multiple LLM calls via API or local models (as

detailed in Sections 3.3 and 4.5), we limited the number of tasks and the depth of tree expansion at each phase (Section 4.4). These constraints may impact the extent to which our results generalize to larger-scale or different task distributions.

## 8 Conclusion

In this paper, we introduced *PrismBench*, a dynamic benchmarking framework that models the evaluation space as a search tree and uses MCTS to systematically explore evaluation scenarios. Unlike prior approaches, *PrismBench* dynamically analyzes how LLMs approach problems, adapt to feedback, and handle increasing complexity using a structured multi-phase pipeline and specialized agents to uncover systematic model weaknesses. Our key contributions are as follows:

- We formalized the space of code generation tasks as a search tree over programming concepts and difficulty levels, allowing for adaptive and targeted evaluation of LLM capabilities.

- We introduce a multi-phase evaluation strategy: (1) Capability Mapping to assess baseline strengths, (2) Challenge Discovery to identify systematic weaknesses, and (3) Comprehensive Analysis to investigate failure patterns and root causes.

- We proposed a set of metrics in order to provide fine-grained insights into model behavior beyond standard pass/fail metrics.

- We conduct experiments across eight state-of-the-art LLMs and show that while larger models demonstrate stronger general capabilities, they still struggle with compositional reasoning, high-difficulty tasks, and consistent test generation.

Additionally, we provide *PrismBench* as an open-source, extensible framework Majdinasab (2025) for conducting such evaluations to allow practitioners to conduct trustworthy, targeted benchmarking strategies as LLM capabilities continue to evolve.

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

# A Appendix

## A.1 Sample Trees

This section presents the minimized versions of the search trees generated for all models examined in this study. To ensure brevity, the trees presented here focus solely on the concepts, difficulty levels, and scores of

each node, along with the phase during which they were generated. The original trees, however, are much more detailed, but they would not fit in the content of this paper. The original trees contain the complete challenge description, the generated solutions, tests, attempts, and the corresponding analysis done at each node. This information allows for a comprehensive and fine-grained evaluation of model behavior at each node in the search tree. We have included the full original trees in our replication package Majdinasab (2025). The nodes for each phase are color-coded for distinction, with yellow nodes representing those generated in Phase 1, green nodes representing those generated in Phase 2, and blue nodes representing those generated in Phase 3. The edges indicate parent-child relationships, with red edges indicating a significant decrease in the child node's TD value compared to its parent, and green edges indicating otherwise. Each node is associated with specific attributes: the concepts related to the node are under "Concepts," the difficulty level under "Difficulty," and the number of times the node has been visited, under "Visits."

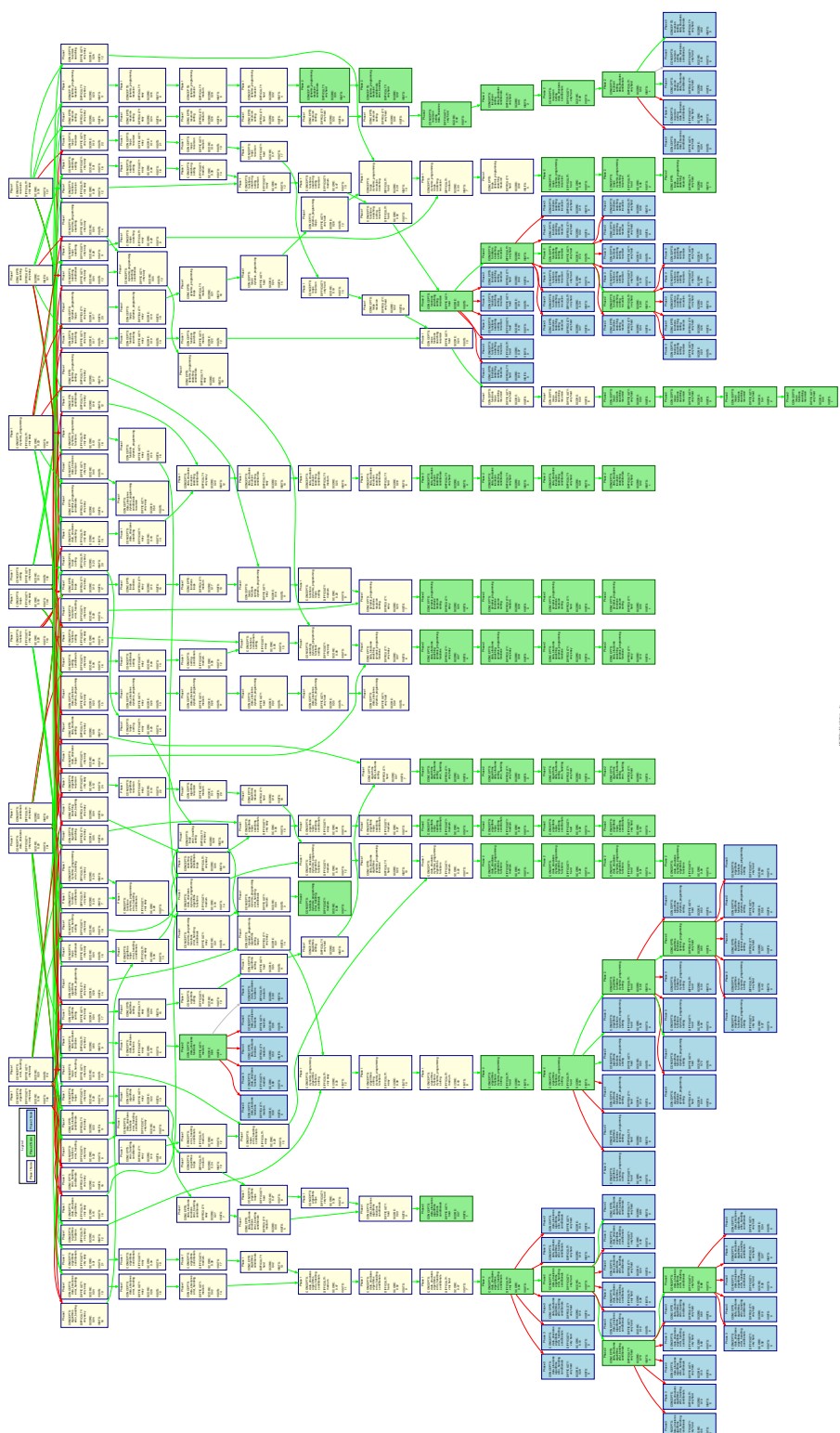

Figure 24: Search tree generated for 4o

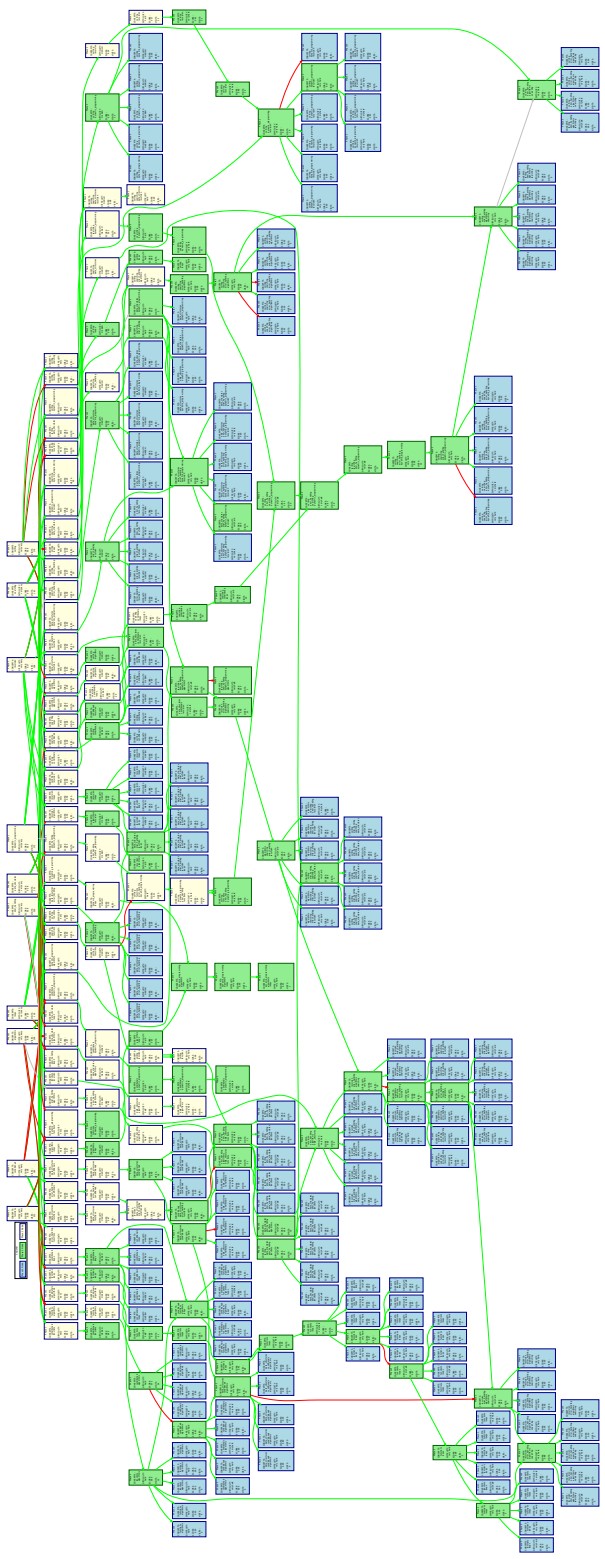

Figure 25: Search tree generated for GPT-OSS

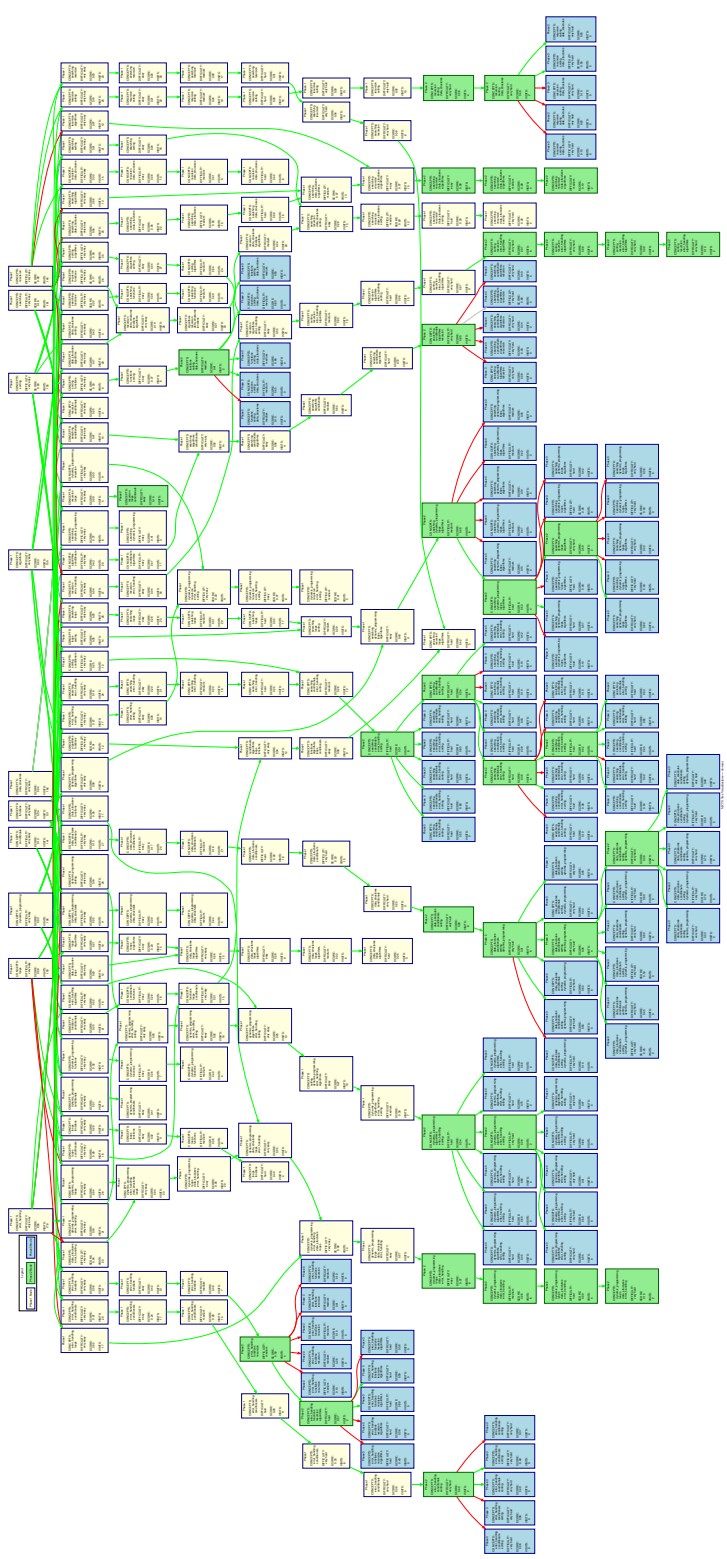

Figure 26: Search tree generated for 4o-M

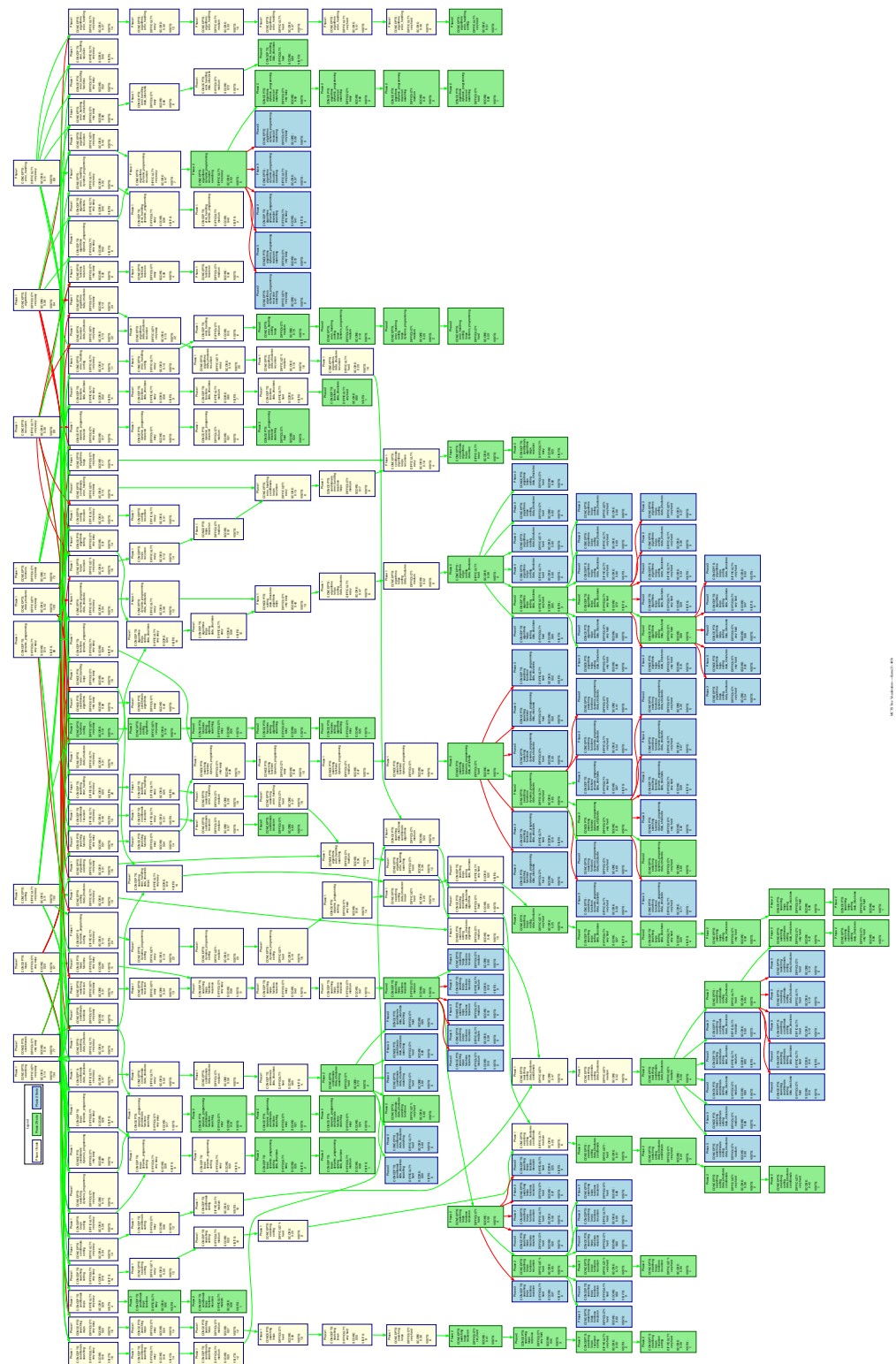

Figure 27: Search tree generated for L-405b

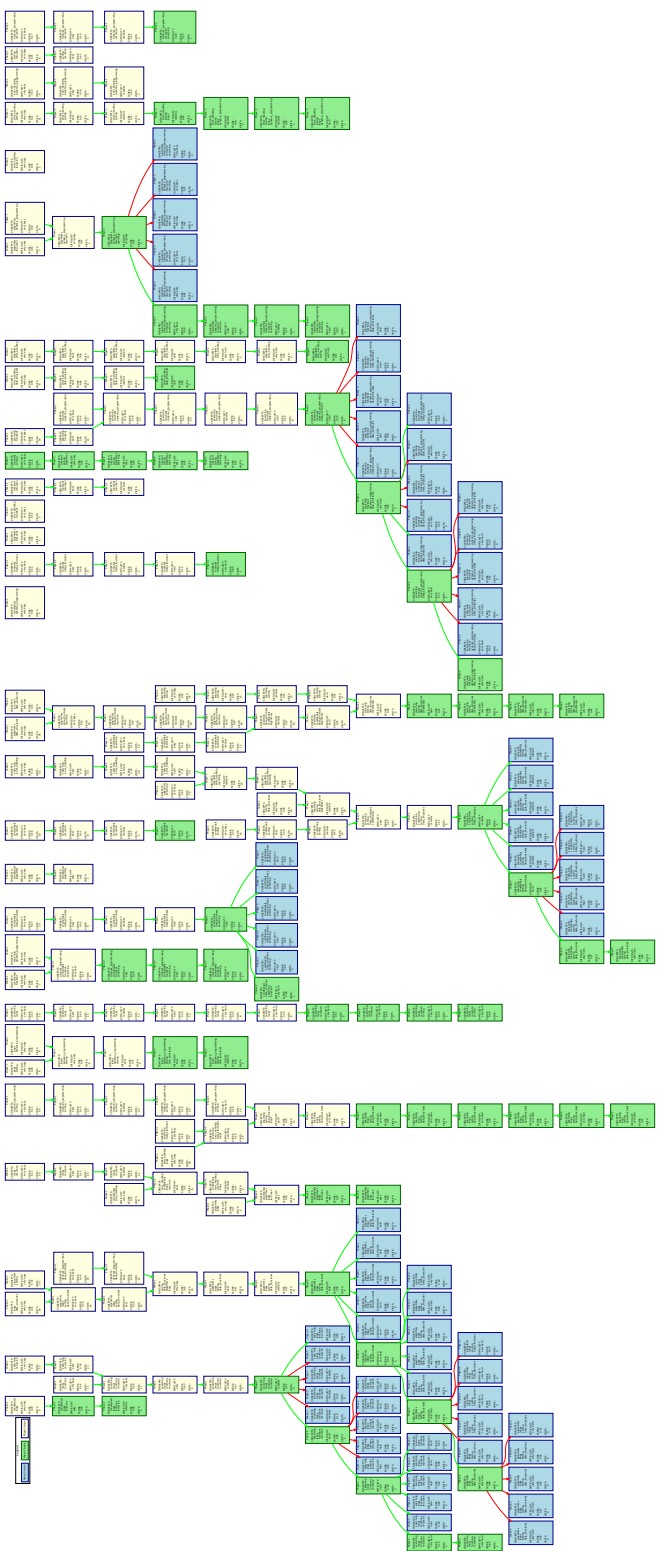

Figure 28: Search tree generated for DS3

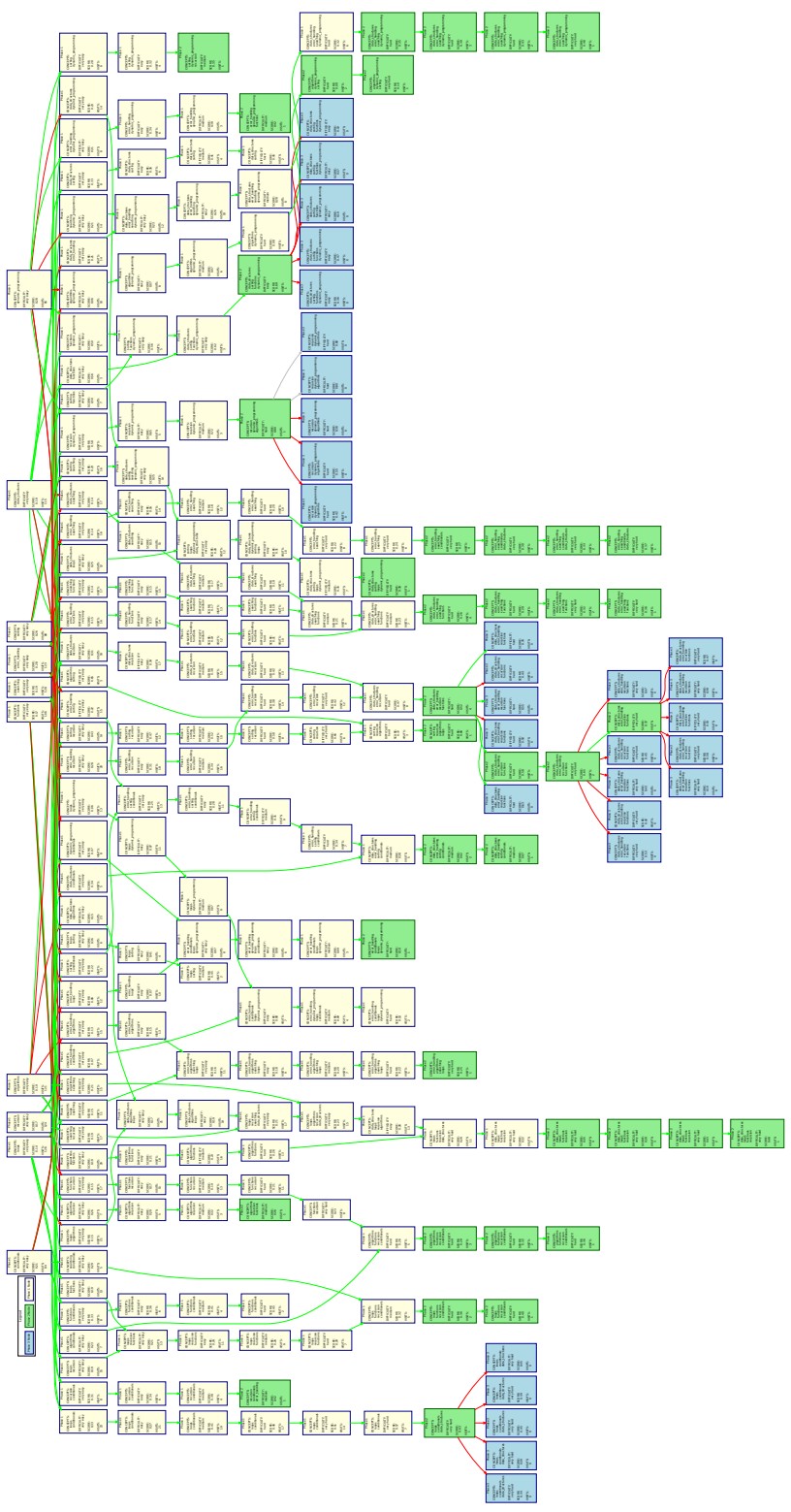

Figure 29: Search tree generated for L4S

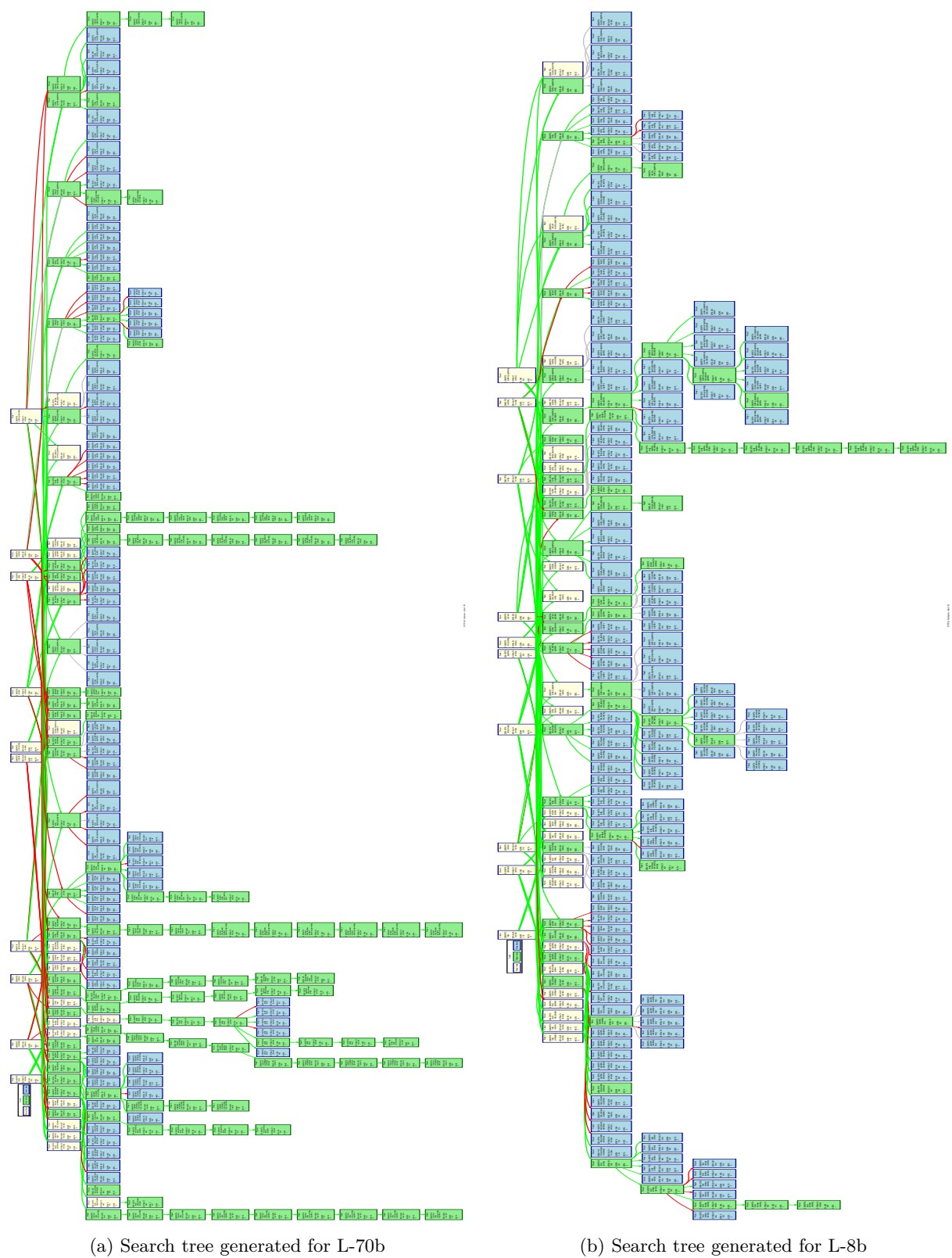

(a) Search tree generated for L-70b  (b) Search tree generated for L-8b

Figure 30: Search trees generated for L-70b (a) and L-8b (b)

