# OpenReview forum: "PrismBench: Dynamic and Flexible Benchmarking of LLMs Code Generation with Monte Carlo Tree Search"
_TMLR — Accepted by TMLR_

### Review · Reviewer_bTMW · 2025-12-30

**Summary Of Contributions:**

The paper aims to devise a model evaluation framework for assessing the capability of different models in solving different kinds of programming tasks, eliciting the capabilities and weaknesses of each model, since these may differ. The framework models the evaluation task as a (variant of) an MDP whose structure is designed to elicit the different capabilities of different models, and concentrate the effort in those regions of the programming challenge search space where the model may have a difficult time solving the challenge.

**Audience:**

Yes

**Audience Explanation:**

Coding capability evaluation is a hot topic of research.

**Claims And Evidence:**

No

**Claims Explanation:**

1. I am not convinced that the idea, in principle, does what it is supposed to do, since there are plenty of hyperparameters involved, and the results may differ between hyperparameter choices. The authors made no attempt to justify them, or show experimentally that their system is robust to this choice in any way.  This has several consequences:
    1.1 The issue of trust - there is no reason to believe that 0.9 is the "right" discount factor, or that $\alpha=0.9$ is the "right" learning rate. Same for penalties and thresholds. What if a different configuration yields a radically different conclusion about LLM capabilities?
    1.2 We cannot use it in a leaderboard for the same reason. One can always claim that their model is put at a disadvantage because of some choice of hyperparameters, and can possibly even prove it: they can use a tuner, like Optuna, to find the best config for their model, and claim that **this** is the right config. Again, for the same reason it also doesn't foster trust, becuase the system can be easy to "game"
2. The MDP is not really an MDP as claimed. The reward is claimed to be a function from $$S \times A \times S$$ to the real numbers, but every phase has its own reward, so I am assuming the phase is also a part of the state. There may be other missing parts I may have mised. The authors need to precisely formulate the model without hand-waving.
3. Section 3.3, bottom of page 8: making separate agents for test development and for problem solving does not reflect real software development - these two things go hand in hand together. There's even an entire methodology that promotes it - test-driven development. Moreover, using the same LLM for writing tests and for solving makes the system also easy to "game" and fosters less trust
4. It is extremely complex. Evaluation is most useful when it's simple and doesn't have a lot of "moving parts", because the interaction between them may be hard to reason about and again results in having a hard time trusting the results.

If I could trust the findings they would be very interesting. Especially the difference between different LLMs in different domains. But due to the above-mentioned comments, I am not convinced the system is trustworthy.

**Requested Changes:**

1. Either justify the hyperparameter choice, explain why you believe the system to be a reasonable evaluation framework for any "reasonable" choice (robustness), or actually prove robustness experimentally by running experiments and showing that the ranking between models across different dimensions remains similar (this one may be expensive, so I'm not sure how feasible it is)
2. Clarify exactly what you mean in MDP
3. Overall, do a better job at convincing why the system is not brittle and is not easy to game by a hyperparameter optimizer or using the fact that tests and problem solving are done using the same model.

---

> ### Author Response · Authors · 2026-01-22
>
> Dear reviewer bTMW, thank you for your thoughtful review and detailed comments. We sincerely appreciate your feedback as they have allowed us to improve our submission. In response to your concerns, we have revised the manuscript accordingly; all changes are numbered by response and highlighted in blue.
>
> Before addressing your concerns, we ground our explanations in RL, as PrismBench is explicitly modeled as an RL environment: the rules, dynamics, and reward structure are fixed and every agent starts from the same state and interacts under the same constraints. Agents are not compared by matching identical trajectories, but by how well they perform under the same setup.
>
> We follow the same principle and explicitly decouple the benchmarking environment from the benchmarking process:
> - MDP is the underlying **benchmarking environment**
> 	- We model the space of evaluation scenarios as an MDP where each state is a (concept, difficulty) pair and is connected to other states via shared concepts (_Sec. 3.1_). Transitions are only possible to reachable states. For example, from <loops, easy> we can only transition to states involving the same concept and cannot jump to <recursion, easy>.
> 	- All agents begin at the same root node and their trajectory is determined by value estimates derived from received rewards.
> - Traversal behavior is the **benchmarking process**
>   - Given the fixed environment, PrismBench measures how effectively a model explores and solves harder states. Each state’s value is computed using the same sandbox and reward function, and TD(0) propagates values backward to guide subsequent exploration.
>
> This makes PrismBench a fixed standard RL environment. With this framing, we address each of your concerns below:
>
> ## R 3.1 Regarding the MDP Definition
>
> We thank you for raising this issue. We agree that our presentation of “phase-specific rewards” could be read as inconsistent with a stationary reward function across the MDP. In response, we revised our MDP formalization in _Sec. 3.2.1_ (marked **R3.1**) regarding the state space, action space, and transition dynamics to accurately formalize our approach.
>
> ## R 3.2 Regarding Hyperparameters
>
> As our aim with PrismBench is to provide an evaluation **framework**, the tuneable parameters presented in Table 1 are _configuration parameters_ to control different benchmarking aspects:
>   - Traversal parameters control how evaluation budget is allocated across the state space, determining which states are sampled and their sampling frequency under a fixed budget (_Sec. 3.4.1_).
>   - Benchmark configuration parameters define what the evaluation prioritizes (_Sec. 3.4.2_). Adjusting them corresponds to different evaluation objectives, e.g., emphasizing program repair versus strict test correctness.
>
> We do not claim a single “correct” configuration, as different configurations correspond to different evaluation objectives. In our experiments, we fix one configuration (_Sec. 4.4)_ and evaluate all models under the same setting to ensure direct comparability.
>
> We thank you for raising this issue as establishing comparability is an important aspect of our work. To clarify these parameters and their effects, we have substantially revised _Sec. 4.4_ (marked **R3.2**) to detail each parameter’s role and to emphasize that results are comparable only under the same settings.
>
> ## R 3.3 Regarding “gaming” via Hyperparameter Optimization
>
> We agree that per-model tuning would undermine trust. As mentioned above, optimizing these parameters does not improving performance under a fixed benchmark, but instead results in selecting a different evaluation objective and/or reallocation of evaluation budget, which breaks comparability. For example, the results of a benchmark with a single solution per challenge (pass@1) cannot be compared to a benchmark with ten solutions per challenge (pass@10). Therefore, evaluations using different configurations are only comparable to other models evaluated under the same configuration.
>
> Additionally, PrismBench’s rewards are grounded in **execution outcomes** of generated solutions/tests (_Sec. 3.3_). Under any fixed configuration, a model cannot increase its score without producing artifacts that execute and pass. Changing traversal settings only affects which states are sampled under a budget and changing benchmark settings only changes which aspect of programming (solution generation, test generation, program repair) is weighted more. These changes cannot produce superficial success.
>
> We kindly direct you to the revised _Sec. 3.5_ where we have included substantially more discussion on how we control for benchmark validity.

---

> > ### Comment · Reviewer_bTMW · 2026-02-05
> >
> > I see. Now I understand the contribution a bit better. I agree that in a given state it's impossible for a model to perform better by "gaming" the system, but the combination of states probed and their frequency of visit does affect an overall score. So I see the contribution mainly as a way to discover strenghts and weaknesses of models, but less about overall performance comparison - because a different hyperparameter choice will lead to a different state probing pattern. A competitor can always claim a leaderboard unjustifiably favors their competitor because of a certain choice of hyperparameters.
> >
> > I do believe that such as system is valuable whenever a more fine-grained "interrogation" of various performance aspects of a model are required.
> >
> > So I believe the weaker fit of the system as a leaderboard score should be written explicitly as a limitation in the paper.

---

> > > ### Author Response · Authors · 2026-02-09
> > >
> > > Dear Reviewer bTMW, thank you for your follow-up and for taking the time to engage with our work. We appreciate your comment and will further highlight the role of experimental settings (_Sec. 4.4_) on the overall performance comparison between models as a threat to external validity in _Sec. 7_.

---

### Review · Reviewer_9Prt · 2026-01-09

**Summary Of Contributions:**

This paper introduces PrismBench, a dynamic benchmarking framework for evaluating LLM code generation. It formulates evaluation as a search problem over a concept–difficulty space and uses Monte Carlo Tree Search (MCTS) with execution-based rewards to adaptively generate challenges and identify model weaknesses. By decomposing evaluation into multiple agents and relying on sandbox execution rather than LLM judges, the framework aims to provide finer-grained, more robust diagnostics than static benchmarks.

Key Strengths:
1. The application of MCTS to dynamically explore specific weaknesses rather than using a fixed dataset is a methodological advancement.
2. By relying on sandbox execution results instead of LLM-based judging, the framework avoids the subjective biases common in "LLM-as-a-judge" approaches.

Key Weaknesses:
1. The multi-agent interaction and MCTS iterations require significantly more computational resources and setup time compared to static benchmarks.
2. The validity of the benchmark heavily relies on the "Generator" agent's ability to produce high-quality, solvable, and correct problem definitions.

**Audience:**

Yes

**Audience Explanation:**

The paper will be of interest to TMLR readers working on LLMs, evaluation methodology, and ml systems. Its dynamic benchmarking framework and use of search-based evaluation offer a novel perspective on diagnosing model capabilities beyond static benchmarks, which is relevant to ongoing research on reliable and robust model evaluation.

**Broader Impact Concerns:**

The paper primarily focuses on improving evaluation methodology for LLM code generation and therefore presents low direct ethical risk.

**Claims And Evidence:**

Yes

**Claims Explanation:**

The submission supports its claims with strong empirical evidence. The authors validate the PrismBench framework by testing it on eight frontier models, demonstrating its ability to expose specific failure modes that static benchmarks miss due to saturation. The efficacy of the MCTS algorithm in dynamically navigating the difficulty landscape is visually and quantitatively supported by the generated search trees and success rate heatmaps. Furthermore, the reliance on sandbox-based execution rather than LLM-as-a-judge provides accurate, objective ground truth for the validity of the generated code, substantiating the claim of a  bias-free evaluation pipeline.

**Requested Changes:**

Critical:
1. While Section 4.3 states that results are averaged over 3 independent runs, the reported confidence intervals (Tables 4 & 5) show significant variance (e.g., $\pm 0.11$). The authors should explicitly analyze the root cause of this instability: does it stem from the stochastic nature of the MCTS exploration path or the evaluated models themselves? Crucially, the authors should discuss whether this variance is large enough to alter the relative rankings of models across different runs.

Strengthening:
1. The paper does not include explicit ablation studies; adding targeted ablations on key design choices (e.g., MCTS parameters, reward formulation, or agent roles) would strengthen the empirical grounding of the framework.
2. While Section 4.4 provides absolute financial costs, a comparative analysis is missing. The authors should discuss the "cost-to-insight" ratio: does the high cost of PrismBench yield proportionally more valuable insights compared to low-cost/free static benchmarks? Additionally, reporting runtime/latency metrics (time to complete a benchmark run) would help users assess the practicality of the framework.

---

> ### Author Response · Authors · 2026-01-22
>
> Dear reviewer 9Prt, thank you for your thoughtful review and thorough comments. We sincerely appreciate your feedback as they have allowed us to improve our submission. In response to your concerns we have revised our manuscript. All changes are numbered according to response number and highlighted within the manuscript in blue color. Below, we respond to your concerns point by point.
>
> ## R 2.1 Regarding the “Generator” Agent
>
> We agree that LLM-based challenge generation at test time can undermine evaluation’s validity. Our aim with doing so was to lower the chances of memorization, which we discuss in detail in *Secs. 2*, and *6*. Importantly, we also explicitly discuss the risks of dynamic challenge generation, our mitigation strategies, and associated limitations in *Secs. 3.5*, and *7*.
>
> We appreciate you highlighting this issue. In response we have added *Sec. 3.5.3* (marked **R2.1**) to make this discussion more explicit and detailed.
>
> ## R 2.2 Regarding Variance Between Runs
>
> Given our methodology, there are two sources of stochasticity:
>   - **LLM stochasticity:** While our approach supports enforcing deterministic sampling parameters, we **intentionally** preserve recommended range settings for our experiments as previous studies have shown that doing so, systematically reduces output diversity and risks underestimating true performance boundaries (*Sec. 3.4.1*).
>   - **Traversal policy:** Given the ε-greedy policies alongside MCTS’s exploration constant and random selection of nodes with the same value, different runs may produce different search trajectories (*Sec. 3.2.2*).
>
> We control for both sources at multiple points in the evaluation process:
>   - Incremental node value updates with TD(0) (Eq 4) smooth out outlier observations (incidental success/failures).
>   - Discounted reward backpropagation through all ancestor nodes (Eq 5) aggregate performance signals throughout the search trajectory so value estimates are not isolated to individual observations.
>   - Phase transitions happen only after node value changes remain within threshold across repeated checks (*Sec. 3.4*). Therefore, evaluation phases proceed only after performance estimates have stabilized across the tree.
>   - MCTS and ε-greedy policies for state selection (Eqs 13-14) result in multiple revisitation of nodes, which in addition to TD(0) value estimations, average out stochastic variations between search trajectories (*Secs. 3.2.2* and *3.4.1*).
>
> Therefore, while individual node values or search trajectories may vary between runs (as reflected in the CIs and discussed in *Sec. 7*), the aggregated tree-level metrics remain comparable and can be further stabilized by deterministic sampling. We also provide per model and cross-model analyses of these results alongside the root causes of observed differences in *Sec. 5.2*.
>
> We thank you for raising this important issue. In response, we have substantially revised *Sec. 3.5* (marked as **R2.2**) where we have consolidated and added discussions on the sources of stochasticity, bias, and variance in our proposed approach and discuss in detail how we address each one.
>
> ## R 2.3 Regarding Cost-to-Insight Ratio
>
> We appreciate your suggestion for a comparative analysis. Our proposed metrics are directly motivated by the strengths and shortcomings of existing approaches, which we discuss in detail in *Secs. 2* and *6*.
>
> In response to your suggestion, we have added *Sec 3.6.5* (marked **R2.3**) to explicitly map our proposed metrics to current standard coding benchmarking metrics, and to clarify the additional insights provided by reporting these metrics across *(concept, difficulty)* pairs across the tree.
>
> Additionally, we kindly direct you to our leaderboard at: <https://prismbench.github.io/Demo/> where we include much more comprehensive metrics and analyses per model which would not fit the contents of the paper.
>
> ## R 2.4 Regarding Run time
>
> We thank you for your suggestion. We have included wall-clock runtimes (end-to-end) for each evaluated model alongside the hardware spec used for the experiments in *Sec. 4.5* of the revised manuscript (marked as **R2.4**).
>
> ## R 2.5 Regarding Ablation Studies
>
> We acknowledge your concern regarding ablation studies. However, we would like to highlight:
> - The tenable parameters in Table 1, control different aspects of the evaluation process and different configurations result in different evaluation objectives. We keep these them fixed across all experiments so that the results are comparable. Furthermore, we have revised *Sec. 4.4* to explain them in more detail (marked **R3.2**) in response to a similar comment from reviewer bTMW.
> - We evaluate different agent roles both in isolation (agents do not have access to each others’ output) and in concert (execution of the agents’ outputs). For each model, we report dedicated metrics and detailed analyses per role in *Sec. 5.2*.

---

### Review · Reviewer_dfij · 2026-01-12

**Summary Of Contributions:**

The paper presents PrismBench, a dynamic benchmark for LLM coding capabilities. Key features include: 1) Model the coding capacities of LLM as a space of programming concepts and difficulties for evaluation. 2) MCTS-guided exploration of the state space which dynamically allocates budget for the evaluation based on the model’s current performance to maximize information gain of the trials. 3) A three-phase multi-agent pipeline that decomposes the benchmark that generates coding challenge dynamically, performs comprehensive evaluation of model answer, and identify systematic issues of the model dynamically.

Strength:
1. This paper uses a novel formation of Markov decision process (MDP) of the evaluation, and uses a multi-agent workflow to overcomes the limitation of static benchmark and dynamic benchmark.
2. The paper conducted extensive evaluation using the proposed benchmark on different models and examined different metrics in details, showcase the ability to adaptively evaluate different models on different programming concepts and difficulties and finding the systematic issues of the model.

Weakness:
1. This work generates leetcode-style coding challenges that focus on one or more programming concepts. While the result can be a proxy, the model's actual performance on real work problems is not clear.
2. The benchmark relies on LLM-based agents, which may introduces variance and bias. In Sec 4.3 and table 2, the paper suggests using L-405b to avoid bias when evaluating 4o. However, the different models are evaluated with different LLM-based agents, it can be hard to directly compare the results.

**Audience:**

Yes

**Audience Explanation:**

LLM coding has been getting great importance in research and practical use. It is important to develop systematic benchmark to understand the model capabilities and further the model development. This work presents a systematic benchmark that does comprehensive evaluation of the LLM coding capabilities.

**Broader Impact Concerns:**

There are few ethical concern in this paper.

**Claims And Evidence:**

Yes

**Claims Explanation:**

The paper proposed to formalize the evaluation as a MCTS-guided exploration over a concept-difficulty tree space, and uses a three-phase multi-agent pipeline for comprehensive evaluation. Section 5 provides extensive experiments results of the proposed benchmark methodology over different LLMs. By studying different metrics in Sec 5.1, the paper shows that the MCTS exploration is able to adaptively allocate the budget and finds the node (concept-difficulty pair) to evaluate that is challenging to the model. Sec 5.2 and 5.3 provides further analysis of how different models perform on the proposed benchmark, showing that the proposed benchmark is able to reveal the strength and limitation of the coding capabilities of different models.

**Requested Changes:**

1. Clarify how the evaluation results on leetcode-style challenges generate to real work problems.
2. Clarify how the evaluation bias/variance introduced by LLM-based agents (see weakness 2) can be mitigated.

---

> ### Author Response · Authors · 2026-01-22
>
> Dear reviewer dfij, thank you for your thoughtful review and thorough comments. We sincerely appreciate your feedback as they have allowed us to improve our submission. In response to your concerns we have revised our manuscript. All changes are numbered according to response number and highlighted within the manuscript in blue color. Below, we respond to your concerns point by point.
>
> ## R1.1 LC-style Challenges and Real-World Transferability
>
> We agree that LC-style problems are only a proxy. PrismBench is designed as a **benchmarking methodology**, not a fixed dataset and our aim is to evaluate LLMs’ capabilities beyond single‑shot code generation to mirror real development workflows. We use LC-style challenges for our experiments as an **instantiation** of end-to-end programming as they are well-established and commonly used for evaluating SEs on their coding capabilities. Specifically:
>   1. Real-world programming rarely involves isolated concepts. Instead, a correct solution requires combination of multiple strategies. Therefore, the evaluation space is defined over concept sets and difficulty levels (*Sec. 3.2.1*). LC-style challenges fit within this formulation.
>   2. Each node is evaluated end‑to‑end. Similar to real-world problems, the model must interpret a natural‑language spec, generate tests and a solution in isolation, execute, and iteratively repair based on run feedback (*Sec. 3.3*). LC-style challenges provide a natural fit for this purpose.
>
> Importantly, in our approach the evaluation space is modeled over concepts and difficulty sets, not the challenges themselves. Therefore, our methodology can be applied to other coding tasks categorizable by concepts and difficulty. For example, similar to SWE-bench, challenges could be annotated GitHub issues where the challenge involves generating a correct fix, running the existing test suite, adding additional tests, and incorporating feedback from test failures. We provide examples in our replication package: <https://github.com/PrismBench/PrismBench/wiki>
>
> We thank you for your suggestion and have further clarified this distinction in the revised manuscript (marked **R1.1**):
> - We have added *Sec. 4.1* (Problem Formulation) to clarify the usage of LC-style challenges for our experiments.
> - We have further clarified that the experimental results might not transfer to real-world coding capabilities in *Sec. 7*‘s threats to external validity.
>
> ## R1.2 Mitigating Bias/Variance from LLM-based Agents
>
> We acknowledge the concern regarding using LLMs as judges. However, we would like to clarify that **PrismBench does not use LLMs as judges for evaluation**. As mentioned in *Secs. 1*, *3.1*, and *3.3*, the entire problem-solving workflow (solution/test generation and program repair) is handled only by the model under evaluation. Specifically:
> 1. All rewards are derived from execution signals in the sandbox (*Sec. 3.4*). These rewards are used to calculate each node’s value and traversal. No LLM judgement is incorporated into the rewards and does not influence the benchmarking process.
> 2. The analyzer agents mentioned by the reviewer serve purely **post-hoc diagnostic roles**. These agents can be entirely removed without affecting evaluation results (*Sec. 4.4*).
> 3. The change between the **analyzer agents** for diagnostic analyses of GPT-4o and L-405b, was done to mitigate the bias of a model evaluating its own solutions/tests (*Sec. 4.4*).
>
> We thank you for your suggestion regarding this clarification, in the revised manuscript:
> - We have added *Sec. 3.5.5* (No Reliance on LLM Judgments) to further clarify the evaluation process, distinguish the usage of analyzer agents, and their roles (marked **R1.2**).
> - Additionally, in response to reviewer 9Prt’s comments (marked **R2.2**), we have substantially revised *Sec. 3.5* to further detail how we control for bias/variance from LLM agents throughout the evaluation process.

---

### Review · Reviewer_Mu9G · 2026-01-18

**Summary Of Contributions:**

The paper proposes PrismBench, a framework for evaluating LLM code generation that moves beyond static datasets. It employs a Reinforcement Learning approach, specifically modeling the evaluation as a Markov Decision Process traversed via Monte Carlo Tree Search. The system uses a multi-agent architecture to dynamically generate problems and test suites based on a concept-difficulty tree. This allows the benchmark to adaptively focus on the frontier of a model's capabilities, spending more compute on areas where the model struggles, and providing fine-grained diagnostic metrics regarding specific error types and concept mastery.

Strength:
1. Unlike typical benchmarks that yield a single pass rate, PrismBench offers deep diagnostic insights. By categorizing failure modes and mapping them to specific concept combinations, it provides actionable feedback for model developers rather than just a leaderboard ranking.
2. The use of MCTS to guide the evaluation is a smart allocation of resources. Instead of wasting evaluation budget on trivial tasks that a model has mastered, the framework naturally converges on the boundary of the model's capabilities, providing a higher resolution view of the model's limitations.

Weakness:
1. The experimental setup uses GPT-4o-mini as the Challenge Designer to benchmark much larger and more capable models like GPT-4o and Llama-3.1-405b. There is a significant risk that the smaller model cannot generate very hard problems that are conceptually difficult enough to truly test the reasoning limits of the larger models. The difficulty measured might stem from ambiguity or poor problem formulation by the weaker designer, rather than true algorithmic complexity.
2. The paper claims to avoid the pitfalls of "LLM-as-judge" by using sandbox execution. However, the validity of an execution-based reward is entirely dependent on the correctness of the LLM-generated test suites. If the test generator agent produces tests with false positives or false negatives, the objective execution signal becomes a noisy, probabilistic metric. The reliance on LLMs to define the ground truth reintroduces the very subjectivity the authors aim to avoid.

**Audience:**

Yes

**Audience Explanation:**

The shift from static to dynamic benchmarking is a critical area of research as models saturate current datasets. The methodology of applying MCTS to evaluation coverage is novel and relevant to researchers focusing on evaluation rigor and model debugging.

**Broader Impact Concerns:**

No significant ethical concerns. The work focuses on improving technical evaluation standards.

**Claims And Evidence:**

Yes

**Claims Explanation:**

The paper provides extensive experimental data across 8 models and demonstrates that the MCTS approach effectively grows the tree in different ways for different models, supporting the claim that it adapts to model capability. The separation of error analysis provides clear evidence of the framework's diagnostic utility. However, the claim of "objective execution-based signals" is slightly overstated given the dependence on generated tests (as noted in Weakness 2), but the overall empirical results support the utility of the framework.

**Requested Changes:**

1. Please provide an audit or study quantifying the accuracy of the generated test suites. Specifically, what is the rate of incorrect assertions generated by the Test Generator? The test validator agent is mentioned, but since it is also an LLM, a human verification or cross-check is needed to establish trust in the reward signal.
2. Address the potential ceiling effect of using GPT-4o-mini. Provide evidence that very hard problems generated by 4o-mini are indeed algorithmically complex and distinct from medium problems, rather than just being longer or more vaguely specified.
3. To contextualize the results, please show how the rankings or concept-mastery scores derived from PrismBench correlate with established static benchmarks or live leaderboards. This would help validate that PrismBench is measuring the same underlying coding capability, but with higher resolution.

---

> ### Author Response · Authors · 2026-01-22
>
> Dear reviewer Mu9G, thank you for your thoughtful review and thorough comments. We sincerely appreciate your feedback as they have allowed us to improve our submission. In response to your concerns we have revised our manuscript. All changes are numbered according to response number and highlighted within the manuscript in blue color. Below, we respond to your concerns point by point.
>
> ## R 4.1 Regarding the Test Suits and Objective Execution
>
> We agree that test reliability is critical, as tests directly shape the rewards. This is precisely why test generation is an explicit part of evaluation: the target model **must** generate tests that can evaluate **any** valid solution without access to the solution’s code (_Sec. 3.3_). This controls two factors simultaneously:
>   - Because the model never sees the solution code, it cannot cheat by tailoring tests to a specific solution.
>   - To produce valid tests, the model must understand the problem. If it misunderstands the spec or reasons incorrectly about what a correct solution may be, it will generate tests that either fail during execution or fail to comprehensively validate it.
>
> As discussed in _Sec. 3.4_, each phase’s reward is designed to maximize information gain by combining multiple execution-derived signals (not LLM judgments), including test-suite signals, rather than a binary outcome. As such, there are two major failure modes:
> - **Incorrect tests or solutions (false negative)**: A correct solution fails due to invalid tests, or an incorrect solution fails even when tests are correct. Both result in execution failures and low reward, which is backpropagated via TD (0), lowering ancestor values. If this happens consistently, these states are automatically selected for evaluation in Phases 2–3 due to their low values.
> - **Misaligned tests or solutions (false positive)**: Insufficient/incorrect tests allow a broken/incorrect solution to pass. Such states will initially receive high rewards and get expanded into finer-grained states until their value stops improving. As the model encounters harder in these expanded states, failures backpropagate and reduce ancestor values, resulting in diminishing returns for that branch. These states are then automatically evaluated further in Phases 2–3 due to their low values.
>
> Therefore, manual auditing of generated test suites is not required: both false positives and false negatives are accounted for by the environment. When the tests/solutions are unreliable, rewards become inconsistent with prior observations along the trajectory, signalling that the model in incapable of generating valid tests/solutions at the given (concept, difficulty). While we acknowledge undetected errors may still occur, repeated test-solution execution and multi-phase revisits minimize them. We provide empirical evidence of this mechanism in _Sec. 5.2_ (pp. 35–39), where flawed test suites lead to measurable reward divergence and triggered diagnostic analysis.
>
> We thank you for raising this important issue. In response, we substantially revised _Sec. 3.5_ to include a detailed discussion of this process (marked **R2.2** and **R4.1**). Regarding the Test Validator agent, we kindly direct you to _Sec. 3.5.5_ (marked **R1.2**) which clarifies the analyzer agents’ roles in response to a similar concern from reviewer dfij.
>
> ## R 4.2 Regarding Using 4o-mini
>
> We acknowledge that the generator’s notion of difficulty may not always align with human judgment, and some challenges may vary in quality. We explicitly discuss these risks in _Sec. 3.5_ and list them as threats to validity in _Sec. 7_.
>
> In short, LLM-based challenge generation introduces two risks:
> - Invalid challenge: Generated challenge is off-concept or contains infeasible/contradictory constraints.
> - Miscalibrated challenge: The actual difficulty of the challenge does not match its intended difficulty (e.g., a challenge generated for hard, actually being medium).
>
> Both cases result in invalid challenges and can invalidate evaluation results. We kindly direct you to _Secs. 3.5.3_ and _3.5.4_ where we discuss these risks and how we mitigate them in detail. We used 4o-mini as the challenge designer for comparability. Since challenge generation depends only on the state’s (concept, difficulty) pair and not on the evaluated model, all models are evaluated under the same generation policy.
>
> We also provide empirical evidence of these mitigation mechanisms: in both _Secs. 5.2_ and _5.3_, trees differentiate model scale and capability in a structured manner. For instance, 4o/GPT-OSS reach deeper depths (“hard/very hard”), while L4s or smaller Llama models remain in shallower depths (“easy/medium” and “medium/hard”), indicating that higher-difficulty regions are measurably harder under execution.
>
> ## R 4.3 Regarding Contextualization
>
> We thank you for this suggestion. We will add a comparison between our approach’s results with SWE-bench and LiveCodeBench to our leaderboard.

---

### Decision · Action_Editor_isST · 2026-02-26

**Recommendation:** Accept as is

**Audience:**

Yes

**Audience Explanation:**

The proposed paper deals with code generation benchmarking LLM, a highly relevant topic for the AI community nowadays.

**Claims And Evidence:**

Yes

**Claims Explanation:**

The paper introduces PrismBench, a framework for evaluating LLM-based code generation that goes beyond traditional static benchmarks. During the rebuttal phase, the authors clarified key points and addressed the reviewers’ concerns, leading to a consensus that the contribution is meaningful and suitable for publication.

Several reviewers noted that the manuscript would benefit from a clearer presentation of the work’s contributions, strengths, and limitations. I strongly encourage the authors to make these aspects more explicit in the final version.